# Endogenous IL-1 receptor antagonist restricts healthy and malignant myeloproliferation

Alicia Villatoro[1,10], Vincent Cuminetti [1,10], Aurora Bernal [1], Carlos Torroja [2], Itziar Cossío[3], Alberto Benguría[4], Marc Ferré[1], Joanna Konieczny[1], Enrique Vázquez [4], Andrea Rubio[3], Peter Utnes[1], Almudena Tello[1], Xiaona You [5], Christopher G. Fenton[6], Ruth H. Paulssen [6,7], Jing Zhang [5], Fátima Sánchez-Cabo [2], Ana Dopazo [4], Anders Vik[8], Endre Anderssen [6], Andrés Hidalgo[3] & Lorena Arranz [1,9] ✉

Here we explored the role of interleukin-1β (IL-1β) repressor cytokine, IL-1 receptor antagonist (IL-1rn), in both healthy and abnormal hematopoiesis. Low IL-1RN is frequent in acute myeloid leukemia (AML) patients and represents a prognostic marker of reduced survival. Treatments with IL-1RN and the IL-1β monoclonal antibody canakinumab reduce the expansion of leukemic cells, including CD34+ progenitors, in AML xenografts. In vivo deletion of IL-1rn induces hematopoietic stem cell (HSC) differentiation into the myeloid lineage and hampers B cell development via transcriptional activation of myeloid differentiation pathways dependent on NFκB. Low IL-1rn is present in an experimental model of pre-leukemic myelopoiesis, and IL-1rn deletion promotes myeloproliferation, which relies on the bone marrow hematopoietic and stromal compartments. Conversely, IL-1rn protects against pre-leukemic myelopoiesis. Our data reveal that HSC differentiation is controlled by balanced IL-1β/IL-1rn levels under steady-state, and that loss of repression of IL-1β signaling may underlie pre-leukemic lesion and AML progression.

IL-1β stands out as initiator of inflammation and blocking its activity in humans is applied in clinical treatments against diseases with an inflammatory component[1]. When dysregulated, chronic inflammation through autoimmune conditions and infections is linked to increased risk of hematological malignancies[1]. Chronic IL-1β administration biases differentiation of normal HSC to the myeloid lineage and reduces their self-renewal properties through a cell-autonomous effect[2–4].

However, IL-1 receptor 1 knockout (IL-1r1-KO) mice have unaffected blood production and normal stem and progenitor bone marrow (BM) compartments, suggesting that IL-1β-induced myeloid priming of HSC occurs under conditions of injury or infection only and tonic IL-1 signaling has none or small basal hematopoietic effects[2,5,6]. In turn, enhanced IL-1β and other members of its signaling pathway, including IL-1RAP, MyD88, IRAK1, IRAK4 and TRAF6, have been involved in AML

[1]Stem Cells, Ageing and Cancer Research Group, Department of Medical Biology, Faculty of Health Sciences, UiT – The Arctic University of Norway, MH2 building level 10, 9019 Tromsø, Norway. [2]Bioinformatics Unit, Fundación Centro Nacional de Investigaciones Cardiovasculares (CNIC), 28029 Madrid, Spain. [3]Area of Cell and Developmental Biology, CNIC, 28029 Madrid, Spain. [4]Genomics Unit, CNIC, 28029 Madrid, Spain. [5]McArdle Laboratory for Cancer Research, University of Wisconsin-Madison, Madison, WI 53705, USA. [6]Genomics Support Center Tromsø, Faculty of Health Sciences, UiT – The Arctic University of Norway, MH building level 9, 9019 Tromsø, Norway. [7]Clinical Bioinformatics Research Group, Department of Clinical Medicine, Faculty of Health Sciences, UiT – The Arctic University of Norway, MH building level 9, 9019 Tromsø, Norway. [8]Department of Hematology, University Hospital of North Norway, 9019 Tromsø, Norway. [9]Associate Investigator, Norwegian Center for Molecular Medicine (NCMM), University of Oslo, 0349 Oslo, Norway. [10]These authors contributed equally: Alicia Villatoro, Vincent Cuminetti. ✉e-mail: lorena.arranz@uit.no

and/or myelodysplastic syndrome (MDS)[6–14]. Furthermore, altered inflammation affects cells from the BM microenvironment, and thereby contributes to malignant hematopoiesis[8]. In this regard, we reported a causal association between high IL-1β-induced damage to the HSC microenvironment and the onset of myeloproliferative neoplasms (MPN)[15].

Hence, IL-1β blockade shows promising therapeutic value in experimental models of MPN, chronic myeloid leukemia (CML), juvenile myelomonocytic leukemia, and AML[10,15–17]. Currently, IL-1β blockade with the human monoclonal antibody (mAb) canakinumab is being tested in a Phase II clinical trial for the treatment of low or intermediate risk MDS and CML (NTC04239157). Another interesting IL-1 blocking agent is the FDA-approved anakinra, the recombinant form of the naturally occurring IL-1 receptor antagonist (IL-1RN), which competes with IL-1 for IL-1r1. Anakinra is currently used as therapy against autoimmune diseases[1] and it has shown therapeutic value in experimental models of MPN and CML[15,17]. However, little is known about the participation of the endogenous IL-1 repressor IL-1RN in healthy and/or malignant hematopoiesis. IL-1RN production follows after IL-1 by roughly the same cell types of hematopoietic and non-hematopoietic origin[18]. In the BM, these include cells from the BM microenvironment, like BM Nestin⁺ mesenchymal stromal cells (MSC)[15] that represent a niche component that controls HSC function[15,19–22]. Despite these intriguing precedents, it is unclear to what extent repression of the IL-1-signaling pathway by endogenous IL-1RN may influence HSC behavior in healthy and diseased hematopoiesis.

Here, we find that low IL-1RN has prognostic value of poor survival in AML patients, it is a common event in AML patients, and it characterizes the lower maturation/differentiation profiles according to the French-American-British (FAB) classification of AML (M0-M3). IL-1RN boost or IL-1β blockade have therapeutic potential for AML patients based on their low IL-1RN. To study the effect of low IL-1RN in the hematopoietic system, we used the IL-1rn-KO strain, which displays IL-1β-induced bias in HSC differentiation towards the myeloid lineage whilst hampering B cell development, via NFκB activation under steady-state conditions. This phenotype is reminiscent of pre-leukemic disease and becomes apparent in the absence of injury or infection. Thus, we demonstrate a critical role for balanced IL-1rn and IL-1β on steady-state HSC function in vivo. Low IL-1rn is present in an experimental *NRAS*^G12D model of biased pre-leukemic myelopoiesis and further loss of IL-1β repression through IL-1rn genetic deletion promotes myeloproliferation, which relies on the BM hematopoietic and stromal compartments. Conversely, treatment with exogenous IL-1rn reverts pre-leukemic myeloproliferation. Our data support that loss of repression of IL-1β through low IL-1RN may originate and worsen hematopoietic disease, and predicts poor survival in AML patients.

## Results

### Low IL-1RN predicts AML progression in patients

To study the expression of IL-1RN in human AML, we reanalyzed publicly available arrays from purified AML blasts in a cohort of 381 AML patients (GSE14468)[23–25] and found that low *IL1RN* was associated with reduced survival rate (Fig. 1a). We separately analyzed more differentiated AML profiles according to FAB and found that low *IL1RN* was a prognostic marker of reduced survival in M4-M5 AML patients ($n = 164$) (Fig. 1a). There were few M0-M3 AML patients with high *IL1RN* expression ($n = 7/217$). Indeed, we identified the lower maturation/differentiation M0-M3 patients according to FAB as those patients with lowest *IL1RN* expression, versus M4-M5 (Fig. 1b). M0-M3 patients displayed high *IL1B* to *IL1RN* ratio (Supplementary Fig. S1A). In a cohort of 19 matched-pair diagnosis-relapsed AML patients (dbGaP accession phs001027)[26], we correlated low *IL1RN* expression in RNA-Seq data from AML blasts with poorer prognosis (Supplementary Fig. S1B), and reduced expression of *IL1RN* at relapse versus diagnosis (Supplementary Fig. S1C).

We studied the potential contribution of CD34⁺ progenitors to unbalanced IL-1RN in AML in a cell-intrinsic fashion. Expression levels of *IL1RN* were reduced in circulating CD34⁺ progenitors from patients versus healthy controls (Fig. 1c), which was particularly evident in the more undifferentiated FAB categories of AML (M0−M3) (Fig. 1d). Gene expression ratios of all *IL1B*, *CASP1* and *IRAK1*, to *IL1RN* were higher in AML patients (Fig. 1e; Supplementary Fig. S1D), suggesting that reduced IL-1RN associates with activation of IL-1β pathway. This was confirmed at the protein level with higher ratio of IL-1β to IL-1RN in CD34⁺ progenitors from AML patients versus healthy donors (Fig. 1f). As previously described, we confirmed higher expression of *IL1RAP* gene and higher frequency of CD34⁺ progenitors expressing IL-1RAP in AML patients compared to healthy donors[13] (Supplementary Fig S1E, F).

As a surrogate of IL-1β signaling pathway activation, we studied NFκB activation by phospho-flow and nuclear translocation in CD34⁺ progenitors from AML patients and healthy controls. The percentage of peripheral blood (PB) CD34⁺ progenitors activated through NFκB activation was higher in AML patients (Supplementary Fig. S1G), as well as the nuclear translocation of the p50/p65 NFκB heterodimer (Fig. 1g; Supplementary Fig. S1H). To confirm the functional role of increased IL-1β signaling in human AML pathogenesis, immunodeficient NSG-SGM3 mice were transplanted with CD34⁺ progenitors isolated from the BM of AML patients, and treated in vivo with either human IL-1β or vehicle. IL-1β treatment promoted expansion of human AML cells in PB and BM of NSG-SGM3 mice (Fig. 1h, i). We compared the therapeutic potential for AML patients in NSG-SGM3 xenografts of boosting IL-1RN and blocking IL-1β. Short-term treatments with IL-1RN or with the human IL-1β mAb canakinumab were efficient in reducing the expansion of human leukemic cells, i.e. myeloid cells, to similar extents in the PB of NSG-SGM3 mice transplanted with CD34⁺ progenitors isolated from the BM of AML patient samples (Fig. 1j; Supplementary Fig. S1I). Long-term treatment with IL-1RN was followed up and confirmed reduced numbers of human leukemic cells, i.e. CD45⁺ leukocytes, CD33⁺ myeloid blasts and CD34⁺ progenitors in the BM of IL-1RN-treated NSG-SGM3 mice (Fig. 1k; Supplementary Fig. S1J). Thus, IL-1RN boost has therapeutic potential for AML patients.

### Deletion of IL-1rn induces myelopoiesis in the absence of immunogenic stimulus

To determine the role of IL-1rn deficiency in hematopoiesis, we characterized the IL-1rn-KO mouse strain, which has not been used before in hematopoiesis. Adult IL-1rn-KO mice had higher cellularity in BM, and increased circulating neutrophils with no abnormalities in spleen (Fig. 2a; Supplementary Fig. S2A, B). The abundance, fractions and proliferation of Lin⁻Sca-1⁺c-Kit⁺ LSK cells were similar between WT and IL-1rn-KO mice (Supplementary Fig. S2C). Fluorescence-activated cell sorting (FACS) analysis of the stem and progenitor cell subsets corresponding to HSC and multipotent progenitors MPP1-MPP5[27–29] revealed reduction of HSC and MPP5, and increase of MPP2 in the BM of IL-1rn-KO mice (Fig. 2b; Supplementary Fig. S2D). Absolute numbers of common lymphoid progenitors (CLP) and common myeloid progenitors (CMP) were unchanged, whereas megakaryocyte erythroid progenitors (MEP) and granulocyte-monocyte progenitors (GMP) were reduced in the BM of IL-1rn-KO versus WT mice (Supplementary Fig. S3A). Numbers of colonies formed ex vivo by hematopoietic stem and progenitor cells (HSPC) were higher in IL-1rn-KO mice (Fig. 2c). Analysis of IL-1rn-KO mice also revealed increased numbers of apoptotic cells in total BM (Supplementary Fig. S3B) but not in HSPC (Supplementary Fig. S3C). We found impact of IL-1rn abrogation in the differentiation bias of hematopoiesis towards the myeloid lineage with significant impairment of B lymphoid development (Fig. 2d; Supplementary Fig. S3D, E). These results indicate that IL-1rn represses HSPC differentiation into the myeloid lineage and enables B cell differentiation under steady-state conditions.

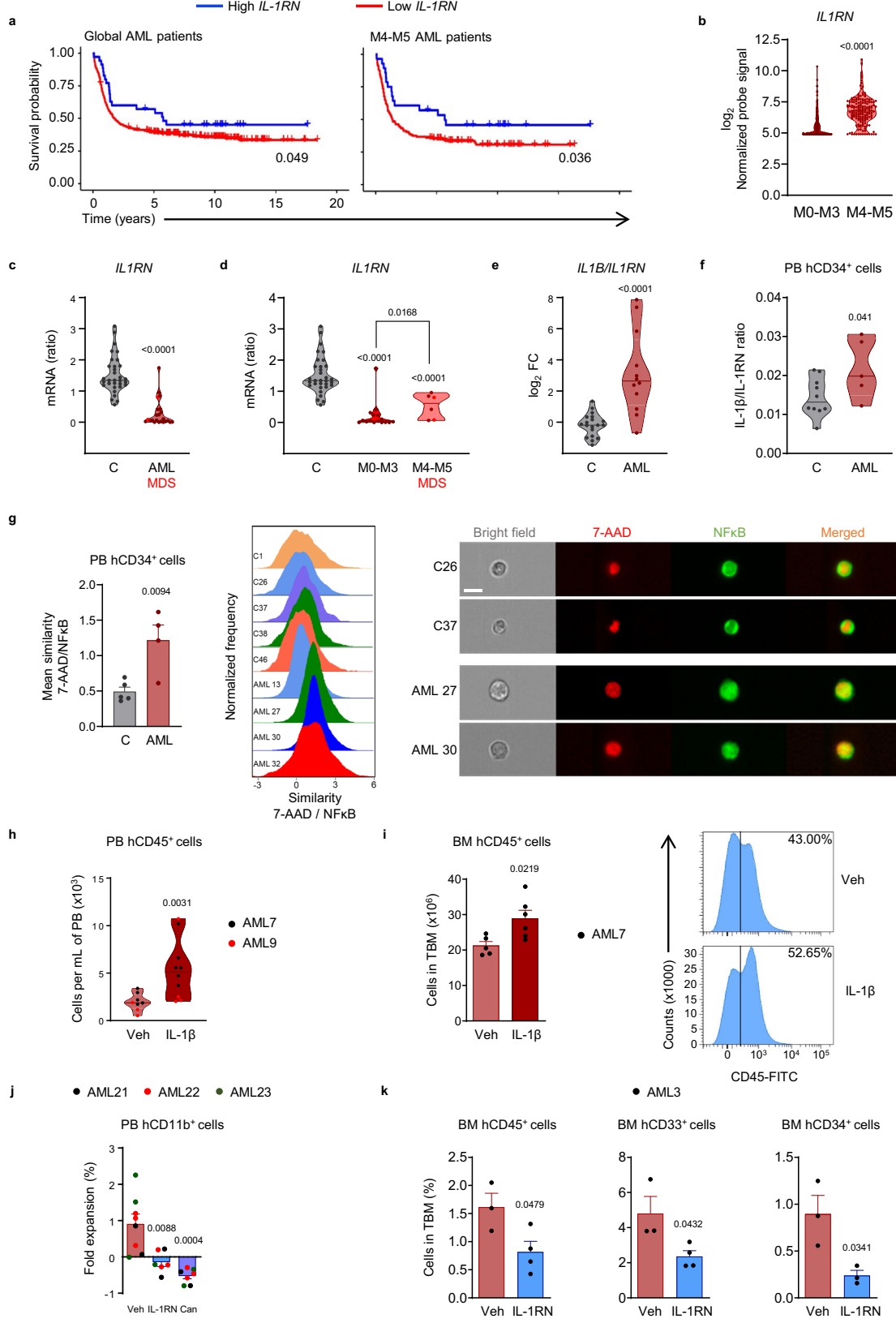

Production of IL-1β in the BM was described under conditions of injury or infection[2]. Our data showed presence of IL-1β in healthy BM and higher IL-1β BM levels in IL-1rn-KO versus WT mice resulting from the absence of IL-1rn, under steady-state conditions (Fig. 2e). There were no changes in other BM cytokines (Supplementary Fig. S3F). We then asked whether IL-1rn genetic deletion resulted in globally enhanced IL-1r1. IL-1r1-expressing cells in the BM were rare but increased in frequency in IL-1rn-KO mice (Fig. 2f). The myeloid bias of progenitors from IL-1rn-KO mice could be prevented in vitro (Supplementary Fig. S3G) and in vivo by treatment with IL-1RN (Fig. 2g, h) or mAb against IL-1β (Fig. 2i, j), but not mAb against IL-1α (Supplementary Fig. S3H). Thus, under steady-state conditions, IL-1rn prevents

**Fig. 1 | Low IL-1RN predicts AML progression in patients. a** Global AML patient survival data ($n = 381$) and M4-M5 AML patient survival data ($n = 164$) analyzed as a function of *IL1RN* expression in AML blasts (GSE14468). *p* value indicates likelihood ratio test. **b** mRNA expression of *IL1RN* in AML blasts (GSE14468) comparing AML subtypes M0-M3 ($n = 224$) versus M4-M5 ($n = 171$). **c** qRT-PCR mRNA expression of *IL1RN* in CD34⁺ progenitors from volunteers (C, $n = 32$) and patients (AML, $n = 21$; MDS, $n = 2$). **d** qRT-PCR mRNA expression of *IL1RN* in CD34⁺ progenitors from volunteers ($n = 32$) and AML subtypes M0-M3 ($n = 16$) and M4-M5 ($n = 4$). MDS together with AML M4-M5 group ($n = 2$). **e** qRT-PCR mRNA expression of *IL1B* versus *IL1RN*, in CD34⁺ progenitors from volunteers ($n = 19$) and AML patients ($n = 12$). FC, fold change. **f** IL-1β versus IL-1RN protein levels in CD34⁺ progenitors from volunteers ($n = 10$) and AML patients ($n = 5$). **g** NFκB nuclear translocation in CD34⁺ progenitors from volunteers ($n = 5$) and AML patients ($n = 4$) (left). Representative histograms (middle) and representative images (right). Scale bar, 10 μm. (**h**–**k**) AML xenografts. **h, i** In vivo treatment with IL-1β or vehicle (Veh). **h** Number of human CD45⁺ hematopoietic cells in peripheral blood (PB) from mice treated with vehicle ($n = 8$) or IL-1β ($n = 10$), over 2 independent experiments. **i** Number of human CD45⁺ hematopoietic cells in total bone marrow (TBM) from mice treated with vehicle ($n = 5$) or IL-1β ($n = 6$) and representative fluorescence-activated cell sorting histograms (cells in TBM, %). **j, k** In vivo IL-1RN boost or IL-1β blockade versus vehicle. **j** Expansion of human CD11b⁺ hematopoietic cells in PB from mice treated with vehicle ($n = 8$), IL-1RN ($n = 6$) or canakinumab (Can, $n = 7$), over 3 independent experiments. **k** Frequency of human CD45⁺, CD33⁺ and CD34⁺ hematopoietic cells in TBM from mice treated with vehicle ($n = 3$) or IL-1RN ($n = 4$). Data are biologically independent samples or animals, and means ± SEM for bar plots or medians for violin plots. Statistical analyses were performed with two-tailed Cox proportional hazard regression adjusted for age and stratified by cytogenetic risk (**a**), two-tailed Mann–Whitney *U* test (**b**–**e**, **h**) or two-tailed Student's *t*-test (**f, g, i**–**k**). *p* values ≤ 0.05 are reported. Source data are provided as a Source Data file.

IL-1β-driven HSPC differentiation into the myeloid lineage at the expense of lymphoid development.

Both IL-1rn-KO long term (LT)-HSC and short term (ST)-HSC upregulated the expression of *Il1b* gene, whilst *Il1r1* was expressed by HSPC and higher in LT-HSC from IL-1rn-KO versus WT mice (Fig. 2k). This suggested engagement of IL-1β-positive feedback loop for IL-1β expression[30] in HSPC in the absence of any immunogenic trigger, due to IL-1rn loss. *Il1rap* gene expression was unchanged in LT-HSC and ST-HSC but reduced in MPP from IL-1rn-KO versus WT mice (Fig. 2k). Myeloid cells, selectively granulocytes, were expanded in the BM of IL-1rn-KO mice (Fig. 2d; Supplementary Fig. S3E) but, unlike HSPC and similarly to monocytes, they showed no engagement of IL-1β-positive feedback loop through IL-1r1 (Fig. 2l). *Il1rap* gene expression was similar in granulocytes and monocytes from IL-1rn-KO and WT mice (Fig. 2l). Compared to WT HSPC, mRNA levels of *Il1rn* were 170- and 20-fold higher in WT granulocytes and monocytes, respectively (Fig. 2m).

To identify specific changes in HSPC driven by IL-1rn loss, we next performed gene expression profiling by RNA sequencing (RNA-Seq) of WT and IL-1rn-KO LT-HSC, ST-HSC and MPP. We identified impact of IL-1rn abrogation on the transcriptional programs of the LT-HSC and the ST-HSC compartments, with 1433 and 2612 differentially expressed genes, respectively (adjusted $p < 0.05$). The effect on MPP was smaller, with 181 genes differentially expressed (Supplementary Fig. S4A). Gene set enrichment analysis (GSEA) for LT-HSC revealed coordinated changes in a variety of genes associated with the immune system, the immune response and leukocyte differentiation (Supplementary Fig. S4B, C). GSEA of ST-HSC dysregulated genes identified changes in similar gene programs (Supplementary Fig. S4D, E). These changes were related to increased NFκB transcription factor activity (Fig. 3a, b), and activation of myeloid differentiation genes including the transcription factor PU.1 (*Spi1*) and some of its target genes like *Cebpa* and *Csf2rb* (Fig. 3a, c) in IL-1rn-KO HSPC compared to WT HSPC, particularly in LT-HSC and ST-HSC. Various RNA-Seq hits involving genes related to myeloid differentiation (*Spi1, Csf2rb*) and genes previously found abnormally expressed in myeloid neoplasias (*Axl, Stat3, Tlr1*) were confirmed by qRT-PCR (Supplementary Fig. S4F–H)[31–33]. The myeloid bias of progenitors from IL-1rn-KO mice could be reverted in vivo by treatment with bortezomib, a proteasome inhibitor that prevents NFκB activation (Fig. 3d).

Myeloid differentiation bias at the expense of B cell development, reduced self-renewal of primitive HSC and expansion of committed MPP seem consistent hematopoietic effects of chronic (20 day) high dose (0.5 μg) IL-1β treatment across studies despite variability[2–4], and in vivo deletion of IL-1rn phenocopies these results under steady-state (Fig. 2b, d). IL-1β-induced hematopoietic effects are dependent on transcriptional programs activated by *Spi1* and *Cebpa*[2–4]. We compared our RNA-Seq datasets with these previous publicly available datasets studying the transcriptional programs activated by IL-1β treatment in C57BL/6 WT mice using LSK Flt3⁻CD48⁻CD150⁺ HSC in a native microenvironment (GSE165810)[3] or LSK Flt3⁻CD48⁺CD150⁻ MPP3, with or without YFP expression, in the context of busulfan conditioning and transplantation (GSE166629)[4]. We found a partial cell type-specific overlap, with a particularly remarkable transcriptional activation of shared genes induced both by IL-1rn deletion in LT-HSC and ST-HSC, and by IL-1β treatment in LSK Flt3⁻CD48⁻CD150⁺ HSC (Supplementary Fig. S4I, J). Of note, the upregulated genes common to all cell subsets and experimental conditions were enriched in NFκB targets, including *Cebpd, Csf2rb, Csf2rb2* or *Spi1* (Supplementary Fig. S4I, J).

Taken together, these results suggest that IL-1β and IL-1rn control myeloid output mainly through the LT-HSC and ST-HSC compartments and transcriptional control of myeloid differentiation pathways which are at least partially dependent on NFκB activation, with contribution of these cells to *Il1b* production in the absence of immunogenic stimuli. IL-1rn-KO mice develop an IL-1β-induced phenotype reminiscent of early hematopoietic disease.

## Deletion of IL-1rn causes damage to the BM stroma in the absence of immunogenic stimulus

Adult IL-1rn-KO mice showed incipient reticulin fibrosis despite non-detectable changes in collagen deposits in the BM (Fig. 4a; Supplementary Fig. S5A). BM MSC numbers, identified as stromal (CD45⁻CD31⁻Ter119⁻) cells positive for CD63[34], CD105 or CD105 Vcam-1[35], and *Nestin* expression[21,36], which partially overlap[21,34–36], were reduced in IL-1rn-KO mice (Fig. 4b; Supplementary Fig. S5B, C). Reanalysis of publicly available single-cell transcriptional data of mouse BM stromal populations (GSE108892)[34] showed that *Cd63* expression largely overlaps with expression of *Lepr, Cxcl12, Kitl* and *Il1rn* (Supplementary Fig. S5D). Therefore, BM CD63⁺ stromal cells were chosen for most follow-up studies. Reduced numbers of BM CD63⁺ cells coincided with increased levels of membrane IL-1r1 (Fig. 4c), suggesting potential IL-1β contribution. IL-1β-induced damage to the stromal compartment was confirmed by in vivo treatment with IL-1RN or mAb against IL-1β, which rescued numbers of BM CD63⁺ stromal cells in IL-1rn-KO mice (Fig. 4d, e).

To better understand BM MSC alterations, genome-wide expression was profiled by RNA-Seq in sorted CD45⁻CD31⁻Ter119⁻CD63⁺ cells. Some of the most downregulated genes in BM CD63⁺ cells derived from IL-1rn-KO mice included MSC- (*Lepr, Adipoq*) and HSC regulatory- (*Cxcl12, Vcam1, Angpt1*) genes (Fig. 4f). These changes were confirmed by qRT-PCR (Fig. 4g). qRT-PCR data also revealed engagement of IL-1β-positive feedback loop in BM CD63⁺ stromal cells from IL-1rn-KO mice through increased *Il1b* expression compared to WT BM CD63⁺ stromal cells and that these cells produce high levels of *Il1rn* in the BM of adult WT mice (Fig. 4g); 2.7-fold higher than WT granulocytes (Fig. 2m). Taken together, absence of IL-1rn leads to IL-1β-induced damage to the stromal compartment of the BM, with contribution of BM CD63⁺ stromal cells to *Il1b* production.

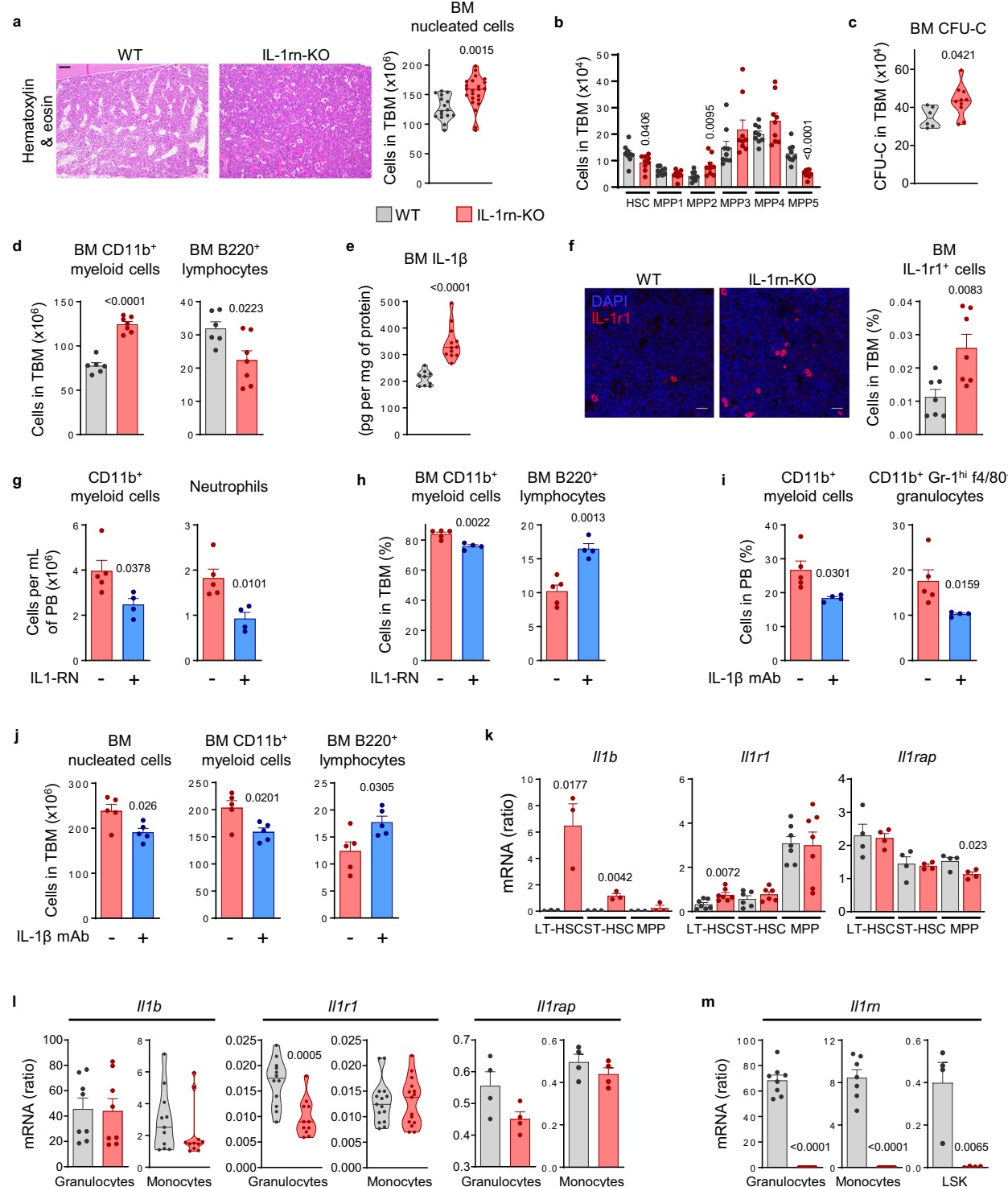

## Identification of the sources of IL-1β and IL-1rn in the bone marrow at the single-cell level

To provide further insights into the main sources of IL-1β and IL-1rn in the mouse BM at the single-cell level, we performed single-cell (sc) RNA-Seq and RNA fluorescent in situ hybridization (FISH) in sorted cells. After integrating the WT and IL-1rn-KO scRNA-Seq data from FACS-sorted CD11b+ cells by the Canonical Cross-correlation Analysis (CCA) algorithm of Seurat, unbiased cell clustering analysis was performed (Fig. 5a). WT cells were then used for annotation according to ImmGen[37] followed by application of the maximum enrichment score ±0.02 to label the clusters[29] using the cell type signatures imported

from GSE137539 dataset for neutrophil-like subsets[38] and according to GSE131834 dataset for monocyte-like clusters[39], and subsequent manual curation. This resulted in 10 clusters that encompass neutrophils at different stages of maturation from G0 to G4, common monocyte progenitors (CMoP I), two subsets of monocytes I and II, and minor fractions of HSC and dendritic cell precursors (PreDC II) (Fig. 5a). CD11b+ myeloid cell proportions were similar in the BM of WT and IL-1rn-KO mice (Fig. 5a), with only minor expansions of G3 neutrophils and CMoP, and reductions in G4 neutrophils and type II monocytes (Fig. 5a). Ingenuity Pathway Analysis (IPA) of "Diseases and Functions" of the different CD11b+ myeloid clusters from IL-1rn-KO

**Fig. 2 | Deletion of IL-1rn triggers IL-1β-induced myelopoiesis in the absence of immunogenic stimulus. a** Representative hematoxylin and eosin stainings of femoral bone marrow (BM) from C57BL/6J wild-type (WT, $n = 8$) and IL-1rn knockout (IL-1rn-KO, $n = 10$) mice; scale bar, 100 μm, and total BM (TBM) nucleated cells (WT, $n = 16$; IL-1rn-KO, $n = 23$). **b** TBM number of Lin⁻Sca-1⁺c-Kit⁺ (LSK) subsets: LSK CD34⁻Flt3⁻CD48⁻CD150⁺, hematopoietic stem cells (HSC); LSK CD34⁺Flt3⁻CD48⁻CD150⁺, multipotent progenitors 1 (MPP1); LSK CD34⁺Flt3⁻CD48⁺CD150⁺, MPP2; LSK CD34⁺Flt3⁻CD48⁺CD150⁻, MPP3; LSK CD34⁺Flt3⁺CD48⁺CD150⁻, MPP4; LSK CD34⁺Flt3⁺CD48⁻CD150⁻, MPP5 ($n = 9$ per group). **c** BM colony-forming unit cells (CFU-C) (WT, $n = 6$; IL-1rn-KO, $n = 10$). **d** TBM number of CD11b⁺ myeloid cells and B220⁺ lymphocytes (WT, $n = 6$; IL-1rn-KO, $n = 7$). **e** IL-1β content in BM extracellular fluid (WT, $n = 9$; IL-1rn-KO, $n = 12$). **f** Representative IL-1 receptor (IL-1r1) stainings (red) of BM; nuclei counterstained with DAPI (blue); scale bar, 20 μm, and frequency of TBM IL-1r1-expressing cells ($n = 7$ per group). **g, h** IL-1rn-KO mice treated with vehicle ($n = 5$) or IL-1RN ($n = 4$). **g** Number of CD11b⁺ myeloid cells and neutrophils in peripheral blood (PB). **h** Frequency of CD11b⁺ myeloid cells and B220⁺ lymphocytes in TBM. **i, j** IL-1rn-KO mice treated with isotype ($n = 5$) or mouse IL-1β monoclonal antibody (IL-1β mAb) ($n = 4$). **i** Frequency of CD11b⁺ myeloid cells and CD11b⁺Gr-1ʰⁱf4/80⁻ granulocytes in PB. **j** TBM nucleated cells, CD11b⁺ myeloid cells and B220⁺ lymphocytes. **k–m** qRT-PCR mRNA expression in **k** LSK subsets: LSK CD34⁻Flt3⁻, long-term HSC (LT-HSC); LSK CD34⁺Flt3⁻, short-term HSC (ST-HSC); LSK CD34⁺Flt3⁺ MPP of *Il1b* ($n = 3$ per group), *Il1r1* (LT-HSC, MPP, $n = 7$; ST-HSC, $n = 6$, per group) and *Il1rap* ($n = 4$ per group). **l** CD11b⁺Gr-1ʰⁱf4/80⁻ granulocytes of *Il1b* ($n = 8$ per group), *Il1r1* (WT, $n = 12$; IL-1rn-KO, $n = 11$) and *Il1rap* ($n = 4$ per group) and in CD11b⁺Gr-1⁺f4/80⁺ monocytes of *Il1b* ($n = 11$ per group), *Il1r1* ($n = 15$ per group) and *Il1rap* ($n = 4$ per group). **m** CD11b⁺Gr-1ʰⁱf4/80⁻ granulocytes ($n = 8$ per group), CD11b⁺Gr-1⁺f4/80⁺ monocytes (WT, $n = 7$; IL-1rn-KO, $n = 8$) and LSK cells ($n = 4$ per group) of *Il1rn*. Data are biologically independent animals, and means ± SEM for bar plots or medians for violin plots. Statistical analyses were performed with two-tailed Student's *t*-test (**a–d, f–h, i** left, **j–m**) or two-tailed Mann–Whitney *U* test (**e, i** right). *p* values ≤ 0.05 are reported. Source data are provided as a Source Data file.

versus WT mice revealed significant enrichment in pathways related to inflammation, proliferation, migration, endocytosis and survival (Supplementary Fig. S6A). Inflammation was a major pathway activated in CD11b⁺ myeloid clusters from IL-1rn-KO versus WT mice, with the exceptions of G1 neutrophils that showed no changes and type I monocytes that showed a reduced inflammatory signature (Supplementary Fig. S6A). Among the differentially expressed genes responsible for these enriched signatures, we found upregulation of several major pro-inflammatory genes (*Hif1a, Csf2rb, Myd88, Cxcr2, Lmo4*)[40–44] as well as downregulation of anti-inflammatory genes (*Nfkbia, Cebpb*)[45,46]. A few genes that control inflammation (*Lyn*)[47] were upregulated in CD11b⁺ myeloid clusters from IL-1rn-KO versus WT mice (Supplementary Fig. S6B).

Neutrophils and particularly the most mature G4 cells were the most abundant CD11b⁺ cell subset expressing *Il1b* and *Il1rn*, and expressed the highest levels of both cytokines, followed by monocytes (Fig. 5a, b). Cytokine-expressing neutrophils and monocytes express mainly *Il1b* or *Il1rn* as evidenced by both scRNA-Seq (Fig. 5b, c) and RNA fluorescent in situ hybridization (FISH) (Fig. 5d; Supplementary Fig. S7A), with small fractions of cells particularly within G4 neutrophils, expressing both cytokines (Fig. 5c).

Similarly, WT and IL-1rn-KO scRNA-Seq data from FACS-sorted LSK cells were integrated using the CCA algorithm of Seurat, which was followed by unbiased cell clustering (Fig. 6a). WT cells from each cluster were used for annotation. Gene modules imported from E-MTAB-9208 dataset were scored over LSK clusters[29], and cluster assignment to each cell type was obtained using the maximum enrichment score ±0.2 as previously described[29] and subsequent manual curation. This resulted in 6 annotated clusters, namely HSC, HSC-MPP1, MPP2, MPP3, MPP4 and MPP5 (Fig. 6a). The proportions of BM LSK cells after clustering of the scRNA-Seq data showed overall similar changes to the immunophenotypic analysis of BM LSK in IL-1rn-KO versus WT mice (Fig. 2b), with qualitative reduction in HSC and MPP5, and expansion of MPP2 (Fig. 6a). Unlike FACS quantification, the scRNA-Seq data showed qualitative expansion of the MPP3 transcriptional state and reduction in lymphoid-biased MPP4 in the BM LSK compartment of IL-1rn-KO mice versus WT (Fig. 6a). IPA of "Diseases and Functions" of the different LSK clusters from IL-1rn-KO versus WT mice revealed significant enrichment in pathways related to tumorigenesis and cell death which were activated in HSC, HSC-MPP1, MPP4 and MPP5 (Supplementary Fig. S8A). MPP2 and MPP3 behaved differently and showed significant activation of pathways related to survival (Supplementary Fig. S8A). Further, tumorigenic pathways were activated in MPP2 but slightly reduced in MPP3 (Supplementary Fig. S8A). Looking into the differentially expressed genes responsible for these enriched signatures, we found upregulation of important genes

known to be involved in AML (*Nmt1, Ifitm3, Crip1, Cd52*)[48–51] and downregulation of genes whose reduced expression is a common feature of acute myeloid leukemogenesis (*Junb*)[52]. We also found upregulation of genes with antiproliferative role (*Ifitm1*)[53] and downregulation of genes whose loss promote cell exhaustion (*Hlf*)[54] in LSK clusters from IL-1rn-KO versus WT mice (Supplementary Fig. S8B).

Expression of *Il1b* and *Il1rn* was not detectable by scRNA-Seq in LSK cells (Supplementary Fig. S8C), but RNA-FISH showed that LT-HSC expressed the highest levels of *Il1b* among LSK subsets, followed by ST-HSC and MPP (Fig. 6b; Supplementary Fig. S9A). Some of the *Il1b*-expressing LT-HSC also had detectable levels of *Il1rn* expression (Fig. 6b; Supplementary Fig. S9A).

The same procedure was used for the annotation of the scRNA-Seq data from FACS-sorted CD63⁺ stromal cells with gene signatures obtained from GSE128423 dataset[35]. We identified 5 clusters, namely arteriolar cells, fibroblasts, MSC, pericytes and sinusoidal cells (Fig. 7a). Arteriolar and sinusoidal cells were reduced whereas pericytes were increased within the CD63⁺ stromal compartment from IL-1rn-KO mice versus WT (Fig. 7a). IPA of "Diseases and Functions" of the CD63⁺ stromal cluster fibroblasts from IL-1rn-KO versus WT mice revealed significant enrichment in pathways related to differentiation into adipocytes, chondrocytes and osteocytes that were activated whereas gene sets coordinating functions like proliferation and inflammation were reduced, as opposed to MSC (Supplementary Fig. S10A). Among the differentially expressed genes responsible for the enriched differentiation signatures, we found important genes involved in adipocyte (*Nr1d1, Zbtb16, Nfia*)[55–57], chondrocyte (*Hspg2, Bmp4, Col3a1*)[58–60] and osteocyte lineage differentiation (*Fbn1, Twist1, Spp1*)[61–63] (Supplementary Fig. S10B). All arteriolar, sinusoidal and pericyte clusters showed significant enrichment and activation in pathways associated with angiogenesis (Supplementary Fig. S10C). In addition, the sinusoidal cluster displayed activation of pathways related to inflammation, atherosclerosis and recruitment of leukocytes (Supplementary Fig. S10C). We checked the differentially expressed genes responsible for these enriched signatures and found upregulation of major pro-inflammatory genes in the CD63⁺ sinusoidal cluster from IL-1rn-KO mice versus WT (*Il1b, Il6, Nlrp3, Csf1, Ccl4, Ccl6, Ccl9, Osm*) (Supplementary Fig. S10D).

We found cells expressing *Il1b, Il1rn* or both within the CD63⁺ stromal cell compartment both by scRNA-Seq (Fig. 7b-c) and RNA-FISH (Fig. 7d; Supplementary Fig. S11A). scRNA-Seq was able to resolve that *Il1b/Il1rn* are mainly produced by MSC and arteriolar cells within the CD63⁺ stromal cell compartment, and most cytokine-expressing MSC express *Il1rn* only whereas arteriolar cells express mainly *Il1b* or both, with a small fraction expressing *Il1rn* only (Fig. 7a−c). A small fraction of sinusoidal cells also expresses *Il1b* only (Fig. 7a−c; Supplementary Fig. S11A).

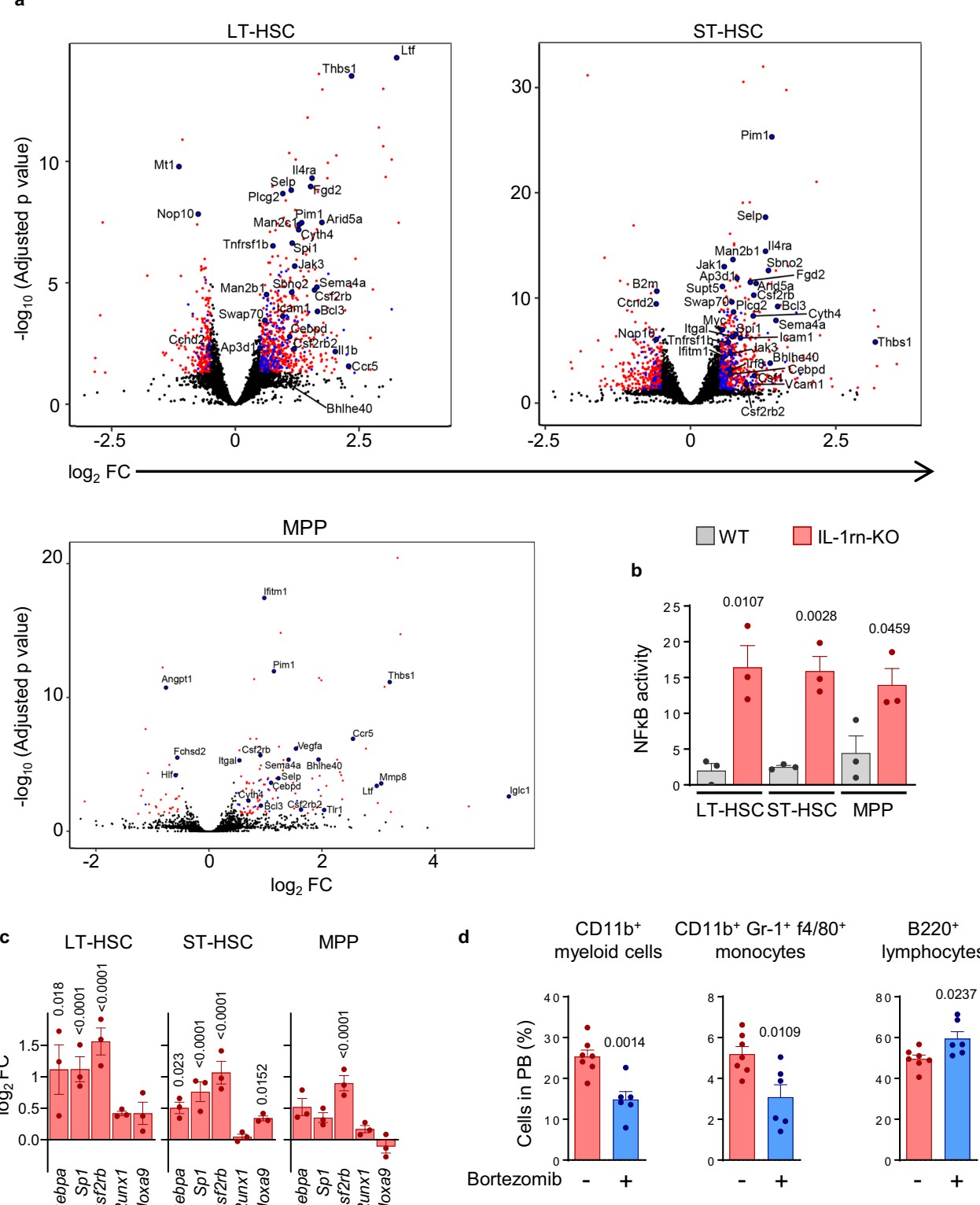

**Fig. 3 | IL-1β-induced myelopoiesis after IL-1rn deletion is mediated by NFκB activation. a**–**c** RNA sequencing in LSK CD34⁻Flt3⁻, long-term HSC (LT-HSC); LSK CD34⁺Flt3⁻, short-term HSC (ST-HSC) and LSK CD34⁺Flt3⁺ MPP from IL-1rn knock-out (IL-1rn-KO) versus C57BL/6J wild-type (WT) mice ($n = 3$ per group). **a** Volcano plots show differentially expressed genes (red dots) and differentially expressed NFκB target genes (blue dots) identified from Synapse ID syn4956655, https://bioinfo.lifl.fr/NF-KB and https://www.bu.edu/nf-kb/gene-resources/target-genes (Supplementary Data 1). The plots depict the top 15 differentially expressed NFκB target genes, NFκB target genes of interest, and genes from those lists shared by another subset. Adjusted $p < 0.05$; fold change (FC), $-0.5 > \log_2 FC > 0.5$. **b** NFκB transcription factor activity calculated based on NFκB target gene expression levels, identified from Synapse ID syn4956655 (Supplementary Data 1). **c** Expression of myeloid differentiation genes. **d** Frequency of CD11b⁺ myeloid cells, CD11b⁺Gr-1⁺f4/80⁺ monocytes and B220⁺ lymphocytes in peripheral blood (PB) from IL-1rn-KO mice treated with vehicle ($n = 7$) or bortezomib ($n = 6$). Data are biologically independent animals, and means ± SEM., except (**a**). Statistical analyses were performed with two-tailed Wald test with Benjamini-Hochberg correction for multiple comparisons (**a, c**) or two-tailed Student's *t*-test (**b, d**). Adjusted $p < 0.05$ (**a, c**) or $p$ values ≤ 0.05 (**b, d**) are reported. Source data are provided as a Source Data file.

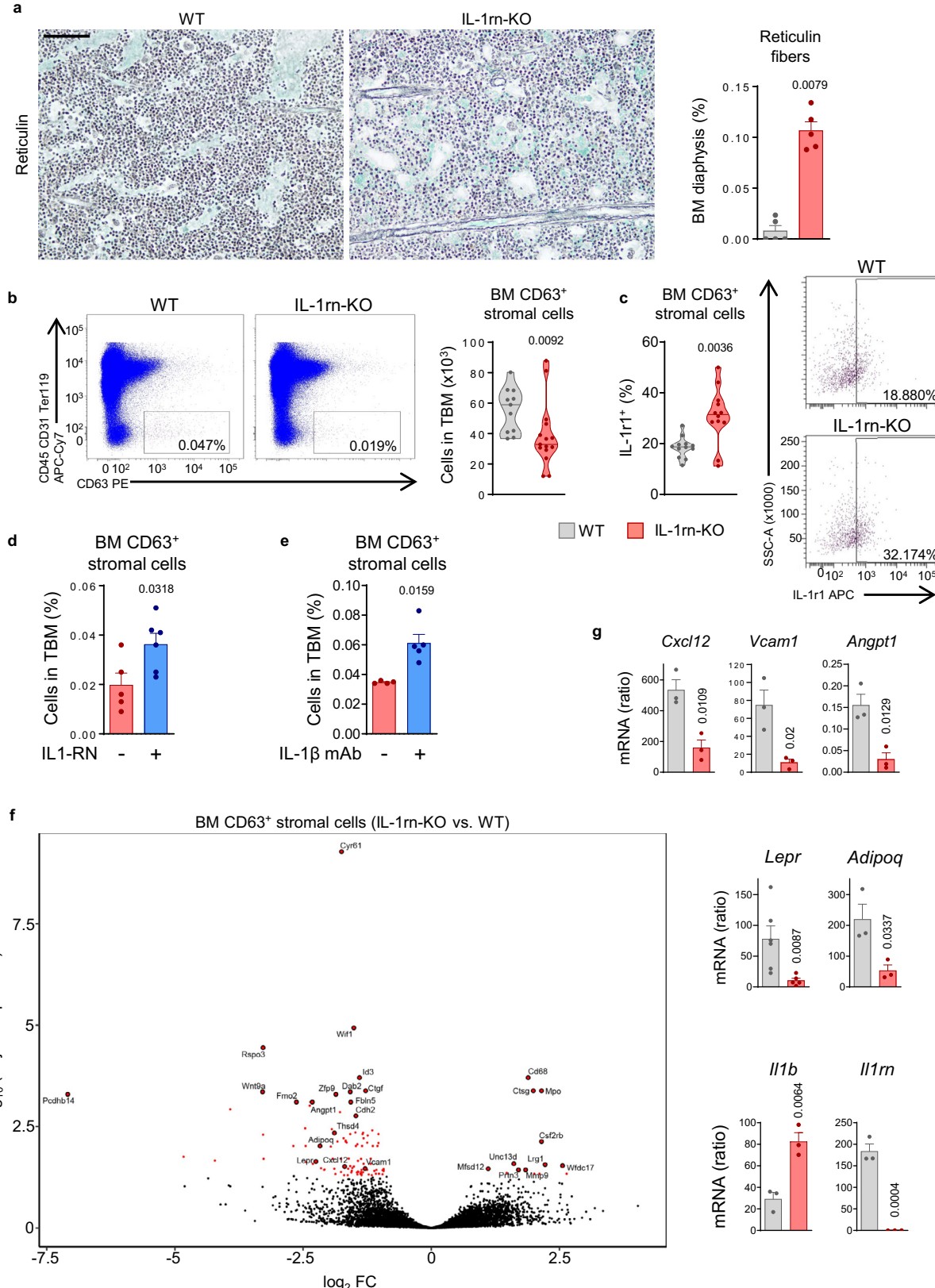

Together, these data highlight that IL-1rn buffers mostly paracrine IL-1β feed-forward loops through its exclusive expression from subsets of CD63⁺ MSC, neutrophils and monocytes, and a small fraction of CD63⁺ arteriolar cells in the BM of WT healthy mice under steady-state conditions. Autocrine responses are rare and were observed mainly in subsets of LT-HSC and CD63⁺ arteriolar cells, and

a small proportion of neutrophils mainly within G4, with expression of both cytokines simultaneously.

**Low IL-1rn is a hallmark of pre-leukemic myelopoiesis in vivo**
The reduction of IL-1RN in AML patients and the effect of IL-1rn loss in HSPC suggest its role in hematological diseases. To address this

**Fig. 4 | Deletion of IL-1rn causes IL-1β-induced damage to the bone marrow stroma in the absence of immunogenic stimulus. a** Representative Gordon and Sweet's stainings of femoral bone marrow (BM) from C57BL/6J wild-type (WT, $n = 5$) and IL-1rn knockout (IL-1rn-KO, $n = 5$) mice; scale bar, 100 µm, and quantification of reticulin fibers (arrows) in BM diaphysis. **b** Representative fluorescence-activated cell sorting (FACS) analysis (cells in TBM, %) and total BM (TBM) number of immunophenotypically defined CD45⁻CD31⁻Ter119⁻CD63⁺ stromal cells (WT, $n = 11$; IL-1rn-KO, $n = 15$). **c** Frequency of TBM CD45⁻CD31⁻Ter119⁻CD63⁺ stromal cells expressing IL-1 receptor (IL-1r1, $n = 12$ per group) and representative FACS analysis (cells in BM CD63⁺ stromal cells, %). **d** Frequency of TBM CD45⁻CD31⁻Ter119⁻CD63⁺ stromal cells in IL-1rn-KO mice treated with vehicle ($n = 5$) or IL-1RN ($n = 6$). **e** Frequency of TBM CD45⁻CD31⁻Ter119⁻CD63⁺ stromal cells in IL-1rn-KO mice treated with isotype ($n = 4$) or mouse IL-1β monoclonal antibody (IL-1β mAb) ($n = 5$). **f** RNA sequencing in BM CD45⁻CD31⁻Ter119⁻CD63⁺ stromal cells ($n = 3$ per group). Volcano plot shows differentially expressed genes (red dots) and depicts genes of interest. Adjusted $p < 0.01$; fold change (FC), $-0.5 > \log_2 FC > 0.5$. **g** qRT-PCR mRNA expression of selected RNA-Seq hits (*Cxcl12, Vcam1, Angpt1, Lepr, Adipoq*), *Il1b* and *Il1rn* in independent biological samples ($n = 3$ per group except *Lepr*, WT, $n = 5$; IL-1rn-KO, $n = 6$). Data are biologically independent animals, and means ± SEM for bar plots or medians for violin plots, except (**f**). Statistical analyses were performed with two-tailed Mann–Whitney *U* test (**a**–**c**, **e**, **g** *Lepr*), two-tailed Student's *t*-test (**d**, **g** except *Lepr*) or two-tailed Wald test with Benjamini–Hochberg correction for multiple comparisons (**f**). $p \leq 0.05$ (**a**–**e**, **g**) or adjusted $p$ values < 0.05 (**f**) are reported. Source data are provided as a Source Data file.

possibility, we used the *Mx1-Cre NRAS^G12D^* mouse model of aberrant pre-leukemic myelopoiesis[64,65], from now on referred to as NRAS-G12D⁺. As previously described, pre-leukemic NRAS-G12D⁺ HSPC outcompete WT cells (Supplementary Fig. S12A)[65]. Primary NRAS-G12D⁺ mice display a mild hematopoietic phenotype, characterized as chronic myelomonocytic leukemia[66,67] with increased circulating white blood cells, particularly myeloid cells. In the BM, numbers of c-Kit⁺ progenitors and LSK HSPC were higher in NRAS-G12D⁺ mice, with abnormal relative amounts of all LSK subsets (Supplementary Fig. S12B–E)[64]. Detailed analysis revealed reduction of HSC, MPP5 and MPP1, and an increase of MPP4 in the BM of NRAS-G12D⁺ mice (Supplementary Fig. S12F). More committed CLP and GMP progenitors were expanded in the BM of NRAS-G12D⁺ versus control mice (Supplementary Fig. S12G).

RAS transduces IL-1β signaling through MyD88, and oncogenic RAS results in IL-1β amplification through persistent activation of the autocrine feedback loop in other types of cancer[68]. IL-1β levels in NRAS-G12D⁺ BM were increased, while levels of IL-1rn were reduced in the BM of diseased mice (Fig. 8a, b). At this stage, BM hypercellularity was prominent, with expansion of cell subsets previously described to produce IL-1-β including neutrophils, monocytes and LSK cells (Fig. 8c). Expression of *Il1b* was not induced in granulocytes or monocytes from the BM of NRAS-G12D⁺ mice compared to healthy controls, and these hematopoietic cells did not show reduced expression of *Il1rn* either (Fig. 8d). In contrast, all HSPC subsets expressed higher levels of *Il1b* measured by qRT-PCR, and *Il1rn* was reduced in NRAS-G12D⁺ versus control MPP only (Fig. 8e). *Il1rap* gene expression was unchanged in any of the hematopoietic cell populations studied (Fig. 8d, e), but the fraction of cells expressing membrane IL-1RAP was higher in NRAS-G12D⁺ versus control LT-HSC, MPP and monocytes (Fig. 8f; Supplementary Fig. S12H). To better understand the specific contribution of HSPC to the IL-1β-induced inflammatory environment of the BM in NRAS-G12D⁺ mice, gene expression profiling was performed by RNA-Seq in LT-HSC, ST-HSC and MPP. The transcriptional programs of both LT-HSC and MPP were influenced by expression of NRAS-G12D, with 894 and 651 differentially expressed genes, respectively. The effect of NRAS-G12D expression on ST-HSC was smaller, with 340 genes differentially expressed (Supplementary Fig. S12I). PCA showed coherent clustering of replicates and revealed that LT-HSC are most influenced by NRAS-G12D expression (Supplementary Fig. S12J). All NRAS-G12D⁺ HSPC subsets exhibited increased NFκB transcription factor calculated activity (Fig. 8g). The myeloid bias of progenitors from NRAS-G12D⁺ mice could be improved in vivo by treatment with bortezomib (Fig. 8h). Taken together, the biased myelopoiesis in NRAS-G12D⁺ mice was at least partially dependent on NFκB activation in HSPC, and HSPC contributed to loss of repression of IL-1β pathway through increased expression of *Il1b* and membrane IL-1RAP, and reduced expression of *Il1rn* in the case of MPP.

We then studied the stromal compartment and found reduction in BM CD63⁺ stromal cell numbers (Fig. 9a) together with increased apoptotic rates in NRAS-G12D⁺ mice (Fig. 9b). BM CD63⁺ stromal cells were sorted and qRT-PCR analyses revealed no changes for *Il1b* and *Il1rap* but reduced expression of *Il1rn* in BM CD63⁺ stromal cells from diseased mice (Fig. 9c). Reduced numbers of Nestin⁺ stromal cells and activation of apoptosis at the transcriptomic level was confirmed using *Nes-gfp* mice that had previously received NRAS-G12D⁺ BM, compared to recipients of control BM (Supplementary Fig. S13A, B). Hence, damage to the microenvironment is present in the NRAS-G12D⁺ mouse model of pre-leukemic myelopoiesis, and MSC contribute to the low IL-1rn in BM.

To minimize potential confounding effects derived from *Cre* expression in osteolineage cells[69] and from transient activation of IFNα signaling through polyI:polyC injection[70], we used a conditional model that constitutively expresses *NRAS^G12D^* in the hematopoietic system under the control of *Vav-Cre*[67]. This mouse model reproduced the main hematopoietic abnormalities of *Mx1-Cre NRAS^G12D^* mice including increased circulating white blood cells particularly monocytes, hypercellularity in the BM, and expansion of immunophenotypically defined LSK HSPC, particularly MPP4, and CLP (Supplementary Fig. S14A–F). Analysis of publicly available gene expression profiling by RNA-Seq of Lin⁻c-Kit⁺Sca-1⁻ (LK) progenitors (PRJNA774277)[67] showed that NFκB transcription factor calculated activity was higher in *Vav-Cre NRAS^G12D^* than in control mice (Supplementary Fig. S14G).

### Deletion of IL-1rn from the hematopoietic or stromal compartments promotes pre-leukemic myelopoiesis, whereas exogenous IL-1rn protects against it

To address the role of IL-1rn in hematopoietic cells under pre-leukemic myelopoiesis, we generated mixed-chimera systems using WT and IL-1rn-KO mice; WT and NRAS-G12D⁺ mice; or IL-1rn-KO and NRAS-G12D⁺ mice, as donors of BM to WT recipients (Fig. 10a). Deletion of IL-1rn in hematopoietic cells exerted a synergistic effect with presence of *NRAS^G12D^* in neighboring hematopoietic cells to promote myeloid output (Fig. 10b). B cells showed no changes in presence of *NRAS^G12D^* irrespective of WT or IL-1rn deficient competitor cells (Fig. 10c).

To investigate the role of IL-1rn deficiency in the stroma under pre-leukemic myelopoiesis, we then performed transplants using WT and IL-1rn-KO mice as recipients of BM from induced disease-free *NRAS^G12D^* control and *Mx1-Cre NRAS^G12D^* mice (Fig. 10d). Numbers of circulating myeloid cells, particularly monocytes, were higher in IL-1rn-KO mice transplanted with diseased NRAS-G12D⁺ cells as compared to WT recipients (Fig. 10e). B cells were expanded only in IL-1rn-KO mice transplanted with diseased NRAS-G12D⁺ cells (Fig. 10f). These results suggest that loss of IL-1β repression through IL-1rn deletion within either the BM hematopoietic or stromal compartment may play a role in pre-leukemic disease. The data further demonstrate that the reduced numbers of BM MSC and their *Il1rn* expression in induced *Mx1-Cre NRAS^G12D^* mice (Fig. 9a, c) contributes to NRAS-G12D⁺ disease.

Conversely, short-term in vivo treatment of induced *Mx1-Cre NRAS^G12D^* mice with IL-1RN (Fig. 10g) ameliorated early signs of

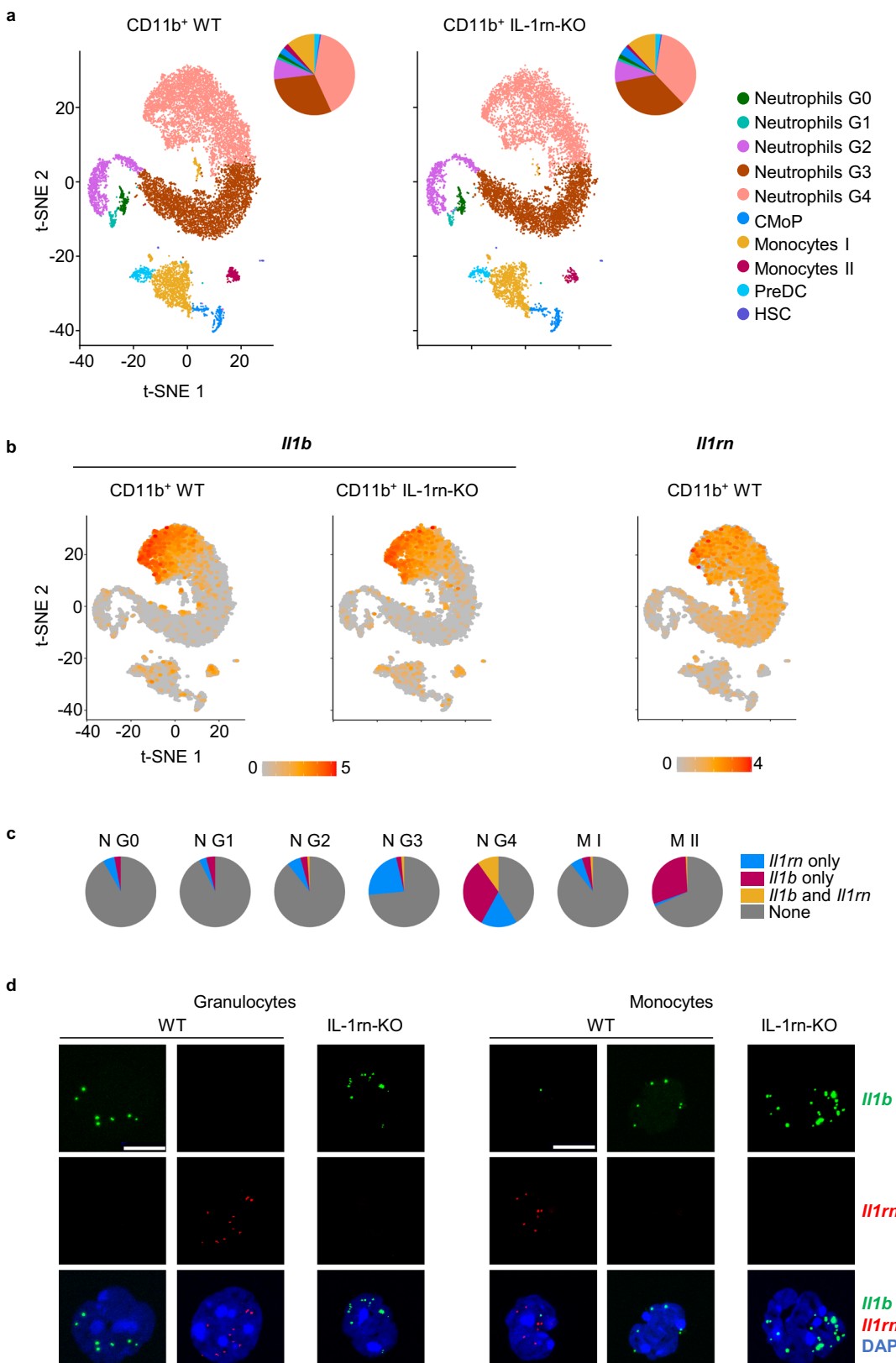

abnormal myelopoiesis by means of reduced circulating myeloid cells, particularly monocytes (Fig. 10h), with no detected changes in B cells (Fig. 10i). Thus, the low IL-1rn conditions contribute to the biased myelopoiesis in NRAS-G12D⁺ mice. These data uncover the protective effect of the endogenous IL-1 repressor IL-1rn against pre-leukemic myelopoiesis.

## Discussion

Despite the emerging role of IL-1β signaling in the pathogenesis of hematological diseases[1,6–12,15,16], little is known about the potential contribution of its endogenous counteracting cytokine IL-1RN to healthy and/or malignant hematopoiesis. Here, we find that low expression of *IL1RN* in purified blasts correlated with negative

**Fig. 5 | Identification of the sources of IL-1β and IL-1rn in the bone marrow at the single-cell level within the myeloid compartment. a–c** Single-cell RNA sequencing in bone marrow (BM) CD11b⁺ myeloid cells from C57BL/6J wild-type (WT, *n* = 1 sample from 2 biologically independent animals) and IL-1rn knockout (IL-1rn-KO, *n* = 1 sample from 2 biologically independent animals) mice. **a** t-SNE plots visualizing colored cluster assignment annotated with the Immgen database followed by scoring of gene signatures imported from GSE137539 dataset over neutrophil-like subsets and those imported from GSE131834 dataset over monocyte-like clusters. Cluster assignment to each cell type was obtained using the maximum enrichment score ±0.2 and subsequent manual curation. Pie charts with cluster distribution are shown to the right of the t-SNE plots. CMoP, common monocyte progenitors I; PreDC, dendritic cell precursors II; HSC, hematopoietic stem cells. **b** t-SNE plots visualizing expression levels of *Il1b* and *Il1rn*. Color bar, log1p(TP10K). **c** Pie charts with proportion of cells expressing *Il1rn* and/or *Il1b* or none, in neutrophil (N) and monocyte (M) clusters from the BM of WT mice. **d** RNA fluorescent in situ hybridization in BM CD11b⁺Gr-1ʰⁱf4/80⁻ granulocytes and CD11b⁺Gr-1⁺f4/80⁺ monocytes from C57BL/6J WT and IL-1rn-KO mice (*n* = 4 biologically independent animals per group). Representative images of cells expressing *Il1b* (green) and/or *Il1rn* (red). Nuclei counterstained with DAPI. Scale bar, 10 μm. Source data are provided as a Source Data file.

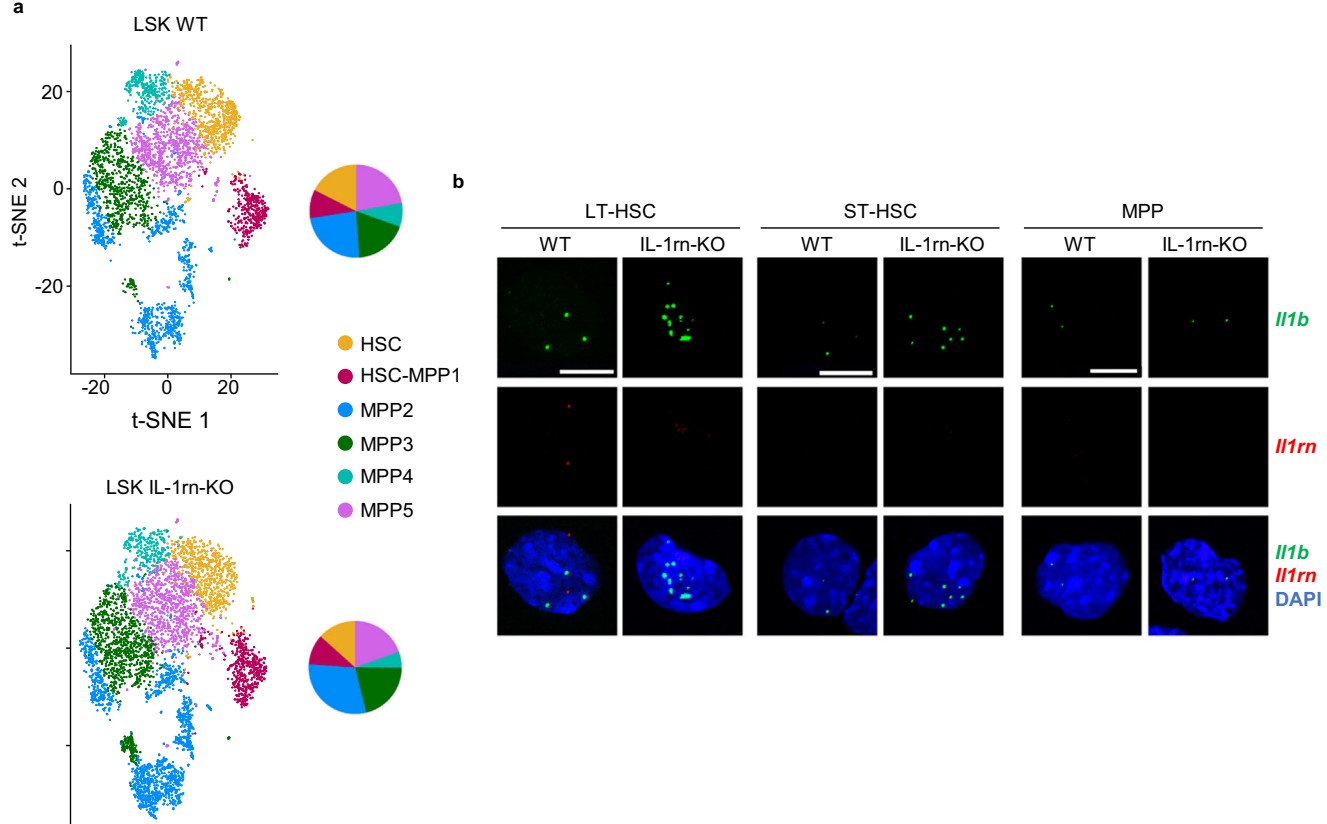

**Fig. 6 | Identification of the sources of IL-1β and IL-1rn in the bone marrow at the single-cell level within the hematopoietic stem and progenitor cell compartment. a** Single-cell RNA sequencing in bone marrow (BM) Lin⁻Sca-1⁺c-Kit⁺ (LSK) cells from C57BL/6J wild-type (WT, *n* = 1 sample from 2 biologically independent animals) and IL-1rn knockout (IL-1rn-KO, *n* = 1 sample from 2 biologically independent animals) mice. t-SNE plots visualizing colored cluster assignment annotated with E-MTAB-9208 dataset using the maximum enrichment score ±0.2 and subsequent manual curation. Pie charts with cluster distribution are shown to the right of the t-SNE plots. HSC, hematopoietic stem cells; MPP, multipotent progenitors. **b** RNA fluorescent in situ hybridization in BM LSK CD34⁻Flt3⁻, long-term HSC (LT-HSC); LSK CD34⁺Flt3⁻, short-term HSC (ST-HSC) and LSK CD34⁺Flt3⁺ MPP, from C57BL/6J WT and IL-1rn-KO mice (*n* = 3 biologically independent animals per group). Representative images of cells expressing *Il1b* (green) and/or *Il1rn* (red). Nuclei counterstained with DAPI. Scale bar, 10 μm. Source data are provided as a Source Data file.

prognosis in a publicly available big cohort of 381 global AML patients, and from those in 164 M4-M5 AML patients, at diagnosis time[23–25] as well as in matched-pair diagnosis-relapsed AML patients[26]. We provide translational data in primary CD34⁺ progenitors from AML patients showing IL-1RN deregulation in newly diagnosed AML patients, particularly those with lower differentiation profiles, making the study of the prognostic value of IL-1RN not possible in this group. IL-1RN deregulation was confirmed in a publicly available cohort of AML patients that used purified AML blasts[23–25]. These data are of particular interest considering AML heterogeneity, as it indicates that low IL-1RN is a hallmark of AML that may contribute to disease aggravation and poor outcome. Future studies aimed at uncovering the underlying molecular mechanisms for IL-1RN reduction in AML will be relevant, and should include methylation in CpG

sites as it was previously involved in relapsed versus diagnosis paired AML samples[26].

NFκB may be activated by multiple signals but in patients, unbalanced IL-1RN in AML CD34⁺ progenitors associated with higher numbers of cells activated through NFκB as well as higher nuclear translocation of the p50/p65 NFκB heterodimer. CD34⁺ cells are a mixture of LT-HSC, ST-HSC and progenitors, so these changes may reflect the differential cell composition and/or selective changes per cell subset, between health and AML. These data are indicative of the enrichment of this pathway in AML CD34⁺ cells and are consistent with previous observations of NFκB activation in human CD34⁺CD38⁻ leukemia stem cells[71]. Chronic IL-1β treatment of NSG-SGM3 mice transplanted with CD34⁺ progenitors isolated from the BM of AML patients promoted AML cell expansion, in agreement with previous

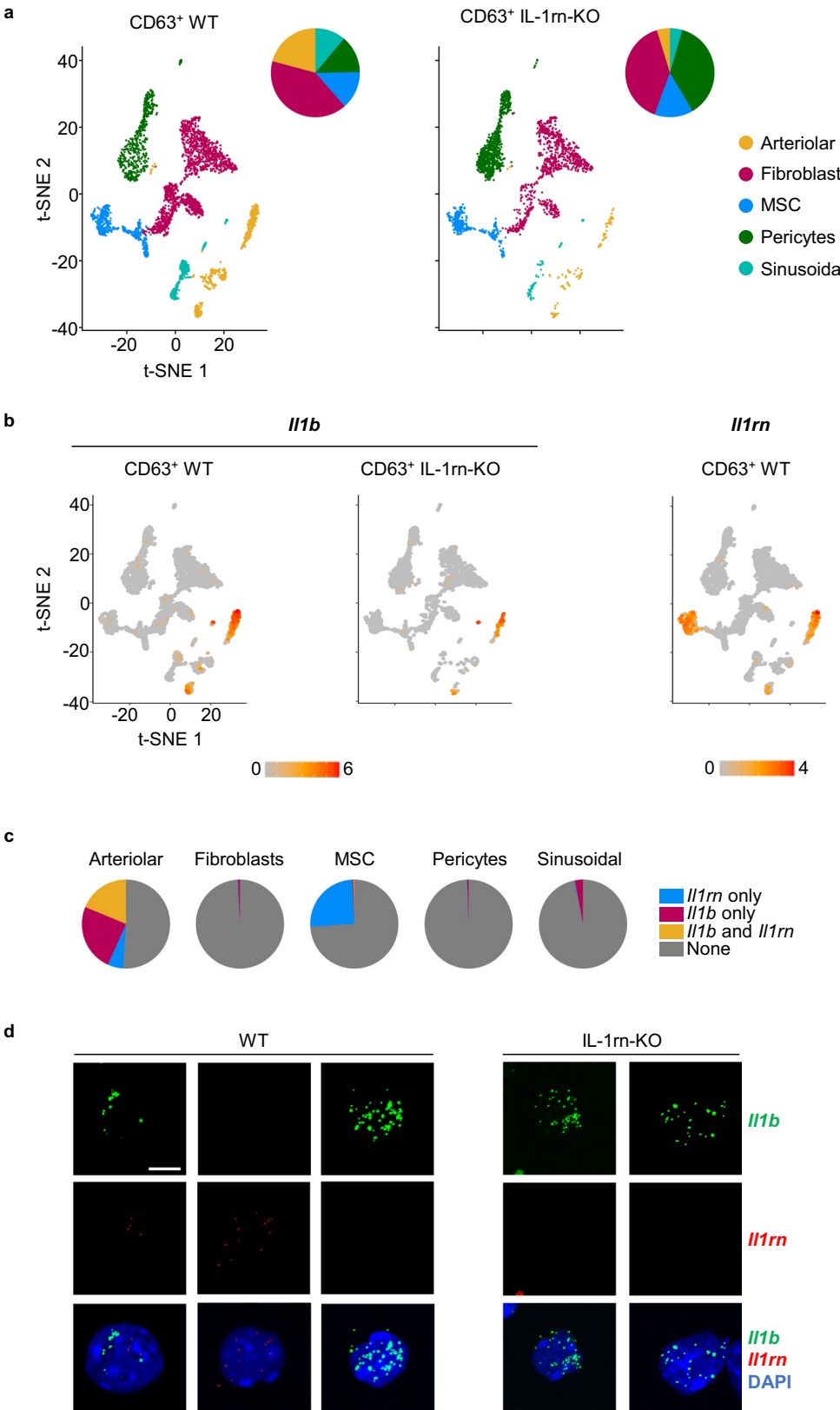

**Fig. 7 | Identification of the sources of IL-1β and IL-1rn in the bone marrow at the single-cell level within the CD63+ stromal cell compartment. a, b** Single-cell RNA sequencing in bone marrow (BM) BM CD45⁻CD31⁻Ter119⁻CD63⁺ stromal cells from C57BL/6J wild-type (WT, $n = 1$ sample from 2 biologically independent animals) and IL-1rn knockout (IL-1rn-KO, $n = 1$ sample from 2 biologically independent animals) mice. **a** t-SNE plots visualizing colored cluster assignment annotated with GSE128423 dataset using the maximum enrichment score ±0.2 and subsequent manual curation. Pie charts with cluster distribution are shown to the right of the t-SNE plot. MSC, mesenchymal stromal cells. **b** t-SNE plots visualizing expression levels of *Il1b* and *Il1rn*. Color bar, log1p(TP10K). **c** Pie charts with proportion of cells expressing *Il1rn* and/or *Il1b* or none in CD63⁺ stromal cell clusters from the BM of WT mice. **d** RNA fluorescent in situ hybridization in BM CD45⁻CD31⁻Ter119⁻CD63⁺ stromal cells from C57BL/6J WT and IL-1rn-KO mice ($n = 4$ biologically independent animals per group). Representative images of cells expressing *Il1b* (green) and/or *Il1rn* (red). Nuclei counterstained with DAPI. Scale bar, 10 μm. Source data are provided as a Source Data file.

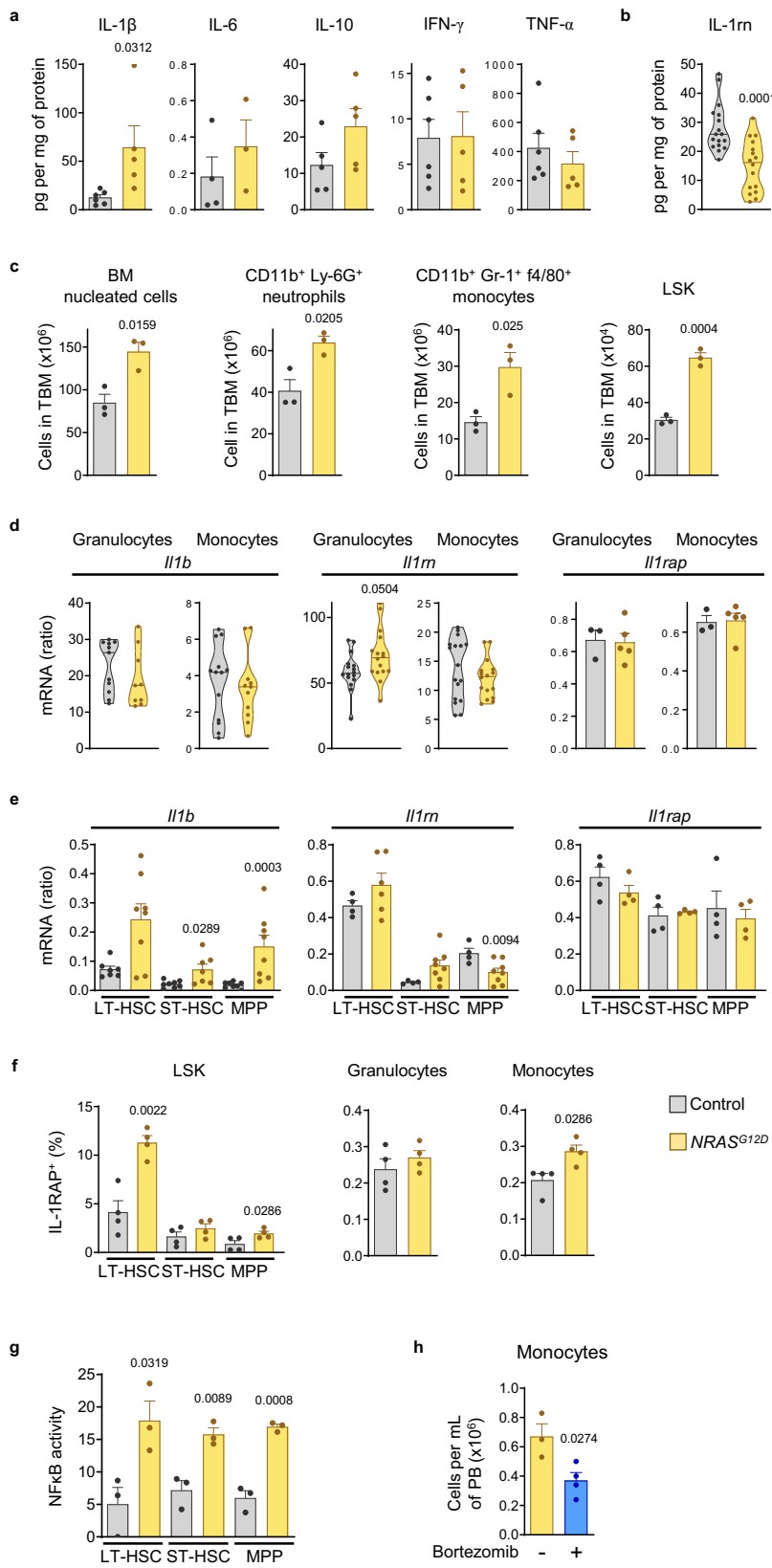

results[6]. Conversely, boosting IL-1RN improved signs of disease in the *NRAS^G12D*-driven myeloid pre-leukemia mouse model and in mouse xenografts from AML patients. IL-1RN boost or IL-1β blockade through treatment with canakinumab, currently under Phase II trial for the treatment of low and intermediate risk MDS and CML (NTC04239157), reduced to similar extent the expansion of human myeloid cells, in NSG-SGM3 mice. Pharmacologically, canakinumab provides advantage over anakinra considering their half-lives of 26 days and 4-6 h, respectively[1]. However, the CANTOS trial (Canakinumab Anti-Inflammatory Thrombosis Outcome Study) raised safety concerns due to higher incidence of deaths due to fatal infection in the treated versus the placebo group[72]. Anakinra

**Fig. 8 | Low IL-1rn is present in a mouse model of pre-leukemic myelopoiesis.**
**a** Cytokine levels in bone marrow (BM) extracellular fluid (BMEF) from control
(*Mx1-Cre⁻ NRAS^G12D*) and NRAS-G12D⁺ (*Mx1-Cre⁺ NRAS^G12D*) mice. IL-1β (control, n = 6;
NRAS-G12D⁺, n = 5), IL-6 (control, n = 4; NRAS-G12D⁺, n = 3), IL-10 (n = 5 per group),
IFN-γ (control, n = 6; NRAS-G12D⁺, n = 5) and TNF-α (control, n = 6; NRAS-G12D⁺,
n = 5) levels are shown. **b** IL-1rn levels in BMEF (control, n = 17; NRAS-G12D⁺, n = 16).
**c** Numbers of cells in total BM (TBM), and TBM number of CD11b⁺Ly6G⁺ neu-
trophils, CD11b⁺Gr-1⁺f4/80⁺ monocytes and Lin⁻Sca-1⁺c-Kit⁺ (LSK) cells (n = 3 per
group). **d, e** qRT-PCR mRNA expression in (**d**) CD11b⁺Gr-1^hif4/80⁻ granulocytes of
*Il1b* (control, n = 11; NRAS-G12D⁺, n = 9), *Il1rn* (control, n = 16; NRAS-G12D⁺, n = 9)
and *Il1rap* (control, n = 3; NRAS-G12D⁺, n = 5) and in CD11b⁺Gr-1⁺f4/80⁺ monocytes
of *Il1b* (control, n = 13; NRAS-G12D⁺, n = 11), *Il1rn* (control, n = 17; NRAS-G12D⁺, n = 16)
and *Il1rap* (control, n = 3; NRAS-G12D⁺, n = 5). **e** LSK subsets: LSK CD34⁻Flt3⁻, long-
term hematopoietic stem cells (LT-HSC); LSK CD34⁻Flt3⁻, short-term HSC (ST-HSC);
LSK CD34⁺Flt3⁺, multipotent progenitors (MPP) of *Il1b* (LT-HSC control, n = 7;

NRAS-G12D⁺, n = 8; ST-HSC control, n = 8; NRAS-G12D⁺, n = 7; MPP, n = 8 per group),
*Il1rn* (LT-HSC control, n = 4; NRAS-G12D⁺, n = 6; ST-HSC, MPP control, n = 4; NRAS-
G12D⁺, n = 8) and *Il1rap* (n = 4 per group). **f** Frequency of TBM CD34⁻Flt3⁻ LT-HSC,
CD34⁻Flt3⁻ ST-HSC, CD34⁺Flt3⁺ MPP, CD11b⁺Gr-1^hif4/80⁻ granulocytes and CD11b
⁺Gr-1⁺f4/80⁺ monocytes expressing IL-1 receptor accessory protein (IL-1RAP, n = 4
per group). **g** NFκB transcription factor activity calculated based on NFκB target
gene expression levels, identified from Synapse ID syn4956655 (Supplementary
Data 1), from RNA sequencing data in LT-HSC, ST-HSC and MPP (n = 3 per group).
**h** Number of monocytes in peripheral blood (PB) from NRAS-G12D⁺ mice treated
with vehicle (n = 3) or bortezomib (n = 4). Data are biologically independent ani-
mals, and means ± S.E.M for bar plots or medians for violin plots. Statistical analyses
were performed with two-tailed Student's *t*-test (**a–d**, **f** LT-HSC, **g, h**) or two-tailed
Mann–Whitney *U* test (**e, f** except LT-HSC). *p* values ≤ 0.05 are reported. Source
data are provided as a Source Data file.

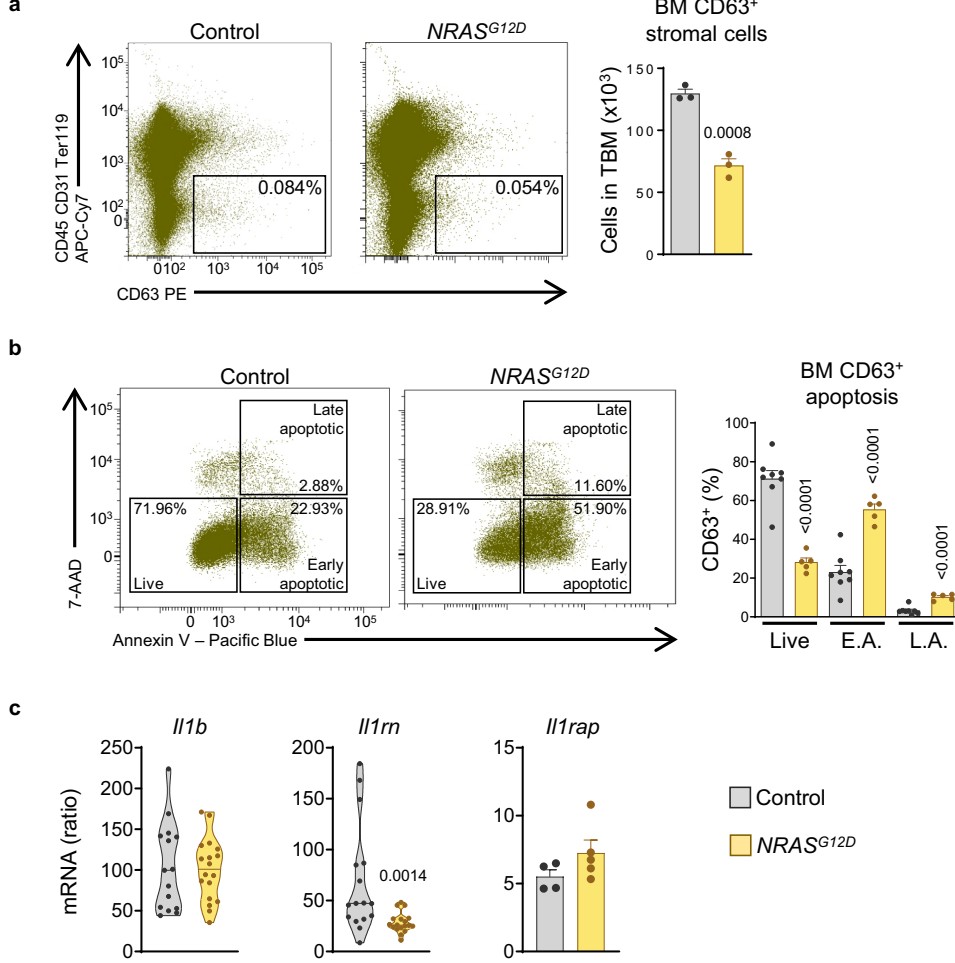

**Fig. 9 | Damage to the bone marrow stroma in a mouse model of pre-leukemic**
**myelopoiesis. a** Representative fluorescence-activated cell sorting (FACS) analysis
(cells in total bone marrow; TBM, %) and TBM number of CD45⁻CD31⁻Ter119⁻CD63⁺
stromal cells from control (*Mx1-Cre⁻ NRAS^G12D*) and NRAS-G12D⁺ (*Mx1-Cre⁺ NRAS^G12D*)
mice (n = 3 per group). **b** Representative FACS analysis (cells in BM CD63⁺ stromal
cells, %) and frequencies of live, early apoptotic (E.A.) and late apoptotic (L.A.) cells
within the CD45⁻CD31⁻Ter119⁻CD63⁺ stromal compartment in TBM from control

(n = 8) and NRAS-G12D⁺ (n = 5) mice. **c** qRT-PCR mRNA expression in
CD45⁻CD31⁻Ter119⁻CD63⁺ stromal cells of *Il1b* (control, n = 15; NRAS-G12D⁺, n = 18),
*Ilrn* (control, n = 16; NRAS-G12D⁺, n = 18) and *Il1rap* (control, n = 4; NRAS-G12D⁺,
n = 5). Data are biologically independent animals, and means ± SEM for bar plots or
medians for violin plots. Statistical analyses were performed with two-tailed Stu-
dent's *t*-test (**a, b, c** except *Il1rn*) or two-tailed Mann-Whitney U test (**c** *Il1rn*). *p*
values ≤ 0.05 are reported. Source data are provided as a Source Data file.

provides the advantages of potentially better regulated physiological
responses and opportunity to quickly discontinue IL-1β blockade in
case of infection. Efforts towards prolonging its half-life to some
extent, to prevent burden of daily injections, by fusion protein
technology and/or development of biodegradable polymers to
increase its steady-state sustained release are underway[73].

Functionally, loss of IL-1rn in vivo biases HSC differentiation into
the myeloid lineage and induces excess myeloid lineage expansion
reminiscent of early hematological disease, through IL-1r1 and IL-1β
overactivation in HSC but not IL-1α, in the absence of injury or infec-
tion. These findings are consistent with previous observations upon
long-term administration of high IL-1β doses[2], but further demonstrate

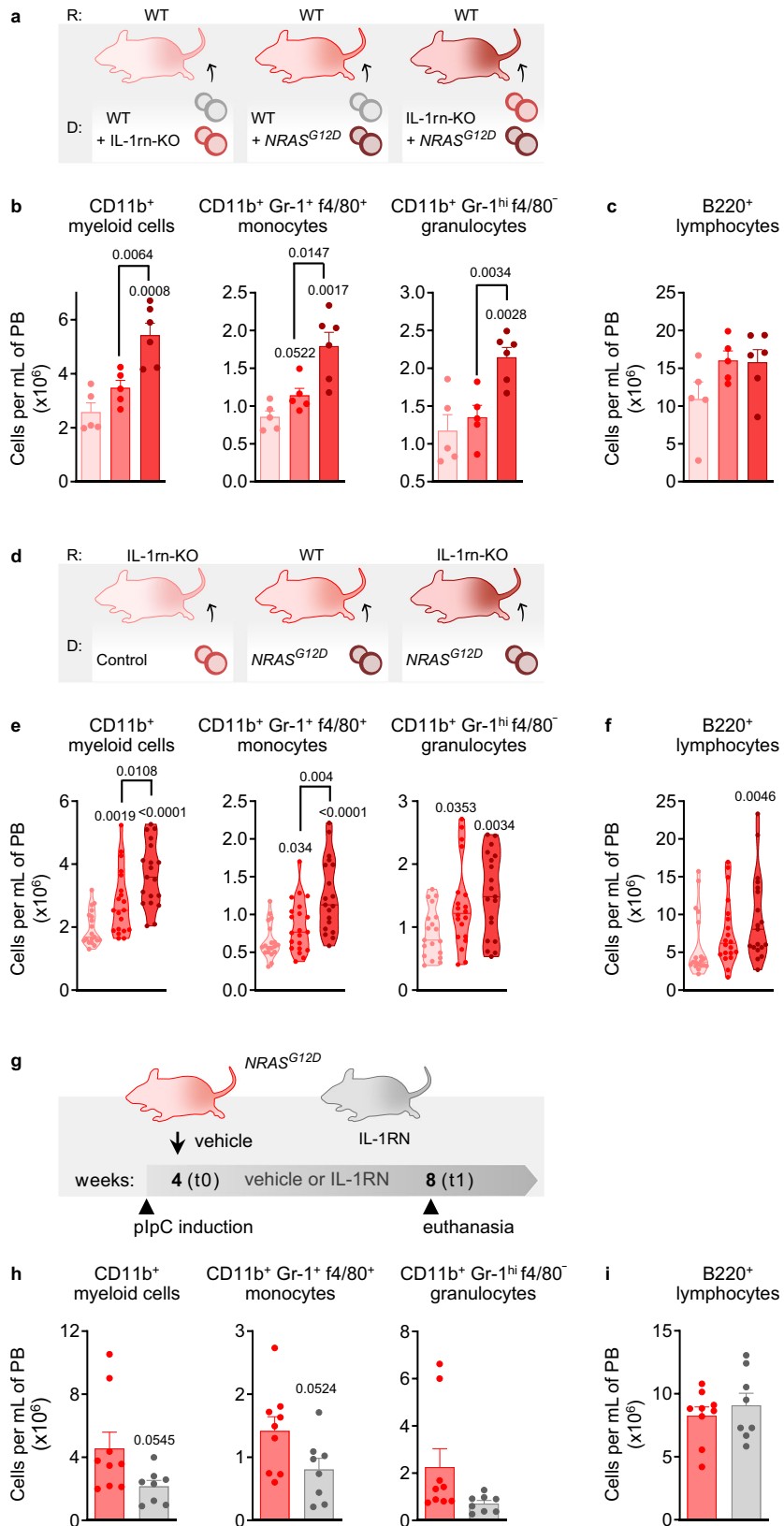

that IL-1rn represses HSC differentiation into the myeloid lineage driven by IL-1β under steady-state conditions. Our mouse studies showed that abnormal myelopoiesis after IL-1rn deletion is mediated mainly through transcriptional activation of IL-1β-induced myeloid differentiation pathways at least partially dependent on NFκB activation in HSPC, which partially overlap with the transcriptional programs activated by long-term administration of high IL-1β doses as shown by reanalysis of publicly available datasets[3,4] and are consistent with our findings in AML patients. Our results suggest that these effects are mediated mainly through the LT-HSC and ST-HSC compartments, with selective reduction in more primitive HSC and MPP5, and increase in MPP2 numbers and colonies ex vivo. Further immunophenotypic

**Fig. 10 | Deletion of IL-1rn from the hematopoietic or stromal compartments promotes pre-leukemic myelopoiesis, and it is therapeutically targetable.**
**a–c** C57BL/6J wild-type (WT) mice were used as recipients in competitive transplants (1:1) of bone marrow (BM) nucleated cells from WT and IL-1rn knockout (IL-1rn-KO) mice (*n* = 5), WT and NRAS-G12D[+] mice (*Mx1-Cre[+] NRAS^G12D*, *n* = 5) or IL-1rn-KO and NRAS-G12D[+] mice (*n* = 6). **a** Illustration of the experimental design. R: recipient. D: donor cells. **b** Number of CD11b[+] myeloid cells, CD11b[+]Gr-1[+]f4/80[+] monocytes and CD11b[+]Gr-1[hi]f4/80[−] granulocytes in peripheral blood (PB). **c** Number of B220[+] lymphocytes in PB. (**d–f**) C57BL/6J WT mice were used as recipients of BM nucleated cells from NRAS-G12D[+] mice (*n* = 20; monocytes *n* = 21), and IL-1rn-KO mice were used as recipients of BM nucleated cells from control (*Mx1-Cre[−] NRAS^G12D*, *n* = 19) and NRAS-G12D[+] mice (*n* = 19), over 4 independent experiments. **d** Illustration of the experimental design. **e** Number of CD11b[+] myeloid cells, CD11b[+]Gr-1[+]f4/80[+] monocytes and CD11b[+]Gr-1[hi]f4/80[−] granulocytes in PB. **f** Number of B220[+] lymphocytes in PB. **g–i** NRAS-G12D[+] mice were treated with vehicle (*n* = 9) or IL-1RN (*n* = 8). **g** Illustration of the experimental design. pIpC, poly-inosine:poly-cytosine. **h** Number of CD11b[+] myeloid cells, CD11b[+]Gr-1[+]f4/80[+] monocytes and CD11b[+]Gr-1[hi]f4/80[−] granulocytes in PB. **i** Number of B220[+] lymphocytes in PB. Data are biologically independent animals, and means ± SEM for bar plots or medians for violin plots. Statistical analyses were performed with two-tailed Student's *t*-test (**b, c, e** myeloid D NRAS-G12D[+]:R IL-1rn-KO vs R WT, monocytes D NRAS-G12D[+]:R WT vs D control:R IL-1rn-KO, D NRAS-G12D[+]:R IL-1rn-KO vs R WT, **h**) or two-tailed Mann–Whitney *U* test (**e** myeloid D NRAS-G12D[+]:R WT vs D control:R IL-1rn-KO, D NRAS-G12D[+]:R IL-1rn-KO vs D control, monocytes D NRAS-G12D[+]:R IL-1rn-KO vs D control, granulocytes, **f**). *p* values ≤ 0.05 are reported. Source data are provided as a Source Data file.

characterization of the BM of IL-1rn-KO mice showed that the myeloid lineage was expanded at the further expense of reduced numbers of MEPs and GMPs, whereas lymphoid lineage repression in the absence of IL-1rn did not involve active coordinated transcriptomic repression in LT-HSC or ST-HSC, or changes in immunophenotypically defined CLP numbers. Our scRNA-Seq analyses confirmed overall the immunophenotypic changes in the proportions of BM HSPC cells but showed qualitative expansion of MPP3 transcriptional state and reduction in lymphoid-biased MPP4 cells in the BM LSK compartment of IL-1rn-KO versus WT mice. Analysis of gene sets enriched at single-cell resolution revealed significant activation in pathways related to tumorigenesis and cell death in the HSPC subsets that were reduced i.e. HSC, HSC-MPP1, MPP4 and MPP5. Conversely, the expanded MPP2 and MPP3 compartments showed significant activation of pathways related to survival, whereas tumorigenic pathways were activated in MPP2 but slightly reduced in MPP3. These opposing and cell type-specific responses in HSPC subsets may be main factors underlying the chronic nature of the hematopoietic abnormalities in IL-1rn-KO mice, which will require additional investigation.

The expanded myeloid compartment in IL-1rn-KO mice may contribute to higher levels of IL-1β in the BM, as previously suggested under IL-1β administration[2]. Unlike myeloid cells, LT-HSC and ST-HSC showed sustained engagement of an IL-1β-positive feedback loop through IL-1r1, suggesting that HSPC are active modulators of inflammation rather than passive receivers. Of note, *Il1rap* gene expression was unchanged in LT-HSC and ST-HSC but reduced in MPP from IL-1rn-KO versus WT mice, which could contribute to the limited effects of IL-1rn deletion on MPP. Together, our data suggest that LT-HSC and ST-HSC play a key role in the chronic inflammatory process driven by IL-1rn deletion through IL-1β production and sustained supply of effector inflammatory cells, in the absence of immunogenic triggers. Selective regulatory mechanisms of *Il1b* expression and responsiveness in HSC, MPP, progenitors and fully differentiated myeloid cells, should be subject of future work.

The sc-RNA-Seq analysis and annotation of BM CD11b[+] myeloid cells showed similar proportions of cell types in the BM of WT and IL-1rn-KO mice, with only a slight increase in more immature G3 neutrophils and CMoP, and reduction in more differentiated G4 neutrophils and type II monocytes. In general, inflammation was a major pathway activated in all IL-1rn-KO CD11b[+] myeloid clusters except in G1 neutrophils that showed no changes and type I monocytes that displayed a reduced inflammatory signature. Thus, despite showing no engagement of an IL-1β-positive feedback loop, effector inflammatory cells are activated in the BM of IL-1rn-KO mice in the absence of immunogenic triggers. The potential functional consequences of this phenomenon on hematopoiesis should be studied in future work.

IL-1rn deletion induced alterations to the BM microenvironment, including fibrosis, reduced MSC numbers and their HSC-regulatory activity at the transcriptomic level, which were caused by IL-1β. This is in accordance with our previous observations in the IL-1β inflammatory conditions driven by JAK2-V617F[+] hematopoietic cells[15].

In view of the overlap with other relevant MSC markers and high expression of *Il1rn*, we used BM CD63[+] stromal cells as a surrogate for BM MSC. The sc-RNA-Seq analysis and annotation of BM CD63[+] cells underscored that they are indeed comprised by MSC, but also related-cell subsets i.e. fibroblasts, arteriolar and sinusoidal cells, and pericytes. Arteriolar and sinusoidal cells were reduced, and pericytes increased within the CD63[+] stromal compartment from IL-1rn-KO mice versus WT. Analysis of gene sets enriched at single-cell resolution highlighted the complex effects of IL-1rn deletion on the CD63[+] stromal compartment, activating gene signatures of differentiation and restricting proliferation in fibroblasts, as opposed to MSC, and stimulating endothelial cell development in vascular and perivascular clusters. Thus, our data indicate that both changes in cell composition and selective gene expression shifts may underlie alterations to the BM microenvironment in the absence of IL-1rn. The strong cell type-specific responses within the CD63[+] stromal compartment will require functional characterization.

Normal BM MSC switch between their pro- and anti-inflammatory phenotypes to initiate inflammation at early stages of damage sensing and to promote its resolution at advanced stages, respectively[74]. Our data suggest that BM CD63[+] stromal cells contribute to inflammation with *Il1b* expression in the absence of IL-1rn with no need of injury or infection sensing. Interestingly, our scRNA-Seq in CD63[+] BM stromal cells from WT mice was able to resolve that *Il1b/Il1rn* are mainly produced by MSC and arteriolar cells, and a small fraction of sinusoidal cells. Most cytokine-expressing MSC in fact express *Il1rn* only whereas arteriolar cells express mainly *Il1b* or both, with a small fraction expressing *Il1rn* only. Sinusoidal cells express *Il1b* only and they may be particularly involved in the induction of inflammation in the absence of IL-1rn, together with MSC.

In the BM of WT mice, our data at single-cell resolution further highlight that IL-1rn buffers mostly paracrine IL-1β feed-forward loops through its exclusive expression from subsets of CD63[+] MSC, neutrophils and monocytes, and a small fraction of CD63[+] arteriolar cells. Autocrine responses are rare and mainly in subsets of LT-HSC and CD63[+] arteriolar cells, and a small fraction of neutrophils mainly within G4, with expression of both cytokines. Selective regulatory mechanisms of *Il1rn* expression from different cell sources should be investigated in future work.

The hematopoietic phenotype observed in IL-1rn-KO mice is reminiscent of that observed in patients of the rare syndrome named deficiency in IL-1RN (DIRA). DIRA is a rare congenital disease caused by loss of function mutations in the IL1RN locus that results in death from severe auto-inflammation and multiorgan failure if not treated with anakinra[75]. DIRA patients suffer from hematopoietic abnormalities including leukocytosis and monocytosis, anemia, thrombocytosis, extramedullary hematopoiesis with enlargement of liver and spleen, BM fibrosis and osteosclerosis[75,76], reminiscent of early hematopoietic disease.

In further support of our data in human AML, we provide multiple evidence demonstrating unbalanced IL-1rn in aberrant myelopoiesis in mouse. We used the NRAS-G12D[+] model of pre-leukemic myelopoiesis

*Mx1-Cre NRAS^G12D*, based on IL-1β amplification mediated by oncogenic RAS described in other types of cancer[68]. This paradigm was true for NRAS-G12D[+] HSPC but no other major producers of IL-1β within the differentiated cell compartment, i.e. granulocytes and monocytes. Consistent with previous findings in human AML reproduced here[13,14], the percentage of LT-HSC, MPP and monocytes expressing membrane IL-1RAP was higher in pre-leukemic mice compared to controls. Whereas MPP were resistant to IL-1β-positive feedback loop in IL-1rn deficient mice, they expressed higher levels of *Il1b* in the presence of oncogenic NRAS, comparable to LT-HSC and ST-HSC. Thus, the high IL-1β levels in the BM of diseased mice were result of a HSPC-selective gene regulation event together with expansion of IL-1β-producing hematopoietic cell types, i.e. HSPC, granulocytes and monocytes, in NRAS-G12D[+] disease. Of note, MPP were the most expanded HSPC subset in NRAS-G12D[+] mice and this was coincident with their abnormally high *Il1b* expression, and selective downregulation of *Il1rn* across all hematopoietic cell types studied, including LT-HSC and ST-HSC. Thus, both human AML CD34[+] cells and mouse pre-leukemic MPP participate to unbalanced IL-1RN. Like IL-1β, selective regulation of IL-1rn production across hematopoietic cells was apparent and its underlying mechanism should be subject of future work. Congruent with our findings in AML patients and IL-1rn-KO mice, low IL-1rn contributed to the biased myelopoiesis in NRAS-G12D[+] mice through pathways at least partially dependent on NFκB activation, which was confirmed in two different mouse models, i.e. *Mx1-Cre NRAS^G12D* and *Vav-Cre NRAS^G12D*. Our data showed that BM CD63[+] stromal cells contributed to the lower levels of IL-1rn in the BM of NRAS-G12D[+] mice by their cellular fragility, as demonstrated by both their increased apoptotic rates and reduced cell numbers, and selective downregulated expression of *Il1rn*. Absence of IL-1rn from either the hematopoietic or the stromal compartments induced faster progression of *NRAS^G12D*-driven pre-malignant hematopoiesis. Thus, both compartments are relevant for IL-1rn production to repress myelopoiesis under neoplastic hematopoiesis. Our data are consistent with the idea that inflammation damages the BM microenvironment, which then establishes abnormal communication with HSPC and thereby plays a role in abnormal hematopoiesis[8,15]. We further suggest that this abnormal communication comprises deficient control of IL-1β-induced inflammation by BM MSC and arteriolar cells in the presence of a pre-leukemic lesion. The study of the potential role for IL-1RN anti-inflammatory properties of BM MSC and arteriolar cells in human AML will be of interest.

Together, loss of repression by IL-1rn resulting in enhanced stimulation of IL-1β signaling may underlie both pre-leukemic lesion and AML progression. Low IL-1rn and subsequent high IL-1β promotes myeloproliferation and inhibits the most primitive HSC, with the contribution of HSC to the low IL-1rn and high IL-1β. These conditions may pose a pressuring environment for the long-term selection of clones with lighter sensitivity to the inhibitory effect of low IL-1rn and high IL-1β. In theory, the potential benefit of this process would be to prevent the exhaustion of the HSC pool. However, HSC clones that thrive and expand in the presence of an otherwise inhibitory signal meet the criteria of functionally malignant leukemia stem cells, and selection of these clones would result in aggressive malignancy. Further work is required to validate this intriguing hypothesis, but if it holds true it could explain the association of low IL-1RN both with myeloproliferation at early or mild stages of disease and with a more primitive state and poor survival in AML patients at diagnose or relapse who are experiencing late or aggressive transformation events. It would also provide grounds to the therapeutic potential of IL-1RN at both stages.

Collectively, our data support that endogenous IL-1RN represses myelopoiesis in HSPC under steady-state and protects from neoplasia, whereas low IL-1RN is a hallmark of AML with prognostic value for reduced survival in AML patients, it contributes to myeloproliferation in the presence of a pre-leukemic lesion and provides a mechanistic rationale for IL-1RN boost or IL-1β blockade therapeutic potential in AML patients (Supplementary Fig. S15). We depict a potential origin for high IL-1β in human AML, and a type of patient that may be particularly good responder to anti-IL-1β therapies over patients that present with high IL-1β resulting from primary lesions in the signaling pathway downstream of IL-1R1.

## Methods

Our research complies with all relevant ethical regulations. Human studies were approved by the Regional Committee for Medical Research Ethics North Norway (REC North 2015/1082). Written informed consent was obtained in accordance with the Norwegian legislation and the Declaration of Helsinki. Participants received no compensation. Mouse experiments were conducted with the ethical approval of the Norwegian Food and Safety Authority (7141, 7960, 8043, 8252, 8408, 8660, 8667, 9005, 21740, 24009, 24065, 24736, 24737), the local Animal Care and Ethics Committees at UiT – The Arctic University of Norway (09/16, 09/21), University of Oslo (Norway; A015) and CNIC (Spain; PROEX 101/18), and in accordance with the Guide for the Care and Use of Laboratory Animals and approved by an Animal Care and Use Committee at UW-Madison (USA; M005328). The latter program is accredited by the Association for Assessment and Accreditation of Laboratory Animal Care.

### Humans
The diagnosis of AML and MDS were established according to the revised criteria of the World Health Organization. Cytogenetic risk group for AML patients was established according to Dohner and colleagues[77]. Risk group for MDS patients was determined according to the revised prognostic scoring of Greenberg and colleagues[78]. Patient group was formed by 24 AML patients and 2 MDS patients; 8 women (30.8%) and 18 men (69.2%) aged 66 years (range 52-94). All samples were collected at diagnosis time. Control group was formed by 35 healthy volunteers; 22 women (62.9%) and 13 men (37.1%) aged 42 years (range 22–79). Group characteristics of patient and healthy donor samples used in each experiment are provided in Supplementary Table S1. Subjects did not provide the full set of data.

### Mice
Age and gender matched *Il1rn*[−/−79], B6.SJL (CD45.1[+]), C57BL/6J, immunodeficient *NOD Scid Gamma* (NSG) mice expressing human *IL3, GM-CSF* and *SCF* (NSG-SGM3)[80,81] (The Jackson Laboratory), *Nes-gfp* (pure C57BL/6J background; gift from S. Mendez-Ferrer, own colony maintained at Charles River Laboratories CRL, Germany)[21], *Mx1-Cre NRAS^G12D*[64,82] (pure C57BL/6J background; gift from P. Garcia, own colony maintained at CRL) and *Vav-Cre NRAS^G12D*[67] (pure C57BL/6J background; J. Zhang) were used in experiments. Phenotyping of *Il1rn*[−/−] mice (pure C57BL/6J background; own colony maintained at CRL) versus C57BL/6J wild-type (WT) mice was performed in either females or males aged 7-61 weeks. *Cre* expression in *Mx1-Cre NRAS^G12D* mice was induced by intraperitoneal (i.p.) injection of one or two doses (in two consecutive days) of 300 μg of poly-inosine:poly-cytosine (poly-I:polyC, Sigma-Aldrich Merck). Phenotyping of *Mx1-Cre NRAS^G12D* mice versus *NRAS^G12D* control littermates was performed in either females or males aged 14–69 weeks, 6–44 weeks after polyI:polyC induction. Animals were used as donors at least 4 weeks after polyI:polyC induction and they displayed splenomegaly. Experimental animals were housed under specific opportunistic and pathogen free environment at the Unit of Comparative Medicine at UiT – The Arctic University of Norway or the Section of Comparative Medicine at University of Oslo. Single-cell RNA-Seq was performed in samples from *Il1rn*[−/−] versus C57BL/6J WT female mice, aged 13–15 weeks. These animals were kept in a specific pathogen free facility at CNIC (Spain). Phenotyping of *Vav-Cre NRAS^G12D* mice versus *Vav-Cre* control littermates was

performed in either females or males aged 6–7 weeks. These animals were housed in the Wisconsin Institutes for Medical Research Vivarium under specific pathogen free Level 2 condition. The maximal leukemic burden permitted was leukocytosis 5-fold over normal healthy values and it was not exceeded. Group and individual characteristics of animals used in each experiment are provided in Supplementary Table S1.

## Transplantation assays

BM transplantation was performed through the tail vein after myeloablation. For most myeloablation experiments, 6–20 week old female mice were whole body irradiated with 9 Gy (in 2 doses separated by 3 h) using an X-RAY source (Rad Source's RS 2000). 4 h after, mice were transplanted intravenously (i.v.) with $2 \times 10^6$ BM nucleated cells in 200 µL of phosphate-buffered saline (PBS, Gibco Thermo Fisher).

In competitive repopulation assays, B6.SJL (CD45.1$^+$) were used as recipients of BM nucleated cells from previously induced *Mx1-Cre NRAS$^{G12D}$* mice, mixed 1:1 with competitor BM nucleated cells isogenic to the recipient. Myeloablation in competitive repopulation assay using B6.SJL (CD45.1$^+$) mice as recipients was performed through i.p. injection of Busulfan (Busilvex®, Pierre Fabre Pharmaceuticals) at 25 mg/kg in 200 µL of saline solution (B. Braun).

To study the role of *Il1rn* deletion in hematopoietic cells to *NRAS$^{G12D}$* oncogene-driven expanded myelopoiesis, irradiated C57BL/6J WT mice were transplanted with equal numbers of C57BL/6J WT and *Il1rn$^{-/-}$* BM nucleated cells (1:1), C57BL/6J WT and *NRAS$^{G12D+}$* (from previously induced *Mx1-Cre NRAS$^{G12D}$* mice) BM nucleated cells (1:1), or *Il1rn$^{-/-}$* and *NRAS$^{G12D+}$* (from previously induced *Mx1-Cre NRAS$^{G12D}$* mice) BM nucleated cells (1:1). To study the role of *Il1rn* deletion in the stroma to *NRAS$^{G12D}$* oncogene-driven expanded myelopoiesis, irradiated *Il1rn$^{-/-}$* and C57BL/6J WT mice were transplanted with *NRAS$^{G12D+}$* (from previously induced *Mx1-Cre NRAS$^{G12D}$* mice) or control (from previously induced *NRAS$^{G12D}$* mice) BM nucleated cells.

To confirm the changes in BM mesenchymal stromal cells (MSC) from *Mx1-Cre NRAS$^{G12D}$* mice, irradiated *Nes-gfp* mice were transplanted with *NRAS$^{G12D+}$* (from previously induced *Mx1-Cre NRAS$^{G12D}$* mice) or control (from previously induced *NRAS$^{G12D}$* mice) BM nucleated cells. *Nes-gfp* male mice were 28 weeks old at the time of transplantation.

For AML xenografts, 7–16 week old male or female NSG-SGM3 mice were whole body irradiated with 2 Gy and, 4 h after irradiation, transplanted with peripheral blood mononuclear cells (PBMC, AML3) or CD34$^+$ (AML7, 9, 21, 22, 23) cells isolated from the BM of AML patients. The number of transplanted cells was $4 \times 10^4$–$10^6$ per mouse. Animals showing human engraftment <0.1% in BM were excluded.

Complete list of reagents used in this work is provided in Supplementary Table S2.

## In vivo pharmacological treatments

In vivo treatments were human IL-1β (Peprotech), human IL-1RN (anakinra – Kineret® Swedish Orphan Biovitrum, Sobi), mouse anti-mouse IL-1β mAb (1400.24.17, Invitrogen Thermo Fisher and Novus Biologicals) and mouse IgG1 kappa isotype control (P3.6.2.8.1, Invitrogen Thermo Fisher), Armenian hamster anti-mouse IL-1α mAb (ALF-161, BioXCell) and Armenian hamster IgG isotype control (polyclonal, BioXCell), human anti-human IL-1β mAb (canakinumab – ILARIS® Novartis) and bortezomib (Thermo Scientific Thermo Fisher).

In the NSG-SGM3 AML xenograft mouse model, treatments started 4–7 weeks post-transplant, when animals evidenced signs of engraftment. Human CD34$^+$ transplanted NSG-SGM3 mice were injected daily i.p. with 25 µg/kg human IL-1β in 100 µL of saline solution or saline solution alone for 16 weeks (AML7) or were injected every other day using the same doses for 4 weeks (AML9). Human BM CD34$^+$ transplanted NSG-SGM3 mice were injected subcutaneously (s.c.) every 12 h with 1600 mg/kg of human IL-1RN in 300 µL of anakinra or saline solution for 4 weeks (AML21, 22, 23). To compare IL-1RN with

IL-1β blockade, mice were injected twice i.p. with 22 mg/kg of anti-human IL-1β monoclonal antibody (mAb) canakinumab, 3 weeks apart. Human BM nucleated cell transplanted NSG-SGM3 mice were injected s.c. every 12 h with increasingly higher doses of human IL-1RN in 100 µL of saline solution or saline solution alone, as follows: 20 mg/kg for 18 weeks; 75 mg/kg for 4 weeks; 150 mg/kg for 7 weeks and 300 mg/kg for 7 weeks, for a total of 36 weeks (AML3).

For IL-1RN treatment in vivo, *Il1rn$^{-/-}$* mice (9–16 weeks) were injected s.c. every 12 h with human IL-1RN in 100 µL of saline solution or saline solution alone for 10 weeks. The dose used to rescue the damage in the stromal compartment was 75 mg/kg per injection. To rescue the damage in the hematopoietic compartment, IL-1RN was injected at 300 mg/kg per injection. For IL-1β mAb treatment in vivo, *Il1rn$^{-/-}$* mice (9–21 weeks) were injected i.p. with 10 mg/kg of anti-mouse IL-1β mAb or IgG1 kappa isotype control in 250 µL of PBS once per week for 10 weeks and twice per week for 8 weeks, for a total of 18 weeks. For IL-1α mAb treatment in vivo, *Il1rn$^{-/-}$* mice (15–22 weeks) were injected i.p. with 10 mg/kg of anti-mouse IL-1α mAb or IgG kappa isotype control in 100 µL of PBS twice per week for 8 weeks. For bortezomib treatment, *Il1rn$^{-/-}$* mice (18–34 weeks) were injected i.p. with 0.8 mg/kg of bortezomib in 100 µL of 0.5% dimethyl sulfoxide (DMSO, Sigma-Aldrich Merck) in PBS or 0.5% DMSO in PBS solution alone once per week for 2 consecutive weeks, and received a last injection 3 weeks after. Animals were analyzed 12 weeks after the start of the treatment.

Treatments were initiated after animals evidenced signs of disease in the *Mx1-Cre NRAS$^{G12D}$* mouse model of expanded myelopoiesis after induction. Primary mutant mice (12–13 weeks; 2 weeks after poly-I:polyC induction) were injected s.c. every 12 h with 600 mg/kg of human IL-1RN in 100 µL of anakinra or saline solution for 4 weeks. Primary mutant mice (59–62 weeks; 52 weeks after polyI:polyC induction) were injected i.p. with 0.8 mg/kg of bortezomib in 100 µL of 0.5% DMSO in PBS or 0.5% DMSO in PBS solution alone once, and received a second and third injection, 4 and 5 weeks after. Animals were analyzed 8 weeks after the start of the treatment.

## Mouse hematopoietic and stromal cell fraction extraction

Blood samples, bones and spleens were processed as previously described[15]. Blood samples from NSG-SGM3 mice were obtained through retro-orbital vein under regular volatile anesthesia or through the saphenous vein; the rest of mice were bled through the saphenous vein. Blood samples were collected in ethylenediaminetetraacetic acid (EDTA)-coated tubes (Microvette, Sarstedt) and analyzed with an hematological analyzer (ABX Pentra XL 80 with software v2.2.2, HORIBA or ProCyte Dx with software v00-34, IDEXX). For FACS analyses, red blood cells were lysed with 0.15 M NH$_4$Cl (Sigma-Aldrich Merck) for 10 min at room temperature (RT).

For BM nucleated cell extraction, immediately after euthanasia, mouse bones (femora, tibiae, hips, humeri, sternum, spine) were harvested, and surrounding tissue was removed. Bones were crushed in a mortar with PBS, and cell suspensions were collected and filtered using a 40 µm strainer (Corning) to obtain single-cell suspensions. Spleen samples were crushed and filtered. For BM stromal cell isolation, bones were crushed in a mortar with collagenase type I (Stem Cell Technologies) and incubated at 37 °C in water bath with orbital shaking at 65 rotation per min for 1 h. Red blood cells lysis was performed at 4 °C. All cell suspensions were then washed with PBS 2% heat-inactivated fetal bovine serum (FBS, Gibco Thermo Fisher), centrifuged 7 min at $380 \times g$ and resuspended in PBS for transplantation or PBS 2% FBS for other analyses. Cells were counted with Neubauer chamber for transplantation, or automated cell counters CASY equipped with OLS CASY v2.5 software (OMNI Life Science) and NucleoCounter® NC-200™ equipped with Nucleoview 2.3 (Chemometec) for other analyses.

For scRNA-Seq of FACS-sorted CD11b$^+$ myeloid cells and HSPC, BM nucleated cells were obtained by flushing BM from 1 femur from one

mouse and 1 tibia from another mouse of the same genotype flushed through centrifugation. The rest of the bones from both mice were pooled for CD63+ stromal cell sorting as previously described using collagenase type I[15]. Depletion of red blood cells was performed in buffer containing 0.15 M KH4Cl (Sigma-Aldrich Merck) and cells were resuspended in PBS 0.05% bovine serum albumin (BSA, Sigma-Aldrich Merck).

Samples from *Vav-Cre NRAS^G12D* experiments were prepared following the protocol described in previous work[83]. Blood counts were obtained using Hemavet 950FS (Drew Scientific). Red blood cells were lysed in NH4Cl solution (Stem Cell Technologies) and resuspended in PBS 2% FBS prior to antibody staining for flow cytometry analyses. Cells were isolated from BM by flushing, lysed in NH4Cl solution, resuspended in PBS 2% FBS and passed through 25 μm cell strainers to obtain single-cell suspensions prior to antibody staining.

### Human CD34+ cell enrichment

PBMC and BM nucleated cells from AML patients, and PBMC from healthy controls were obtained by density gradient centrifugation (Lympholyte®, Cedarlane), and were used for CD34+ cell enrichment using immune magnetic technology (Stem Cell Technologies), following manufacturer recommendations. PBMC, BM nucleated cells and CD34+ cell suspensions were counted with CASY cell counter and adjusted for further analyses. When studies were not performed immediately, cells were cryopreserved in FBS containing 10% DMSO and kept in liquid nitrogen tank. Frozen human cells were thawed, and DMSO and cell debris were washed with FBS, PBS 2% FBS and PBS in consecutive centrifugations at RT ($380 \times g$, 5 min).

### Fluorescence-activated cell sorting (FACS)

Antibodies used and their dilutions are listed in Supplementary Table S2. Cells were incubated with 4′,6-diamidino-2-phenylindole (DAPI, Sigma-Aldrich Merck) or DRAQ7 (BD Biosciences, anti-human IL-1RAP staining) for dead cell exclusion. Immunophenotype of HSPC was defined as linage-negative (Lin⁻), Sca-1⁺ and c-Kit⁺; LSK. LSK were further defined as long-term hematopoietic stem cells (LT-HSC; LSK CD34⁻Flt3⁻), short-term HSCs (ST-HSC; LSK CD34⁺Flt3⁻), and multipotent progenitors (MPP; LSK CD34⁺Flt3⁺). Detailed FACS analysis of the stem and progenitor cell subsets corresponding to HSC and MPP1-MPP5 was performed as previously described[27–29]. Long-term (LT-HSC) were further defined as HSC (LT-HSC CD150⁺CD48⁻). Short-term (ST-HSC) were further defined as MPP1 (ST-HSC CD150⁺CD48⁻), MPP2 (ST-HSC CD150⁺CD48⁺), MPP3 (ST-HSC CD150⁻CD48⁺) and MPP5 (ST-HSC CD150⁻CD48⁻). MPP were further defined as MPP4 (MPP CD150⁻CD48⁺). Lineage-negative hematopoietic progenitor subsets were defined as committed common lymphoid progenitors (CLP; c-Kit^low Sca-1^low CD127⁺), common myeloid progenitors (CMP; c-Kit⁺Sca-1⁻ (LK) CD34⁺FcRγ⁻), megakaryocyte erythroid progenitors (MEP; LK CD34⁻FcRγ⁻), and granulocyte-monocyte progenitors (GMP; LK CD34⁺FcRγ⁺). Analyses were acquired on LSRFortessa™ or Canto™ II cell analyzer (BD Biosciences) equipped with BD FACS-Diva™ Software v8.0.1 and 9.0 (BD Biosciences). Analyses were performed with BD FACSDiva™ Software v8.0.1 and v9.0, and Flow Jo v.9 and v10.7.1_CL.

For cell sorting, cells were stained with the specific antibodies to identify LT-HSC, ST-HSC and MPP after enrichment (RNA-Seq) or not (qRT-PCR, RNA-FISH) in lineage-negative cells by immune magnetic depletion of differentiated cells. Briefly, BM nucleated cells were incubated with biotin mouse lineage depletion antibody cocktail (BD Bioscences) followed by streptavidin magnetic beads (BD Biosciences) and magnetic depletion, according to the manufacturer recommendations. LSK cells were sorted from total BM with no enrichment step. For MSC sorting, cells were identified as CD45.2⁻Ter119⁻CD31⁻CD63⁺. Myeloid cells were identified as CD11b⁺, granulocytes as CD11b⁺Gr-1^hi f4/80⁻ and monocytes as CD11b⁺Gr-1⁺f4/80⁺, for cell sorting. Cells were

sorted using FACS Aria III cell sorter (BD Biosciences) equipped with BD FACSDiva™ Software v8.0.1 and 9.0. FACS-sorted CD11b⁺ myeloid cells, HSPC and CD63⁺ stromal cell sorting was performed as described above using FACS Aria II cell sorter (BD Biosciences) equipped with BD FACSDiva™ Software v6.1.3.

### Anti-mouse IL-1RAP antibody generation

An anti-mouse IL-1RAP mAb through standard phage display technology was generated at ProteoGenix (France), following their own manufacturing process, using the purified extracellular domain of mouse IL-1RAP protein (Sinobiological). Briefly, scFv rabbit naïve LiAb-SFRab™ phage library (ProteoGenix), with high diversity of $1.09 \times 10^{10}$ different clones, was incubated in a tube coated with mouse IL-1RAP (50 μg/mL) for 5 rounds. Elution of phage binders was performed with Glycine-HCl (Sigma-Aldrich Merck) followed by neutralization. Eluted phages were added to *E. coli* TG1 (Lucigen) to determine their concentration. Then, eluted phages were amplified using TG1 and helper phage (NEB), precipitated with PEG/NaCl (Sigma-Aldrich Merck) followed by resuspension and used in phage pools screening through polyclonal phage ELISA to detect phage clones with the best affinity for IL-1RAP. Amplified eluted phages from each round were incubated in a plate coated with 4 μg/ml of mouse IL-1RAP or blocking buffer alone (Thermo Fisher). To determine binding, anti-phage-horseradish peroxidase antibody (ProteoGenix) was incubated, followed by tetramethyl benzidine (ProteoGenix) and HCl. Readout was performed at 450 nm. Highest phage enrichment was found in panning rounds 4 and 5, as indicated by good specific binding to IL-1RAP and low background. Thus, these two rounds were selected for monoclonal ELISA. Single TG1 clones, 96 from round 4 and 96 from round 5, were randomly picked and cultured with helper phage. Clones were centrifuged and supernatants collected for monoclonal phage ELISA against mouse IL-1RAP, following the procedure described above. 28 positive clones were identified and sequenced, resulting in 3 different unique sequences identified. These 3 unique clones were retested by ELISA against mouse IL-1RAP, which confirmed specificity and strong binding to mouse IL-1RAP. Clone R5P1-A1 was the most redundant sequence obtained in the monoclonal screening, which means that it was highly enriched and it is likely a good binder for mouse IL-1RAP. This clone was expressed as a recombinant full-length rabbit IgG. The sequence of this scFv is as follows:

R5P1-A1 – VH: FR1-**CDR1**-FR2-**CDR2**-FR3-**CDR3**-FR4
QSVKESGGRLVTPGTPLTLTCTVS**GFSLSNYA**MSWVRQAPGKGLE WIGF**ISSSGST**YYASWAKGRFTISKTSTTVDLKITSPTTEDTATYFC**ARGW NYFNL**WGQGTLVTVSS

R5P1-A1 – VL: FR1-**CDR1**-FR2-**CDR2**-FR3-**CDR3**-FR4
ELVLTQPASVQVNLGQTVSLTCTAD**TLSRSY**ASWYQQKPGQAPVL LIY**RDT**SRPSGVPDRFSGSSSGNTATLTISGAQAGDEAAYYC**ATSDGSG STYQVV**FGGGTQLTVT

The cDNA of these variable heavy (VH) and variable light (VL) sequences were chemically synthesized in a suitable way for optimal expression in CHO-like mammalian cells, then sub-cloned into mammalian cell expression vector pTXs1 (ProteoGenix) to obtain the full-length sequences of the heavy (HC) and light (LC) chains of the rabbit mAb IgG of kappa type (A1). The expected corresponding protein produced is as follows:

R5P1-A1-VH
MKHL**W**FFLLLVAAPRWVLSQSVKESGGRLVTPGTPLTLTCTV SGFSLSNYAMSWVRQAPGKGLEWIGFISSSGSTYYASWAKGR FTISKTSTTVDLKITSPTTEDTATYFCARWNYFNLWGQGTLVTV SS**GQPKAPSVFPLAPCCGDTPSSTVTLGCLVKGYLPEPVTVT WNSGTLTNGVRTFPSVRQSSGLYSLSSVVSVTSSSQPVTCNVAH-PATNTKVDKTVAPSTCSKPTCPPPELLGGPSVFIFPPKPKDTL-MISRTPEVTCVVVDVSQDDPEVQFTWYINNEQVRTARPPLREQQF NSTIRVVSTLPIAHQDWLRGKEFKCKVHNKALPAPIEKTISKARGQ PLEPKVYTMGPPREELSSRSVSLTCMINGFYPSDISVEWEKNGKA**

EDNYKTTPAVLDSDGSYFLYSKLSVPTSEWQRGDVFTCSVMHEA
LHNHYTQKSISRSPGK

Features:

SIP: [1:19]

**Rabbit HC constant region: [132:454]**

R5P1-A1-VL

MVLQTQVFISLLLLWISGAY-
GELVLTQPASVQVNLGQTVSLTCTADTLSRSYASWYQQKPGQAPVL-
LIYRDTSRPSGVPDRFSGSSSGNTATLTISGAQAGDEAAYYCATSDGSG-
STYQVVFGGGTQLTVT**DPVAPTVLIFPPAADQVATGTVTIVCVAN-
KYFPDVTVTWEVDGTTQTTGIENSKTPQNSADCTYNLSSTLTLTST-
QYNSHKEYTCKVTQGTTSVVQSFNRGDC**

Features:

SIP: [1:20]

**Rabbit LC constant region: [131:233]**

An endotoxin-free DNA preparation was performed for the vector constructions obtained. Plasmids were transiently co-transfected in ProteoGenix in-house XtenCHO™ cell line (PX-XTE-001, 300 mL culture in animal product-free medium) using Xten transfection protocol (ProteoGenix). Culture medium was collected when viability dropped <50% (14 days post-transfection), and recombinant antibodies were purified on a Protein A resin (Thermo Scientific Thermo Fisher) using a standard method of clarification by $0.22\,\mu m$ filtration, equilibration, binding, and wash with PBS pH 7.5 (Gibco Thermo Fisher), elution with 20 mM citric acid pH 2.7, neutralization with 1 M Tris HCl (Sigma-Aldrich Merck) pH 9.0, and pool of fractions of interest and buffer exchange versus PBS pH 7.5 plus sterilization by $0.22\,\mu m$ filtration. The purified antibody had migration patterns in non-reduced/reduced SDS-PAGE with Coomassie blue staining (Sigma-Aldrich Merck) indicative of high purity (>90%), of integrity of HC and LC, and of antibody correct assembling. The purified antibody was concentrated to 1 mg/mL before performing FITC conjugation (Sigma-Aldrich Merck), which resulted in an average of 2.65 FITC molecules bound per antibody. The conjugation to FITC did not affect the antibody integrity, as assessed by non-reduced/reduced SDS-PAGE with Coomassie blue staining. As a result, we obtained 0.71 mg of R5P1-A1-FITC at 0.84 mg/mL, and purity >90% based on full-length antibody observed on non-reduced SDS-PAGE.

## Imaging flow cytometry

Quantification of NFκB nuclear localization was performed using Amnis® NFκB Translocation Kit (Luminex Corporation) following supplier protocol. Briefly, $10^5$–$5 \times 10^5$ human CD34+ progenitors enriched from PB of healthy donors or AML patients and cryopreserved, were fixed after thawing. Fixed cells were stained with anti-human NFκB (p50) Alexa Fluor 488 antibody and 7-Aminoactinomycin D (7-AAD) for nuclear staining. Samples were visualized and raw data recorded on an Amnis® ImageStream®X Mk II imaging cytometer using INSPIRE® software v200.1.620.0. Negative and positive controls of NFκB activation were acquired using fixed THP-1 cells (DSMZ, line ACC16) unstimulated or stimulated with lipopolysaccharide (LPS, Sigma-Aldrich Merck) at $1\,\mu g/mL$ for 1 h, respectively. To this end, THP-1 cells were maintained in Roswell Park Memorial Institute (RPMI) 1640 medium (Gibco Thermo Fisher) supplemented with 10% FBS and 1× penicillin–streptomycin-glutamine (Gibco Thermo Fisher), and replated before confluence every three days. Single-stained fixed THP-1 cells were used for compensation controls. Raw data were processed and analyzed with IDEAS® software version 6.2, using Nuclear Localization built-in analysis wizard[84]. Single cells were gated using Area and Aspect ratio features followed by gating of double-positive cells for NFκB and 7-AAD. Similarity between NFκB and 7-AAD images was then calculated for each cell by measuring the pixel intensity correlation between both channels, giving each cell a similarity score correlated with the proportion of NFκB localized in the nucleus. Mean similarity scores of each sample were used to compare control and AML groups.

## Phospho-flow

Phospho-flow was used to measure NFκB p65 phosphorylation. Human CD34+ cells enriched from PBMC of AML patients and healthy controls were fixed after thawing with $100\,\mu L$ of 4% formaldehyde (Thermo Scientific Thermo Fisher) per $10^6$ cells for 15 min at RT, washed with PBS and permeabilized with pre-chilled 90% methanol (VWR) for 10 min at 4 °C. Cells were washed twice with PBS 2% FBS and stained with Alexa Fluor 647 mouse anti-NFκB p65 (pS529) or correspondent IgG2b kappa isotype control according to the supplier protocol. Antibodies and dilutions used are listed in Supplementary Table S2.

## Proliferation and apoptosis assays

To analyze proliferation, mice were injected i.p. with 2 mg 5-Bromo-2-Deoxyuridine (BrdU) (BD Biosciences). Mice were euthanized 24 h post-injection and BM nucleated cells were collected as described above. Lineage-negative cells were enriched and stained with antibodies to identify CD34⁻ LSK (LT-HSC) and CD34+ LSK (ST-HSC/MPP) cells. After fixation, permeabilization and antigen-revelation, cells were co-stained with anti-BrdU APC and DAPI. Analysis of apoptotic cells was performed as previously described[15]. Stromal cells were enriched in CD45.2⁻Ter-119⁻CD31⁻ cells by immune magnetic depletion, using biotinylated antibodies against CD45.2, Ter-119 and CD31 (BD Biosciences) followed by addition of streptavidin magnetic beads (BD Biosciences) and magnetic depletion. BM nucleated cells or CD45.2⁻Ter-119⁻CD31⁻ stromal cells were stained with surface antibodies to identify LSK or CD45.2⁻Ter-119⁻CD31⁻CD63+ cells respectively, washed with cold PBS and subsequently stained at RT with Pacific Blue-conjugated Annexin V antibody and 7-AAD (Life Technologies Thermo Fisher) following manufacturer guidelines. Antibodies and dilutions used are listed in Supplementary Table S2.

## Cell culture

Colony-forming unit cell (CFU-C) assay was performed as previously described[85]. Briefly, $2 \times 10^4$ BM nucleated cells or $2 \times 10^5$ spleen cells were cultured in low-adherence 35 mm dishes for colony growth (Stem Cell Technologies) with 1 mL methylcellulose (MethoCult™, Stem Cell Technologies). Samples from each mouse were studied in duplicate. Colonies of more than 50 cells were counted after 7 days of incubation. For ex vivo differentiation, $4 \times 10^5$ lineage-negative cells/mL were cultured in 1.3 mL of Stem Span medium (Stem Cell Technologies), supplemented with 20 ng/mL thrombopoietin, 125 ng/mL stem cell factor and 50 ng/mL Flt3-ligand (Peprotech), in low-adherence 6-well plates (Stem Cell Technologies) for 5 days. Only suspensions with >65% lineage-negative cells were used. IL-1RN at 1000 ng/mL of anakinra or saline solution was added twice per day every 12 h. Fresh medium was added on day 3. Cells were harvested on day 5 and checked for differentiation by staining with anti-mouse CD11b Alexa Fluor 647 and flow cytometry. Samples were studied in triplicate. Antibodies and dilutions used are listed in Supplementary Table S2.

## Cytokine analyses

BM extracellular fluid (BMEF) was obtained as previously described and used for cytokine analyses[15]. BM extracellular fluid was isolated from one femur and one tibia of each mouse, and proteins were quantified (Pierce™ BCA Protein Assay Kit, Thermo Fisher) with spectrophotometer VersaMax microplate reader equipped with SoftMax Pro v5.4.1 (Molecular Devices). To measure the level of intracellular cytokines in human PB CD34+ cells, total proteins were isolated with Trizol (Tri Reagent®, Sigma-Aldrich Merck) and quantified. Intracellular protein samples were normalized to a concentration of 200 μg/mL and used for multiplex analysis. Multiplex immunoassays were performed using Bio-Plex 200 System with Bio-Plex Manager v6.2 (Bio-Rad). IL-1rn in mouse samples was analyzed with Mouse IL-1rn AimPlex bead-based immunoassay (Aimplex Biosciences) using LSRFortessa™ cell analyzer. Data were calculated using FCAP Array™ Software v3.

Results were expressed as pg per mg of protein. For human IL-1β, ratios versus IL-1RN were calculated for each individual.

## Immunofluorescence and histology

For immunofluorescence, femoral bones were processed as previously described[15], and 15 μm cryostat longitudinal sections were used. Antibodies and dilutions used are listed in Supplementary Table S2. At least 2 different sections and a total of 8 images at magnification 400X per animal were quantified using ImageJ v1.8.0. For histological studies, femoral bones were used as previously described[15]. For quantification, one section of the entire femur at 400X was used per mouse and analyzed with ImageJ.

For immunofluorescence, femoral bones were processed as previously described[15]. Briefly, femoral bones were collected and fixed overnight (O/N) with 4% freshly made formaldehyde, followed by decalcification for 1 week with 0.25 M EDTA (Sigma-Aldrich Merck) at 4 °C. Samples were further incubated with 15% and subsequently 30% sucrose (Sigma-Aldrich Merck), 24 h each, at 4 °C for cryoprotection, and embedded in Tissue-Tek® OCT™ (Sakura Finetek Europe). Samples were frozen on dry ice and stored at −80 °C until use. For IL-1r1 staining, 15 μm cryostat longitudinal sections (CM1350, Leica) of femoral bones were blocked with 2% BSA and permeabilized with 0.05% Triton X-100 (Sigma-Aldrich Merck) for 1 h at RT. Samples were incubated for 1.5 h with goat anti-mouse IL-1r1 (R&D systems) 1:50 in PBS 2% BSA 0.05% Triton X-100. Then, secondary donkey anti-goat IgG Cy3 (Abcam) staining was performed at 1:200 in PBS 2% BSA 0.2% Triton X-100 for 1.5 h. Antibodies used are listed in Supplementary Table S2. Nuclear staining was performed by incubation with DAPI (1 μg/mL) for 10 min. Shortly after addition of mounting medium (Vectashield, Vector Laboratories) to prevent photobleaching, images were acquired with confocal laser scanning microscope (Leica SP5) and Leica Application Suite Advanced Fluorescence software v4.0. At least 2 different sections and a total of 8 images at magnification ×400 per animal were quantified using ImageJ software.

For terminal deoxynucleotidyl transferase (TdT) dUTP nick end labeling (TUNEL, Sigma-Aldrich Merck) staining, 15 μm cryostat sections of the whole femoral BM were permeabilized with 0.5% Triton X-100 for 25 min and then 0.1% Triton X-100 in PBS (PBT) for 5 min at RT. Samples were stained with 150 μl of the following working solution: 400 μl TdT reaction buffer (containing 125 μl TdT buffer 5×, 25 μl CoCl₂ and 475 μl PBS, Invitrogen Thermo Fisher), 50 μl biotin-dUTP (Sigma-Aldrich Merck) and 50 μl TdT enzyme for 75 min at 37 °C. Then, Cy3-Streptavidin conjugate (ZyMAX™ grade, Invitrogen Thermo Fisher) was added at 1:200 in PBT for 1.5 h, followed by nuclear staining with DAPI. Negative control was stained with 150 μl TdT reaction buffer, and positive control was incubated with deoxyribonuclease I (DNase I, recombinant, ZYMO research) in DNase buffer (50 mM Tris HCl, pH 7.5, 1 mg/ml BSA) for 25 min at RT. Samples were mounted and confocal images were acquired with laser scanning confocal microscope (Zeiss LSM780) equipped with Confocal Zeiss software ZEN 2011 SP3 black edition (Release 8.1.0, Build 484). At least 2 different sections and a total of 8 images at magnification 400X per animal were quantified using Confocal Zeiss software.

For histological studies, femoral bones were fixed O/N in 4% freshly made formaldehyde, paraffined (VWR) and sectioned. Histological stainings were kindly performed by Roisin Doohan (Department of Comparative Medicine, Histopathology, CNIC). Hematoxylin (Sigma-Aldrich Merck) and eosin (Sigma-Aldrich Merck) conventional staining was performed on deparaffinized sections followed by re-hydration. Hematoxylin staining was performed using Harris solution and counterstaining with eosin Y solution. Sections were washed in distilled water, and quickly dehydrated by immersion into increasing concentrations of ethyl alcohol (70%, 95%, absolute; VWR). Sections were cleared with xylene (VWR) and mounted in DPX (Casa Alvarez). Histological images were acquired with Olympus BX51 equipped with digital camera (Olympus DP71) and Cell-F software (DP-BSW Ver.3.3.1).

To visualize collagen, Masson's trichrome staining was performed. Sections were fixed again for 1 h in Bouin's solution (Sigma-Aldrich Merck) at 56 °C and stained with Weigert's iron hematoxylin working solution for 10 min, followed by Biebrich Scarlet-acid fuchsin solution (Sigma-Aldrich Merck) for 10–15 min. Phosphomolybdic (Acros Organic) - phosphotungstic (Sigma-Aldrich Merck) acid solution was added for 10–15 min or until collagen lost the red staining. Sections were transferred directly to fast green solution (Sigma-Aldrich Merck) and stained for 1 min, briefly rinsed and incubated with 1% acetic acid (Riedel-de Haen) solution for 2–5 min. After a brief rinse in distilled water, dehydration was quickly performed in 70%, 95% and 100% ethyl alcohol. Finally, sections were cleared with xylene and mounted with DPX. For quantification, 1 section of the entire femur at 400X was used per mouse and diaphysis was analyzed with ImageJ software.

Gordon and Sweet's staining protocol was used to visualize reticulin fibers. Briefly, 1% acidified potassium permanganate (Sigma-Aldrich Merck) was used for 5 min to oxidize deparaffinized sections, followed by 1% oxalic acid (Sigma-Aldrich Merck) to decolourize. Then, sections were treated with 2.5% iron alum (Sigma-Aldrich Merck) for 15 min. Sections were impregnated in ammoniacal silver solution for 2 min and reduced with 10% aqueous formaldehyde (VWR) for 2 min. Ammoniacal silver solution is a filtered solution of 10% aqueous silver nitrate (Sigma-Aldrich Merck) and 3% sodium hydroxide (Sigma-Aldrich Merck) in distilled water, with NH₃ (Sigma-Aldrich Merck) to prevent precipitates. Afterwards the sections were incubated in gold chloride (Sigma-Aldrich Merck) for 2 min, fixed with 5% aqueous sodium thiosulphate (Sigma-Aldrich Merck) and counterstained with fast green for 15 s. Sections were rinsed in distilled water and quickly dehydrated by immersion into 70%, 95% and 100% ethyl alcohol. Sections were cleared with xylene before mounting with DPX. For quantification, 1 section of the entire femur at 400X was used per mouse and analyzed with ImageJ software v1.8.0.

## RNA fluorescent in situ hybridization (FISH)

LT-HSC, ST-HSC, MPP, granulocytes, monocytes and CD63⁺ stromal cells were FACS-sorted and cytospun at the following amounts per condition; $2 \times 10^4$ for granulocytes and monocytes, $1.6 \times 10^4$ for CD63⁺ stromal cells, $1.3 \times 10^4$ for ST-HSC and MPP, and $4 \times 10^3$ for LT-HSC. Cells were fixed 10 min with 4% formaldehyde and kept O/N at 4 °C. RNA-FISH was performed the next day using the ViewRNA™ ISH Cell Assay Kit (Invitrogen Thermo Fisher) following the kit instructions. Briefly, cells were permeabilized with Detergent Solution QC, digested with protease at 1:1000 dilution in PBS and incubated 3 h at 40 °C with no probes (negative control), probes against *Gapdh* (type 4, positive control) or probes against *Il1b* (type 4) and *Il1rn* (type 1 or 6 probes) at 5:100 in probe diluent. Samples were washed to remove probes not bound. Bound probes were amplified with PreAmplifier mix for 1 h at 40 °C, followed by Amplifier mix for 1 h at 40 °C. Label Probes (LP) targeting *Gapdh* (LP4-FITC), *Il1b* (LP4-FITC) and *Il1rn* (LP1-Cy3 or LP1-Cy5) were added for 1 h at 40 °C. Nuclei were counterstained with DAPI (Invitrogen Thermo Fisher), and samples were washed and covered with coverslip. Samples were scanned under confocal microscope (Zeiss LSM780). At least 3 representative pictures were acquired at magnification 630X and analyzed with ImageJ.

## RNA isolation and quantitative reverse transcription PCR

RNA isolation, reverse transcription and gene expression analysis were performed as previously described[15]. RNA isolation was performed using Trizol following the manufacturer guidelines. For FACS-sorted cells, Dynabeads™ mRNA DIRECT™ kit (Invitrogen Thermo Fisher) was used. Reverse transcription was performed using the High-Capacity cDNA Reverse Transcription System (Applied Biosystems Thermo

Fisher), following the manufacturer recommendations. Quantitative PCR were performed with QuantStudio 7 Flex Real-Time PCR System 384-well equipped with QuantStudio Real-Time PCR Software v1.3 (Applied Biosystems Thermo Fisher). The expression level of each gene was calculated by interpolation from a standard curve using QuantStudio Real-Time PCR Software v1.7.1. Briefly, a standard curve was performed by doing serial dilutions of cDNA after reverse transcription of mouse or human universal reference total RNA (Clontech TaKaRa Bio). All values were normalized with mouse *Gapdh* or *B2m*, or human *B2M* as endogenous housekeeping genes. qRT-PCR mRNA expression of *IL1RN* relative to *B2M* in immune magnetically enriched CD34$^+$ circulating progenitors from healthy volunteers was compared to AML patients classified according to FAB categories; M0-M3 versus M4-M5. The two MDS patients in the cohort were considered together with the AML M4-M5 group based on the presence but altered distributions of BM differentiated monocytic lineage cells in MDS patients. For human *IL1B*, *IRAK1*, and *CASP1* gene expression normalized to *B2M* was expressed as a ratio versus *IL1RN* normalized to *B2M* in each sample. Ratios in individual samples were then expressed as a fold change (FC) versus the mean ratio of control samples, and the log$_2$ of these values were calculated (log$_2$FC). Primers used are listed in Supplementary Table S2.

## RNA sequencing and bioinformatic data analysis

RNA-Seq data from FACS-sorted LT-HSC, ST-HSC, MPP and CD63$^+$ stromal cells obtained from the BM of *Il1rn$^{-/-}$* and C57BL/6J WT female mice aged 22-25 weeks have accession number GSE126428. RNA-Seq data from FACS-sorted LT-HSC, ST-HSC and MPP obtained from the BM of male *NRAS$^{G12D+}$* (induced *Mx1-Cre NRAS$^{G12D}$*) mice and control (induced *NRAS$^{G12D}$*) littermates 6 weeks after polyI:polyC induction, aged 34 weeks, have accession number GSE126625. RNA amplification, RNA-Seq library production and RNA-Seq analysis were performed at Eurofins Genomics (Germany). RNA-Seq data from FACS-sorted *Nes*-GFP$^+$ cells obtained from the BM of *Nes-gfp* male mice transplanted at the age of 28 weeks with BM cells from previously polyI:polyC induced control (induced *NRAS$^{G12D}$*) or *NRAS$^{G12D+}$* (induced *Mx1-Cre NRAS$^{G12D}$*) mice and analyzed 4 weeks after the transplant, have accession number GSE157038. RNA amplification and RNA-Seq library production were performed at the Genomics Support Center Tromsø (GSCT, UiT – The Arctic University of Norway).

Total RNA was isolated using the Arcturus PicoPure$^{TM}$ RNA isolation kit (Arcturus – Thermo Fisher) from small numbers of FACS-sorted cells ($6 \times 10^3$–$8 \times 10^4$).

For accessions GSE126428 and GSE126625, RNA was amplified and prepared for RNA sequencing (RNA-Seq) using the SMART-Seq® v4 Ultra® Low Input RNA kit (Clontech TaKaRa Bio). The RNA-Seq library was prepared with the Low Input Library Prep Kit v2 (Clontech TaKaRa Bio) to construct index-tagged cDNA. Libraries were sequenced on the Illumina HiSeq 2500 (Illumina) following the standard sequencing protocol with the TruSeq SBS Kit v4 and the following software versions: HCS 2.2.58, RTA 1.18.64. RNA amplification, RNA-Seq library production and RNA-Seq analysis were performed at Eurofins Genomics (Germany). At least 3 biological replicates were used per experimental group. The quality, quantity and the size distribution of the Illumina libraries were determined using the Fragment Analyzer High Sensitivity NGS Fragment Analysis Kit (Advanced Analytical). Fastq files containing reads for each library were extracted and demultiplexed using bcl2fastq-1.8.4 pipeline with bcl2fastq Conversion Software v1.8.4, followed by alignment and quantification using STAR v2.7.8, Samtools v1.9 and Gffread v2.2.1.3. Raw read counts were created using Subread v2.0.1. for GSE126428 or Rsubread v1.22.2 for GSE126625 with featureCounts[86]. Genome assembly GCF_000001635.25 and the associated annotation [https://www.ncbi.nlm.nih.gov/assembly/GCF_000001635.25] were used. Only reads overlapping coding sequences were counted. All reads mapping to the same gene were summed. Only

reads with unique mapping positions and a mapping quality score of at least 1 were considered for read counting. Supplementary alignments were ignored for read counting. Paired-end reads that mapped to different chromosomes or with unexpected strandedness were ignored. Paired-end reads were counted as single fragments. Reads mapping to multiple genes were assigned to the gene with the largest number of overlapping bases. Forward and reverse fastq reads, raw Counts, normalized counts per million (CPM), metadata and Eurofins Genomics sequencing statistics are accessible at GEO archive.

For RNA-Seq data analysis, RStudio v1.4.1106 and R v4.1.2 with the Bioconductor DESeq2 package v1.32.0 [https://bioconductor.org/packages/release/bioc/html/DESeq2.html] and edgeR v3.34.1 package [https://bioconductor.org/packages/release/bioc/html/edgeR.html] were used to generate lists of differentially expressed transcripts, Venn diagrams, volcano plots and principal component analysis (PCA) using ggplot2 v3.3.5, ggvenn v0.1.9, ggrepel v0.9.1 and patchwork v1.1.1. Gene symbols were derived using bitr and gseGO functions from the bioconductor clusterProfiler package [https://bioconductor.org/packages/release/bioc/html/clusterProfiler.html] v4.0.0 and the bioconductor org.Mm.eg.db v3.13.0 annotation library from the supplied entrez identifiers [https://bioconductor.org/packages/release/data/annotation/html/org.Mm.eg.db.html] to ensure accepted gene symbol nomenclature. Raw counts were used as recommended by the DESeq2 package. Only genes with at least 1 CPM in at least half of the samples in any of the LT-HSC, ST-HSC, MPP or CD63$^+$ conditions were kept; a total of 16552 transcripts in the IL-1rn-KO project (GSE126428). Only genes with an average mean raw count total of 16 or greater in any of the LT-HSC, ST-HSC or MPP were kept; a total of 11881 in the NRAS-G12D project (GSE126625). Differentially expressed transcripts were considered, with an adjusted $p < 0.05$ for each of the LT-HSC, ST-HSC, MPP, and CD63$^+$ stromal cells versus their respective controls. The DESeq2 results for LT-HSC, ST-HSC, and MPP were used to generate the Venn diagrams. Publicly available datasets of interest from GSE165810[3] and GSE166629[4] were reanalyzed using the same procedure as GSE126428 and used to generate Venn diagrams for comparisons with LSK subsets from this project. Expression level of selected genes in individual samples from GSE126428 was calculated as log$_2$FC of the normalized CPM in each IL-1rn-KO sample and the mean normalized CPM in WT samples.

The LT-HSC, ST-HSC, MPP and CD63$^+$ stromal cell volcano plots from GSE126428 show genes with an absolute $-0.5 > \log_2\text{FC} > 0.5$. Genes in red for LT-HSC, ST-HSC and MPP volcano plots have an adjusted $p < 0.05$. Genes in blue are identified as NFκB targets from cd34+_stem_cells_-_adult_bone_marrow transcription factor network from Synapse ID syn4956655[87], [https://bioinfo.lifl.fr/NF-KB] (Dr. K. Gosselin, Institute of Biology of Lille, France) and [https://www.bu.edu/nf-kb/gene-resources/target-genes] (Dr. T. Gilmore, University of Boston, USA). NFκB targets in human cd34+_stem_cells_-_adult_bone_-marrow transcription factor network[87] were determined by integrating transcription factor sequence motifs with promoter and enhancer activity data from the FANTOM5 project that uses cap analysis of gene expression (CAGE-analysis)[88], and they are available on Synapse (ID syn4956655). Human target genes were converted to the mouse genome using biomaRt v2.48.3[89]. We used two additional databases of NFκB target genes found in [https://www.bu.edu/nf-kb/gene-resources/target-genes] (Dr. T. Gilmore, University of Boston, USA) and [https://bioinfo.lifl.fr/NF-KB] (Dr. K. Gosselin, Institute of Biology of Lille, France), which contain mainly target genes in human with their corresponding RefSeq accessions and link to the publications where the association to NFκB was established. These compilations mainly include genes with checked binding, i.e. strong experimental evidence of NFκB direct control, such as chromatin immunoprecipitation or promoter transactivation, but also few putative genes, i.e. the gene has a κB site in the promoter, but has not clearly been shown to be controlled by NFκB, or the gene expression is associated with increased

NFκB activity, but has not been shown to be a target directly. They also contain checked and putative NFκB target genes in mice. Human target genes were converted to their mouse orthologs using the one2one association from Ensembl Compara Database. The complete NFκB gene list (n = 2132) is detailed in Supplementary Data 1. The plots depict the top 15 differentially expressed NFκB target genes, NFκB target genes of interest, and genes from those lists shared by another subset. Non-differentially ($p > 0.05$) expressed genes with expression lower than base mean count 500 counts per million were removed from all diagrams and few genes with extreme $log_2FC$ were removed to give a better visual representation. These lists were used to determine NFκB target genes in datasets of interest from GSE165810[3] and GSE166629[4] (Supplementary Data 1). Venn diagrams depict upregulated genes shared by all LSK subsets under comparison that are identified as NFκB targets. Genes in red for CD63$^+$ stromal cell volcano plot have an adjusted $p < 0.05$. The top 10 most differentially expressed genes and various genes of interest are labeled in the diagram.

In the gene set enrichment analysis (GSEA), only gene sets with an adjusted $p < 0.05$ were considered. The gseGO function from the clusterProfiler package was used to enrich for biological processes. Genelist was supplied as the Euclidian distance of $-log_{10}$ adjusted $p$ and $log_2FC$. This ensured that genes ranked with higher significance and higher FC values were given more weight. Kernel density plots using core enriched genes from enriched gene sets were plotted using the bioconductor ggplot2 package. The gseaplot2 function of the bioconductor enrichplot v1.12.1 package was used to draw the GSEA plots of the significantly enriched gene sets.

PCA in LT-HSC, ST-HSC, and MPP from the NRAS-G12D project GSE126625 was done using the plotPCA function of the DESeq2 package. The top 5000 most variable genes after rlog normalization were used to create the PCA plot.

The method for quantifying activity levels of NFκB targets in LT-HSC, ST-HSC and MPP from accessions GSE126428 and GSE126625 and LK from publicly available accession PRJNA774277[67], was adapted from pathway level analysis of gene expression[90]. The method performs PCA on the subset of the gene expression matrix that only includes those genes that are members of the gene set of interest (i.e. NFκB target genes from Synapse ID syn4956655[87]). The sample scores on the first principal component capture the major pattern of variation for the gene set and are used as an estimate of overall gene set activity. Three modifications were made to the original microarray-based method to adapt it for RNA-Seq data. First, due to the high number of genes with very low counts present in cell lines, the step in the algorithm where gene expression is normalized by variance was removed. This reduces the risk of low count genes being overestimated by the normalization. Second, as activity levels are compared between different datasets, it is necessary to correct for the arbitrary choice of positive direction of PCA. As such, a check of the PCA loadings between datasets was included and the positive direction chosen to obtain the maximum consistency between the PCA loadings, or rotation matrixes, between datasets. Third, to simplify interpretation and plotting, the scale was adjusted by subtracting the minimum activity score for a dataset to obtain only positive activity values. Stem cell relevant NFκB target genes were taken from the human cd34+_stem_cells_-_adult_bone_-marrow_derived network found in Synapse ID syn4956655[87]. Human target genes were converted to the mouse genome using biomaRt v2.48.1[89] before their use as a gene set to calculate NFκB activities. Per sample transcription factor activities between groups were compared using a standard t-test. This NFκB gene list (n = 1832) is detailed in Supplementary Data 1.

For accession GSE157038, RNA was amplified with the NEBNext Single cell/Low input RNA libray kit v. 3.0 (NEB#E6420) following the protocol to the fragmentation step. cDNA was fragmented to target size of 150 base pairs with Covaris M220. Libraries were prepared with TruSeq RNA library preparation kit v. 2.0 (RS#122-2001,

Illumina) starting the protocol from "perform end repair" to final library. Libraries were sequenced with the NextSeq550 instrument with System Suite v4.0 (Illumina) at the GSCT (UiT – The Arctic University of Norway). Two biological replicates were used per experimental group. Fastq files were trimmed for adapters using flexbar as recommended by the NEBNext® Single Cell/Low Input cDNA kit workflow. Transcript quantification of fastq files were performed using Salmon v1.5.1[91] with the commands–numBootstraps 100–seqBias–gcBias–validateMappings–minScoreFraction 0.2–consensusSlack 0.1. Gene level estimation was performed with Sleuth v0.30.0[92]. Forward and reverse fastq reads, differential gene expression and normalized CPM from FACS-sorted Nes-GFP$^+$ (GSE157038) are accessible at GEO archive. GSEA was performed as previously described using curated gene sets from the Molecular Signatures Database.

Single-cell RNA-Seq data from FACS-sorted LSK, CD11b$^+$ myeloid cells and CD63$^+$ stromal cell cells obtained from the BM of Il1rn$^{-/-}$ and C57BL/6J WT female mice aged 13–15 weeks, have accession number GSE197594. ScRNA-Seq library production was performed at the Genomics Unit of CNIC (Spain). Cells were quantified and viability was checked using the Countess III cell counter with software v1.0.296.782 (Thermo Fisher). Single cells were encapsulated into emulsion droplets using the Chromium Controller (10x Genomics). Each cell suspension was loaded into one port of a Chromium Next GEM Chip G (10x Genomics) with a target output of $4 \times 10^3$–$10^4$ cells. ScRNA-Seq libraries were prepared using the Chromium Next GEM Single-Cell 3′ Kit v3.1 (10x Genomics) following the manufacturer instructions using a SureCycler 8800 thermal cycler (Agilent Technologies). The average size of each library was calculated using a high sensitivity DNA chip on a 2100 Bioanalyzer (Agilent Technologies) and the concentration was determined using the Qubit fluorometer (Invitrogen Thermo Fisher). Individual libraries were diluted to 10 nM and pooled for sequencing. Library pool was sequenced at 700 pM in paired-end reads (28 bp Read1, 10 bp Index1, 10 pb Index 2 and 90 bp Read2) using one P3 flow cell (100 cycles) on a NextSeq 2000 with Controller v1.4.1 and RTA3 (Illumina). FastQ files for each sample were obtained using cellranger mkfastq pipeline (10x Genomics).

Extraction and demultiplexing of the scRNA-Seq was performed with bcl2fastq v2.20.0.422, cellranger demux v6.1. Cell Ranger v6.1.2 was used to align the sequencing reads to the mm10 mouse transcriptome cellranger 1.2.0 build [https://cf.10xgenomics.com/supp/cell-exp/refdata-cellranger-mm10-1.2.0.tar.gz], filtering, barcode counting and to quantify the expression of transcripts in each cell. All downstream analyses were implemented using R v4.0.3 and R v4.1.1 with RStudio 2022.07.2, and the package Seurat v4.0.5[93]. For each FACS-sorted experiment (CD11b$^+$, LSK and CD63$^+$ stromal cells), raw counts from WT and IL-1rn-KO samples were merged and low-quality cells were filtered removing those with less than 2000 counts, less than 800 genes and more than 10% mitochondrial genes content. WT and IL-1rn-KO samples were then normalized, log transformed and integrated together using the Canonical Cross-correlation Analysis (CCA) algorithm with default parametrization as implemented in Seurat R package to avoid technical biases. Integrated samples were then clustered using Louvain algorithm over a Shared Nearest Neighbor (SNN) graph using 20 PCs and a 0.5 resolution for the CD11b$^+$ and LSK experiments and 0.1 for the CD63$^+$ stromal cell experiment. tSNE based dimensionality reduction was used for representation purposes. ggplot2 v3.3.5 was used to draw tSNE plots, bar-charts and pie charts.

WT cells from each cluster were used for annotation. To annotate the cells within the CD11b$^+$ experiment, we used the R package SingleR v1.4.0[94] with the Immunological Genome Project (ImmGen) database[37], annotating by cluster. After that, we manually projected the expression of signature genes to re-annotate any wrongly assigned clusters. Gene signatures imported from GSE137539 dataset were scored over neutrophil-like subsets[38] and those from GSE131834 dataset were

scored over monocyte-like clusters[39], using the AddModuleScore from Seurat package. Cluster assignment to each cell type was obtained using the maximum enrichment score ±0.2 as previously described[29] and subsequent manual curation. To annotate the LSK experiment, we used the same procedure described above with gene modules from Cabezas-Wallscheid and Trumpp LSK data (E-MTAB-9208)[29]. CD63+ stromal cell experiment was annotated using the same procedure with gene signatures obtained from Scadden BM stromal data (GSE128423)[35]. For the latter, gene signatures were obtained from the top 100 markers for each cell type with a log2FC over 0.25, not duplicated between cell types and present in our dataset.

We performed differential expression analyses between IL-1rn-KO and WT cells for each annotated cell type using two-tailed Wilcoxon Rank Sum test. For all differential expression analyses, only the genes detected in a minimum percentage of 10% of the cells in either of the two populations under comparison were reported and the testing was limited to genes which showed, on average, at least 0.25-fold difference (log2) between the two groups of cells. Complete lists of detected genes are provided for each FACS-sorted experiment; CD11b+, LSK and CD63+ stromal cells, as well as for all annotated clusters within those cell subsets (Supplementary Data 2-4).

Ingenuity Pathway Analysis (IPA) from Qiagen was used to perform an enrichment analysis over the Diseases & Functions knowledge base with the set of significantly (Benjamini-Hochberg adjusted p value <0.05) differentially expressed genes for each cell type. From each function or disease, we only kept statistically significant enrichments (Benjamini-Hochberg adjusted p value) using a right-tailed Fisher's Exact Test.

Normalized counts for all genes and samples were downloaded from publicly available scRNA-Seq analysis to characterize mouse BM mesenchymal stromal cells GSE108892[34] and used to generate t-SNE plots visualizing the expression level of selected genes in the identified cell clusters.

Expression data using normalized signal intensities were downloaded from GSE14468[23–25], and used for survival analysis of 381 AML patients, containing 164 M4-M5 AML patients, and violin plots of 395 AML patients comparing the FAB classification of AML subtypes M0-M3 versus M4-M5. An average of probes '212657_s_at', '212659_s_at', '216243_s_at' specific to the human gene IL1RN ENST00000409930 was used in the analysis of survival. Probe "216243_s_at" specific to the *IL1RN* transcript 1, NM_173842, was used in the analysis of expression. Survival data were gently provided by contributor author Dr. P. J. M. Valk (Department of Hematology, Erasmus University Medical Center, Rotterdam, The Netherlands). Patients with overall survival of less than 16 days, patients aged less than 18 years of age and patients of unknown age or unknown cytogenetic risk were excluded from the study. Survival curves were plotted for the entire population and for the M4-M5 subpopulation with the function ggsurvplot from the survminer R package, and the Cox proportional hazards regression model. *IL1RN* low and high expressing samples were determined in the global population based on continuous gene expression values and classified using an optimal cutoff based on the split with minimal p value, which had the most significant association with overall survival. The optimal cutoff was identified as 7.970175 (log2 normalized signal) corresponding to the 91st percentile. Cox proportional hazards regression analysis was perfomed using log2-transformed gene expression values. Likelihood ratio test was used to test significance. RStudio v1.4.1106 and R v.4.1.2 with the R packages data.table v1.14.0, ggplot2 v3.3.3, GEOquery v2.62.2, gridExtra v2.3 and survival v3.2-11 were used in the analysis.

Expression data using Reads Per Kilobase per Million mapped reads (RPKM) were downloaded from GSE83533[30], and used for relapse-free probability analysis and violin plots of 19 matched-pair diagnosis-relapsed AML patients. Accompanying relapse-free time data were downloaded from the accession phs001027 from the database of Genotypes and Phenotypes (dbGaP) using the dbGaP File Selector and the SRA Run Selector. Relapse-free time data were plotted using the Kaplan-Meier method. *IL1RN* low and high expressing samples were classified using a mean cutoff. Cox proportional hazards regression analysis was perfomed using log2-transformed counts, as previously described. Log rank test was used to test significances. RStudio v1.4.1106 and R v.4.1.2 with the R packages data.table v1.14.0, ggsci v2.9, ggplot2 v3.3.3, survival v3.2-11 and survminer v0.4.9 were used in the analysis.

## Statistical analyses

Data are expressed as bar plots with mean and standard error of the mean (SEM) when $n < 10$, and as violin plot with median when $n \geq 10$. Individual data points are shown for human and mouse studies. Expression of selected genes from scRNA-Seq studies are expressed as violin plots with mean and no individual data points for cells. Statistical analyses were performed using Prism 9 software (GraphPad) and the R software environment. Statistical significance was evaluated by unpaired two-tailed Student's *t*-test or Mann–Whitney *U* test where appropriate. Statistical significance in competitive repopulation assays in B6.SJL mice and in *IL1RN* RNA-Seq data from dbGaP accession phs001027 (GSE83533) from paired diagnosis-relapse AML patient samples were assessed with paired two-tailed Student's *t*-test. Age-adjusted and cytogenetic risk-stratified survival curves from GSE14468 were carried out using the Cox proportional hazards regression model. Likelihood ratio test was used to test significance. Relapse-free probability analysis from GSE83533 was carried out through the Kaplan–Meier method. Cox proportional hazards regression model followed by log rank test was used to test significances. A *p* value less than 0.05 was considered significant. Adjusted *p* values in RNA-Seq and scRNA-Seq data analyses were derived from raw *p* values assessed with two-tailed Wald test or two-tailed Wilcoxon Rank Sum test, respectively, corrected for multiple testing using the Benjamini-Hochberg procedure. GSEA in RNA-Seq was also corrected with Benjamini–Hochberg for multiple comparisons. Statistically significant enrichments in IPA from scRNA-Seq were assessed using a right-tailed Fisher's Exact Test with Benjamini-Hochberg correction for multiple comparisons.

Mice were randomized to treatment groups, without blinding. Animals that showed symptoms of disease or health issues unrelated to aberrant myelopoiesis were excluded from the study, i.e. obesity, loss of weight unrelated to experimental conditions or excessive, tumor masses, skin inflammation, etc. Criteria applied for mouse termination before the established end point were in accordance with the Norwegian Food and Safety Authority. Not pre-established statistical outliers in mouse studies were excluded using Grubbs or Dixon test where appropriate. For in vivo experiments, sample size was estimated based on the minimum number of animals required to obtain biologically meaningful results in studies of hematopoiesis. Number of repetitions are specified when in vivo experiments were repeated several times. Cohort size and numbers of cells transplanted in xenografts was informed by the total number of available cells.

For human studies, all available samples were used. Outliers in human studies were not excluded.

## Reporting summary

Further information on research design is available in the Nature Portfolio Reporting Summary linked to this article.

## Data availability

All data generated in this study are provided in the article file, Supplementary Information, Supplementary Data and Source Data files. The

various RNA-Seq data generated in this study have been deposited in the GEO database under accession codes: GSE126428, GSE126625, GSE157038. The scRNA-Seq data have accession code: GSE197594. Public datasets and databases used in this study in addition to the ones generated are the following: GSE14468, GSE165810, GSE166629, GSE137539, GSE131834, GSE128423, GSE83533, GSE108892, E-MTAB-9208, Synapse ID syn4956655, [https://bioinfo.lifl.fr/NF-KB] (Dr. K. Gosselin, Institute of Biology of Lille, France), [https://www.bu.edu/nf-kb/gene-resources/target-genes] (Dr. T. Gilmore, University of Boston, USA), PRJNA774277, Genome assembly GCF_000001635.25, Ensembl Compara Database [https://www.ensembl.org/info/docs/api/compara/index.html], Molecular Signatures Database [https://www.gsea-msigdb.org/gsea/msigdb], Immunological Genome Project (ImmGen) database [https://www.immgen.org], phs001027 from the database of Genotypes and Phenotypes (dbGaP) [https://www.ncbi.nlm.nih.gov/projects/gap/cgi-bin/study.cgi?study_id=phs001027.v4.p1]. Further information and requests should be directed to and will be fulfilled by the corresponding author, L.A. (lorena.arranz@uit.no). Source data are provided with this paper.

## Code availability

The code to reproduce the main analyses presented in this manuscript will be available on GitHub upon publication (https://github.com/SAC-lab-UiT). Requests for scripts related to bioinformatics should be directed to and will be fulfilled by the corresponding author, L.A. (lorena.arranz@uit.no).

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

## Acknowledgements

We thank K. Tasken, J. Saarela and the NCMM at the University of Oslo (UiO), S. Kanse (UiO) and B. Smedsrød (UiT), for access to facilities. We acknowledge Center for Medical Genetics and Molecular Medicine, Haukeland University Hospital (Bergen, Norway) and R. Hovland for karyotyping, FISH, translocation and DNA analyses of AML and MDS patients included in this study, and Department of Pathology, Oslo University Hospital (Oslo, Norway) and S. Spetalen for deep sequencing. L.M. Gonzalez, L.T. Eliassen, X. Zhang, M. Ristic and other members of L. Arranz group, O.P. Rekvig, R. Doohan, L.D. Håland, M.I. Olsen, A. Urbanucci, J. Landskron, K.B. Larsen, R.A. Lyså and UiT Advanced Microscopy Core Facility, UiO and UiT Comparative Medicine Units, for assistance. P. Garcia and S. Mendez-Ferrer for providing *NRAS^{G12D}* and *Nes-gfp* mice, respectively. P. Garcia and L. Kurian for careful reading of the manuscript. E. Tenstad (Science Shaped) for artwork in schematics. We would also like to thank the AML and MDS patients, and healthy volunteers, who donated biological samples. Our work is supported by a joint meeting grant of the Northern Norway Regional Health Authority, the University Hospital of Northern Norway (UNN) and UiT (Strategisk-HN06-14), Young Research Talent grants from the Research Council of Norway, (Stem Cell Program, 247596; FRIPRO Program, 250901), and grants from the Norwegian Cancer Society (6765150), the Northern Norway Regional Health Authority (HNF1338-17), and the Aakre-Stiftelsen Foundation (2016/9050) to L.A. *Vav-Cre NRAS^{G12D}* experiments were supported by NIH grant R01CA152108 to J.Z.

## Author contributions

A. Villatoro and V.C. performed experiments, analyzed data, prepared figures and contributed to Methods writing. A. Villatoro performed in vivo treatments. V.C. performed IL-1α mAb treatment, set up methodology, compiled source data and contributed to bioinformatic analyses. A. Bernal, M.F., A.T. and J.K. performed experiments. C.T. performed bioinformatic analyses of RNA-Seq data from IL-1rn-KO versus WT mice, and together with A.R. and F.S.-C. performed bioinformatic analyses of scRNA-Seq data. I.C., A. Benguría, E.V., A.D. and A.H. performed scRNA-Seq. M.F. set up methodology. P.U. performed relapse-free probability analysis in AML patients, bioinformatic analyses of RNA-Seq in *Nes*-GFP^+ cells, contributed to GSEA and analyzed genes of interest in GSE108892. X.Y. and J.Z. provided hematopoietic data from *Vav-Cre NRAS^{G12D}* versus control mice. C.G.F. performed bioinformatic analyses of RNA-Seq data from *Mx1-Cre NRAS^{G12D}* versus control mice. R.H.P. performed RNA-Seq in *Nes*-GFP^+ cells. A. Vik recruited AML patients to the study and provided clinical data. E.A. performed survival analyses in AML patients and quantified activity levels of NFκB targets. A.H. contributed to data discussion and interpretation. L.A. conceptualized the overall study, set up methodology, designed, performed and supervised the experiments and bioinformatic analyses, performed in vivo transplantation assays, analyzed and interpreted data, prepared figures and wrote the manuscript. All authors edited the manuscript and agreed on its final version.

## Funding

## Competing interests

The authors declare no competing interests.
