## [Peer Review File · Nature Communications]

Endogenous IL-1 receptor antagonist restricts healthy and malignant myeloproliferationREVIEWER COMMENTS

Reviewer #1 (Remarks to the Author):

In this paper by Villatoro et al the authors study the endogenous IL-1 receptor antagonist IL1RA/IL1RN in healthy and malignant hematopoiesis. They show that lower IL1RN RNA expression levels correlate with worse prognosis in AML patients; and that in mice with deletion of IL1RN myelopoiesis is increased and correlates with corresponding molecular signatures, as well as defects in the bone marrow stroma are detectable. They also demonstrate lower IL1RN levels in an NRASG12D mutant mouse model, and that some of these effects are mediated by IL1RN-deficient stroma, which is one of the most novel aspects of the paper. While overall interesting and on an important topic, the paper suffers from several major weaknesses that would be critical to address to draw any meaningful conclusions.

Major:

1. There are inconsistencies in the presentation of the survival curves (Figure 1). The effect seems modest and is borderline significant. Also, 2 contradictory significance values are provided (0.048 in the text; 0.035 in the figure) – which one is correct ?
2. The authors found particularly low IL1RN level in M0-M3 AML, which are the more immature/aggressive subtypes. This begs the question of what are the IL1RN levels in normal HSPC of the different lineages. Is IL1RN naturally low in more immature HSPC and therefore co-tracks with prognosis, without actually being downregulated if one compares with appropriate cell types ?
3. This is a particularly relevant question as the data provided in Figure 1C is not at all convincing and indeed shows no relevant/significant difference (the case numbers are low; and looking at the data the effect is extremely modest, driven by only 2 events/outliers, and it is hard to reconcile how the p value of this can be significant – how was this calculated). Another question on this experiment: were G-CSF mobilized healthy PB CD34+ cells used ? If not, how did the authors obtain enough cells to run these assays (CD34+ cells are extremely rare in healthy PB).
4. In the xenograft experiments, how do the authors distinguish engrafted AML cells from normal HSPC (which are always co-transplanted). Is it possible that IL1RN just reduces myelopoiesis in general, irrespective of healthy/AML; i.e. is there any specificity to AML cells ?
5. In all human studies (and also the mouse studies in Figure 2), did the authors also look at IL1RAP levels on their cells ? This seems to be essential for 2 reasons: (1) IL1RAP has been reported as a predominant mechanism of IL1R pathway activation in AML (70% of patients), and (2) IL1RN acts via disrupting IL1R1-IL1RAP interaction which is essential for signaling. Without that information it is hard to truly interpret any of the data presented.
6. It is unclear to this reviewer why the authors analyzed transcriptional programs in the NRAS-G12D model. Just because these mice have (modestly) reduced IL1RN levels (amongst many other alterations) does not mean the measured GE profiles tell us anything about what IL1RN does. Similarly, what was the rationale for the bortezomib treatment ? This is a broad spectrum proteasome inhibitor with very pleiotropic effects. This also seems to be completely unrelated to any studies related to IL1RN.
7. What appears to be a critical experiment that is missing in this context would be treatment with either IL1RN (to restore normal levels) or canakinumab to demonstrate that this pathway is indeed functionally relevant in the context of NRASG12D (and not just correlated).

Minor:

8. On page 3, the authors cite some prior papers but should also cite the initial studies reporting IL1RAP overexpression in AML/MDS which represent the most frequent (~70% of AML patients) mechanism of activation of this pathway as well as its therapeutic targeting. (Barreyro L et al., Blood 2012; and Askmyr M et al., Blood 2013)
9. The last sentence in the abstract is incorrect and a dramatic overstatement. The study certainly *does not* suggest that enhanced stimulation of the IL1R pathway does not play a role. There is plenty of evidence in the literature (some of which the authors cite) supporting this fact. At best they show an additional contributing mechanism (decrease of repression). This has to be toned down / rephrased.

Reviewer #2 (Remarks to the Author):

Villatoro et al provide several lines of evidence to suggest that the endogenous IL-1RA protein plays an important role in HSC differentiation and progression to hematological malignancies. Their data suggest that reduction in IL-1RA expression levels correlate with a more severe disease state in human AML and that IL-1RA possesses some type of buffering capacity for IL-1beta-dependent pro-tumorigenic effects in the hematopoietic system and. While the importance of the IL-1 system for (auto) inflammatory diseases is undisputed, its precise role in diverse malignant diseases is still controversial with no therapeutic approach established. In this respect, the data are very interesting, advance our knowledge on a crucial pathophysiological role of IL-1RA and also point to potential therapeutic benefits using Canakinumab or Anakinra for certain forms of leukemia. The manuscript is well written and the data match the conclusions. A particular strength is the thorough and systematic characterization of blood and bone-marrow cell populations and the use of clinical and preclinical models. Limitations of this study include a lack of mechanistic data, in particular concerning the autoregulatory loop that balances IL-1beta versus IL-1RA in the absence of prototypical triggers of IL-1beta secretion. The data set could also be improved concerning the downstream effects observed in the IL-1RA knock-out animal models including the implication of the NF-kB system.

Specific points:

Fig. 1G: The authors measure phosphorylation of p65 NF-kB at Serine 529 as a readout for activation. This is not justified. Ser529 is a relatively poorly characterized P-site phosphorylated by CK2 whose role in IL-1 or TNF signaling is still unclear. In order to demonstrate that loss of IL-1RA indirectly activates the NF-kB system, degradation of Ikb α or nuclear translocation of p65 needs to be determined.

Fig.2F: The authors stain cells for the low abundant IL1R1 aiming to localize the IL-1b responsive cells that may mediate IL-1b effects in the bone-marrow of animals lacking IL-1RA. However, the IL-1 system is not strongly regulated at the level of the IL1R1. This receptor is expressed ubiquitously in most cell types to permit a fast and broad tissue response to newly synthesized IL-1a/b. To clarify the source of IL-1b and the responding cells in IL-1RA ko cells, it would be more appropriate to investigate the expression of IL-1b and IL-1RA in situ using single cells techniques such RNA-FISH or specific PLA. Such approaches would also clarify if IL-1RA buffers an autocrine or paracrine IL-1b-driven feedforward loop in the bone-marrow of healthy or diseased animals.

Fig.2O: The classification of NF-kB activity (on the Y-axis) of the graph as deduced from the RNA-seq data is unclear. The authors should provide much more detail on the bioinformatics analyses that made them to suggest that in animals lacking IL-1RA prototypical NF-kB target genes are upregulated. They should also provide according data, for example the presence of NF-kB binding sites within the enhancers or promoters of the top-ranked DEGs.

Other major points:

(1) IL-1RA inhibits the activities of IL-1a and IL-1b equally well. Constitutive functions of IL-1a have been described; in particular, the uncleaved IL-1a precursor can enter the nucleus and activate gene expression. The authors should discuss the possibility that IL-1a also contributes to the hematopoietic phenotypes of cells lacking IL-1RA. Have they tried anti IL-1a antibodies?

(2) A small number of children have been reported that carry LOF mutations of the IL1RA locus. These children develop severe auto-inflammatory disease, which is called DIRA syndrome. If DIRA patients also suffer from abnormal hematopoiesis, this would lend strong support to the main conclusions of this study. The authors should cover this point in their discussion.

Reviewer #3 (Remarks to the Author):

The studies by Villatoro and colleagues make important and novel contributions to our understanding of inflammatory (focusing on IL1) signaling and its roles in myeloid leukemogenesis. The studies are comprehensive and quite rigorous. Statistical analyses are solid, and the model systems are largely quite physiological and disease relevant.

They find lower expression of IL1RN (IL1 receptor antagonist) in more primitive AMLs, and lower IL1RN is clearly shown to correspond to worse outcomes for patients with AML. Using KO mice, they present convincing and novel data showing that endogenous IL-1RN represses myelopoiesis

in HSPC and enables B-cell development under steady-state conditions. They further show that IL1RN expression in both HSPC and in stromal cells limits oncogenic NRAS mediated myeloid pre-leukemic disease, using mixed chimeras with HSPC and BM stroma from mice of different genotypes. Their analyses of both HSPC autonomous and BM stromal roles for IL1RN is a particular strength of their studies. Importantly, they show that blocking IL1 (using anakinra or anti-IL1) can reduce NRAS mediated disease, and also alleviate the impacts of IL1RN KO on myelopoiesis (and also reduce the expansion of human AML cells in immunocompromised mice, consistent with earlier work).

There are a few issues, most relatively minor but a couple more significant, that should be addressed in order to realize the full impact of this work. Most of these issues could be addressed either in text changes or by new analyses of datasets (including of others).

Significant issues:

1) Materials and Methods are too cursory for some parts. We need to know the genetic background for mouse models (and how many backcrosses into a background). More information is needed for transplantations – for example, what dose and schedule of irradiation (X-ray? Gamma?) was used. Details for treatments with anakinra or antibody, as well as for BrdU incorporation in cells in mice, are also needed. Finally, I get that they calculated NFkB activity based on target levels in RNAseq data, but I didn't see details on how this was done (or a reference for it).

2) While they clearly show that low IL1RN is associated with worse survival from AML, since the primitive phenotype is associated with poor responses to chemo, and they show that more primitive AMLs express less IL1RN, the authors need to separately analyze more primitive and more differentiated (M4/5) leukemias for the predictive value of IL1RN. IL1RN expression may simply mark maturation state. Also, it is surprising that low IL1RN is associated with a more primitive state, as IL1 signaling (and genetic disruption of IL1RN as shown here) promotes myelopoiesis; this deserves some sort of comment.

3) They present comprehensive analyses of the IL1RN KO mice, showing promotion of myelopoiesis and clear changes in gene expression programs in HSPC (HSC and other more short-term progenitors). The relevance of these results would be expanded by comparisons to previous phenotypic and gene expression analyses of HSC and other HSPC in mice chronically treated with IL1b – how overlapping or non-overlapping are results? The studies that come to mind are from Pietras and colleagues (in addition to the cited paper, there are two recent reports in JEM with such analyses: PMID 33914855 and 33857288).

4) For the Mx1-Cre model of oncogenic NRAS activation, they need to acknowledge potential complications of studying inflammatory signaling in a model that requires an inflammatory signal (pI/C) to induce oncogene expression. Unless I'm not following the model, it also appears that NRAS will be oncogenically activated in the majority of HSPC plus many other cell types that would activate CRE with pI/pC. This is a complication, but this could be fixed by ensuring that key results with the mice with direct activation of oncogenic NRAS are reproduced in the experiments using chimeras created with the cells from these mice. Some of this is indeed demonstrated in Fig 5, particularly for biased myelopoiesis. I don't expect that they repeat the RNAseq analyses in the chimeric context, but if they could reproduce a couple of key results (like for NFkB target activation) in the chimeric context, it would provide validation for their non-chimeric results.

Minor:

1) They show that "The percentage of PB CD34+ progenitors activated through NFkB activation was higher in AML patients (Fig. 1G)." It's not really clear what this tells us, as CD34+ progenitors are a mixed population of multiple progenitors (including stem cells and more short-term progenitors). So if ~78% of these are positive for NFkB, what does that mean, particularly when comparing to AML?

2) Regarding "Analysis of IL-1rn-KO mice also revealed increased apoptosis in total BM cells (Supplementary Fig. S2F)", they can't conclude this, as the increased detection of apoptotic cells could be due to reduced phagocytosis.

3) For experiments using bortezomib, they conclude that biased myelopoiesis is "dependent on NFκB activation in HSPC"; given that proteasome inhibition will impact other targets, this conclusion needs to be tempered.

4) For Fig 4I, the apoptotic rates for stromal cells appear unrealistic, and likely reflect cell death that occurs during harvest of the MSC. These data doubtfully reflect apoptosis happening in vivo. There may instead be differences in cellular "fragility".

5) In the Discussion, they note that "Anakinra provides the advantage of potentially better regulated physiological responses." This is purely optional, but I would suggest that they may want to note that a concern with the Cantos trial with canakinumab was deaths due to infection/sepsis. The ability to quickly discontinue IL1 inhibition could be actually advantageous.

REVIEWER COMMENTS

Reviewer #1 (Remarks to the Author):

In this paper by Villatoro et al the authors study the endogenous IL-1 receptor antagonist IL1RA/IL1RN in healthy and malignant hematopoiesis. They show that lower IL1RN RNA expression levels correlate with worse prognosis in AML patients; and that in mice with deletion of IL1RN myelopoiesis is increased and correlates with corresponding molecular signatures, as well as defects in the bone marrow stroma are detectable. They also demonstrate lower IL1RN levels in an NRASG12D mutant mouse model, and that some of these effects are mediated by IL1RN-deficient stroma, which is one of the most novel aspects of the paper. While overall interesting and on an important topic, the paper suffers from several major weaknesses that would be critical to address to draw any meaningful conclusions.

We thank the Referee for the revision of our work, the positive comments and constructive criticism.

Major:

1. There are inconsistencies in the presentation of the survival curves (Figure 1). The effect seems modest and is borderline significant. Also, 2 contradictory significance values are provided (0.048 in the text; 0.035 in the figure) – which one is correct?

Both were correct and they were explained in the RNA sequencing and bioinformatic data analysis Section from Supplementary methods. The Cox proportional hazards regression analysis showing the contribution of *IL1RN* expression level to predict survival considered together with age and cytogenetic risk was indicated in the text. The survival plot showed the likelihood ratio test comparing two Cox models; this one considering *IL1RN* high/low expression, age and cytogenetic risk, and the other one considering age and cytogenetic risk only. This analysis answers the question of whether addition of *IL1RN* expression level improves the survival model using established factors known to influence AML patient survival. For clarity, we have kept the likelihood ratio test.

We have now also revisited and refined the survival analysis by removing the samples with unknown cytogenetic risk group and plotting the actual survival and censored data points, rather than the Cox model prediction results. We are also providing the results for the global AML patient dataset and the M4-M5 AML patient dataset (Fig. 1A).

2. The authors found particularly low IL1RN level in M0-M3 AML, which are the more immature/aggressive subtypes. This begs the question of what are the IL1RN levels in normal HSPC of the different lineages. Is IL1RN naturally low in more immature HSPC and therefore co-tracks with prognosis, without actually being downregulated if one compares with appropriate cell types ?

This comment is in line with the significant issue 2 from Reviewer 3. Our data show that low levels of expression of IL1RN is a common event in AML patients, but it is indeed particularly evident in more immature phenotypes. To answer the question of whether IL1RN may be lower in more immature HSPC and therefore associates with prognosis, we have now separately analyzed more differentiated AML for the predictive value of IL1RN and found that low IL1RN is a prognostic marker of reduced survival in M4-M5 AML patients. This is an indication that low IL1RN expression does not simply mark maturation state (Fig. 1A).

Our mouse data further argues against this point. qRT-PCR in WT mice showed expression of *Il1b* and *Il1rn* in LSK HSPC (Fig. 2K and 2M). Further analyses through RNA-FISH indicated that LT-HSC expressed the highest levels of *Il1b* among LSK subsets, followed by ST-HSC and MPP (Fig.

4F; Supplementary Fig. S5D). A fraction of the *Il1b*-expressing LT-HSC also had detectable levels of *Il1rn* expression (Fig. 4F; Supplementary Fig. S5D). These data were reproduced in *NRAS*^{G12D} controls with no Cre, whose LT-HSC compartment expressed the highest levels of both *Il1b* and *Il1rn* among LSK subsets (Fig. 5E).

3. This is a particularly relevant question as the data provided in Figure 1C is not at all convincing and indeed shows no relevant/significant difference (the case numbers are low; and looking at the data the effect is extremely modest, driven by only 2 events/outliers, and it is hard to reconcile how the p value of this can be significant – how was this calculated). Another question on this experiment: were G-CSF mobilized healthy PB CD34+ cells used? If not, how did the authors obtain enough cells to run these assays (CD34+ cells are extremely rare in healthy PB).

The difference is very significant in Figure 1C ($p < 0.001$) and AML CD34+ cells show surprisingly low values that are homogeneous across AML patients. The rare events are those that show more similar values to the values in healthy donors. The numbers used were 32 controls and 21 AML + 2 MDS patients, and differences were studied with unpaired two-tailed t-test. Following the editor recommendation, since both groups have more than 10 values, we have changed the format of the figure to a violin plot. We hope this will facilitate the understanding of the figure.

We did not use G-CSF or any other mobilizing agent. Indeed, numbers of PB CD34+ cells are low in healthy donors. We used Dynabeads™ mRNA Purification Kit, which allows downstream study of the expression of several genes in less than 5 000 cells.

4. In the xenograft experiments, how do the authors distinguish engrafted AML cells from normal HSPC (which are always co-transplanted). Is it possible that *Il1rn* just reduces myelopoiesis in general, irrespective of healthy/AML; i.e. is there any specificity to AML cells?

We did not distinguish engrafted AML from normal HSPCs in the xenograft experiments. This is a common procedure in the field as the main output is expected from the AML cells when using donor CD34+ cells isolated from the BM of AML patients at diagnosis with overt AML.

As outlined throughout the manuscript, our experiments in mice demonstrate a critical role for balanced *Il-1rn* on steady-state myelopoiesis, with its deletion resulting in expanded myelopoiesis that becomes apparent in the absence of injury or infection. Thus, repression by *Il-1rn* is required for healthy and balanced myelopoiesis. Low *Il-1rn* is present in biased pre-leukemic myelopoiesis and *Il-1rn* genetic deletion further promotes myeloproliferation. Conversely, treatment with exogenous *Il-1rn* reverts pre-leukemic myeloproliferation. Thus, loss of repression by *Il-1rn* contributes to pre-leukemic myelopoiesis and its administration rescues balanced myelopoiesis despite presence of a driver mutation. Taken together with our data on AML patients and xenografts, we can suggest that biased myelopoiesis, either in presence or absence of malignancy, could potentially be rebalanced to normal levels with *Il-1RN*.

5. In all human studies (and also the mouse studies in Figure 2), did the authors also look at *IL1RAP* levels on their cells? This seems to be essential for 2 reasons: (1) *IL1RAP* has been reported as a predominant mechanism of *Il1R* pathway activation in AML (70% of patients), and (2) *Il1rn* acts via disrupting *IL1R1-IL1RAP* interaction which is essential for signaling. Without that information it is hard to truly interpret any of the data presented.

Thanks for this suggestion to complement our data. We have now confirmed higher gene expression of *IL1RAP* and higher frequency of CD34+ progenitors expressing membrane *IL-1RAP* in AML patients compared to healthy donors (Supplementary Fig. S1E and S1F), as previously described. In the *Il-1rn*-KO mouse model, *Il1rap* gene expression is mostly unchanged in myeloid cells and HSPC subsets, with exception of a modest reduction in MPP, versus wild-type mice (Fig. 2K-L). Of note, MPP are the LSK subset more resistant to *Il1rn* deletion. We have also studied the expression of *Il1rap* in the *Mx1-Cre*⁺ *NRAS*^{G12D} mouse model and found unchanged gene expression in myeloid cells and HSPC subsets isolated from pre-leukemic mice compared to controls (Fig. 5D-E). Given that there is no mouse monoclonal antibody commercially available, we

have generated a monoclonal antibody to measure IL-1RAP protein expression in this model. Consistent with the findings in human AML, the percentage of LT-HSC, MPP and monocytes expressing IL-1RAP was higher in preleukemic mice compared to controls (Fig. 5F; Supplementary Fig. S6H). This suggests that IL-1RAP may be regulated through translational and post-translational mechanisms. We agree that the overview of the pathway is now more complete with the additional IL-1RAP studies.

6. It is unclear to this reviewer why the authors analyzed transcriptional programs in the NRAS-G12D model. Just because these mice have (modestly) reduced IL1RN levels (amongst many other alterations) does not mean the measured GE profiles tell us anything about what IL1RN does. Similarly, what was the rationale for the bortezomib treatment? This is a broad spectrum proteasome inhibitor with very pleiotropic effects. This also seems to be completely unrelated to any studies related to IL1RN.

RAS transduces IL-1 β signaling through MyD88, and oncogenic RAS results in IL-1 β amplification through persistent activation of the autocrine feedback loop in other types of cancer. As explained in the Introduction, IL-1RN production follows after IL-1 by roughly the same cell types of hematopoietic and non-hematopoietic origin. However, the levels of IL-1rn in the NRAS-G12D mouse model are reduced. The rationale behind studying the transcriptional programs in these mice was to check NF κ B activation, main pathway responsible for abnormal myelopoiesis in IL-1rn-KO mice. By doing so, we confirmed activation of NF κ B and low levels of IL-1rn together with high IL-1 β in NRAS-G12D mice. This encouraged us to test the functional relevance of both NF κ B (bortezomib treatment; Fig. 5H) and low IL-1rn (Fig. 6).

We agree that proteasome inhibition will impact additional targets to NF κ B, so this conclusion has been tempered (requested too by Reviewer 3 in Minor comment 3). The conclusion related to bortezomib experiment reads now: "the biased myelopoiesis in NRAS-G12D+ mice was at least partially dependent on NF κ B activation in HSPC".

7. What appears to be a critical experiment that is missing in this context would be treatment with either IL1RN (to restore normal levels) or canakinumab to demonstrate that this pathway is indeed functionally relevant in the context of NRASG12D (and not just correlated).

We provided these data in figure 6, where we show that deletion of IL-1rn from the hematopoietic or stromal compartments promotes pre-leukemic myelopoiesis, whereas exogenous IL-1rn (anakinra) protects against it.

Minor:

8. On page 3, the authors cite some prior papers but should also cite the initial studies reporting IL1RAP overexpression in AML/MDS which represent the most frequent (~70% of AML patients) mechanism of activation of this pathway as well as its therapeutic targeting. (Barreyro L et al., Blood 2012; and Askmyr M et al., Blood 2013)

These references have been added. Thank you for pointing them out.

9. The last sentence in the abstract is incorrect and a dramatic overstatement. The study certainly *does not* suggest that enhanced stimulation of the IL1R pathway does not play a role. There is plenty of evidence in the literature (some of which the authors cite) supporting this fact. At best they show an additional contributing mechanism (decrease of repression). This has to be toned down / rephrased.

The sentence now reads: "Our data support that HSC differentiation is controlled by balanced IL-1 β /IL-1rn levels under steady-state, and suggest that loss of repression resulting in enhanced stimulation of IL-1 β signaling may underlie pre-leukemic lesion and AML progression".

Reviewer #2 (Remarks to the Author):

Villatoro et al provide several lines of evidence to suggest that the endogenous IL-1RA protein plays an important role in HSC differentiation and progression to hematological malignancies. Their data suggest that reduction in IL-1RA expression levels correlate with a more severe disease state in human AML and that IL-1RA possesses some type of buffering capacity for IL-1beta-dependent pro-tumorigenic effects in the hematopoietic system and. While the importance of the IL-1 system for (auto) inflammatory diseases is undisputed, its precise role in diverse malignant diseases is still controversial with no therapeutic approach established. In this respect, the data are very interesting, advance our knowledge on a crucial pathophysiological role of IL-1RA and also point to potential therapeutic benefits using Canakinumab or Anakinra for certain forms of leukemia. The manuscript is well written and the data match the conclusions. A particular strength is the thorough and systematic characterization of blood and bone-marrow cell populations and the use of clinical and preclinical models. Limitations of this study include a lack of mechanistic data, in particular concerning the autoregulatory loop that balances IL-1beta versus IL-1RA in the absence of prototypical triggers of IL-1beta secretion. The data set could also be improved concerning the downstream effects observed in the IL-1RA knock-out animal models including the implication of the NF-kB system.

We thank the Referee for the revision of our work, the positive comments and constructive criticism.

Specific points:

Fig.1G: The authors measure phosphorylation of p65 NF-kB at Serine 529 as a readout for activation. This is not justified. Ser529 is a relatively poorly characterized P-site phosphorylated by CK2 whose role in IL-1 or TNF signaling is still unclear. In order to demonstrate that loss of IL-1RA indirectly activates the NF-kB system, degradation of Ikb α or nuclear translocation of p65 needs to be determined.

Following the Referee recommendation, we have studied the nuclear translocation of the p50/p65 NFkB heterodimer with Amnis Imaging Flow Cytometer. The results have confirmed increased NF-kB translocation in CD34+ progenitors from AML patients versus healthy donors (Figure 1G and Supplementary Fig. S1H; phospho-NFkB in new Figure S1G).

Fig.2F: The authors stain cells for the low abundant IL1R1 aiming to localize the IL-1b responsive cells that may mediate IL-1b effects in the bone-marrow of animals lacking IL-1RA. However, the IL-1 system is not strongly regulated at the level of the IL1R1. This receptor is expressed ubiquitously in most cell types to permit a fast and broad tissue response to newly synthesized IL-1a/b. To clarify the source of IL-1b and the responding cells in IL-RA ko cells, it would be more appropriate to investigate the expression of IL-1b and IL-1RA in situ using single cells techniques such RNA-FISH or specific PLA. Such approaches would also clarify if IL-1RA buffers an autocrine or paracrine IL-1b-driven feedforward loop in the bone-marrow of healthy or diseased animals.

Our data indicate that there is a certain degree of regulation of the pathway at the level of *Il1r1* expression as shown by higher expression in LT-HSC from IL-1rn-KO versus WT mice in bulk RNA-Seq and qRT-PCR (Fig 2K, 2O). *Il1r1* is expressed 1000-fold higher in WT LSK cells than in WT myeloid cells, further illustrating heterogeneity in expression levels among cell types (Fig. 2K, 2L). We were also able to detect higher frequency of cells (Fig. 2F), and specifically CD63⁺ stromal cells (Fig. 3C), expressing membrane IL-1r1 in the BM of IL-1rn-KO mice versus WT.

Compared to WT HSPC, mRNA levels of *Il1rn* were 170- and 20-fold higher in WT granulocytes and monocytes, respectively (Fig. 2M). BM CD63⁺ MSC produce the highest levels of *Il1rn* in the BM of adult WT mice (Fig. 3G); 2.8-fold higher than WT granulocytes (Fig. 2M). Regarding *Il1b*, the highest levels of expression were shown by granulocytes, followed by CD63⁺ stromal cells and monocytes, with LSK subsets expressing low levels 570-fold lower than granulocytes (Fig. 2K, 2L and 3G). Unlike myeloid cells, LT-HSC and ST-HSC showed sustained engagement of an IL-1 β -

positive feedback loop through increased *Il1b* expression in IL-1rn-KO versus WT mice (Fig. 2K and 2L). Of note, *Il1rap* gene expression was unchanged in LT-HSC and ST-HSC but reduced in MPP from IL-1rn-KO, which could contribute to the limited effects of IL-1rn deletion on MPP (Fig. 2K). These data suggested that LT-HSC and ST-HSC play a key role in the chronic inflammatory process driven by IL-1rn deletion through IL-1 β production and sustained supply of effector inflammatory cells. Our qRT-PCR results from BM CD63+ MSC also indicated that these cells contribute to inflammation with higher *Il1b* expression in the absence of *Il1rn* (Fig. 3G).

Following the Referee recommendation, we have studied the expression of *Il1b* and *Il1rn* by single-cell techniques, including both scRNA-Seq and RNA-FISH in FACS-sorted cells (Fig. 4). We decided to use FACS-sorted cells given that in situ methods would not allow to dissect cell compartments of interest simultaneously to their expression of *Il1b* and *Il1rn*, particularly considering the numbers of markers required for HSC identification. Although this way we lose the information of local interactions, FACS-sorting allows for better definition of the cell sources of *Il1b* and *Il1rn*. We also combine two techniques for better accuracy, meaning scRNA-Seq which allows quantification but depicts only highest expressing cells in a minimum percentage of 10% of the cells, with imaging as suggested by the Referee by highly sensitive RNA-FISH that in turn does not allow quantification. In agreement with the data in bulk, at the single-cell level, we found that neutrophils were the most abundant CD11b+ cell subset expressing *Il1b* and *Il1rn*, and expressed the highest levels of both cytokines, followed by monocytes (Fig. 4A-C). Cytokine-expressing neutrophils and monocytes express mainly *Il1b* or *Il1rn* as evidenced by both scRNA-Seq (Fig. 4C) and RNA-FISH (Fig. 4D; Supplementary Fig. S5A), with a small fraction of neutrophils expressing both cytokines (Fig. 4C-D; Supplementary Fig. S5A). Expression of *Il1b* and *Il1rn* was not detectable by scRNA-Seq in LSK cells (Supplementary Fig. S5C), but RNA-FISH showed that LT-HSC expressed the highest levels of *Il1b* among LSK subsets, followed by ST-HSC and MPP (Fig. 4F; Supplementary Fig. S5D). Some of the *Il1b*-expressing LT-HSC also had detectable levels of *Il1rn* expression (Fig. 4F; Supplementary Fig. S5D). We found cells expressing *Il1b*, *Il1rn* or both within the CD63+ stromal cell compartment both by scRNA-Seq (Fig. 4G-I) and RNA-FISH (Fig. 4J; Supplementary Fig. S5E). scRNA-Seq was able to resolve that *Il1b/Il1rn* are mainly produced by MSC and progenitors within the CD63+ stromal cell compartment, and most cytokine-expressing MSC express *Il1rn* only whereas progenitors express mainly *Il1b* or both, with a small fraction expressing *Il1rn* only (Fig. 4I). Together, these data highlight that IL-1rn buffers mostly paracrine IL-1 β feed-forward loops through its exclusive expression from subsets of CD63+ MSC, neutrophils and monocytes, and a small fraction of CD63+ progenitors in the BM of WT healthy mice under steady-state conditions. Autocrine responses are rare and were observed mainly in subsets LT-HSC and CD63+ progenitors, and a small fraction of neutrophils, with expression of both cytokines simultaneously.

In the new light of the scRNA-Seq, we are now also able to suggest that IL-1 β and IL-1rn control homeostasis and inflammatory responses in primitive HSC and MPP5, expand myeloid output through the control of transcriptomic pathways in MPP2-MPP3 and repress lymphoid output on the cluster MPP1 leading to unidentifiable prototypical lymphoid-biased MPP4 transcriptional program (Fig. 4E; Supplementary Fig. 5B).

The following secure token has been created to allow review of record GSE197594 while it remains in private status: klslememzvrhwh.

Fig.2O: The classification of NF-kB activity (on the Y-axis) of the graph as deduced from the RNA-seq data is unclear. The authors should provide much more detail on the bioinformatics analyses that made them to suggest that in animals lacking IL-RA prototypical NF-kB target genes are upregulated. They should also provide according data, for example the presence of NF-kB binding sites within the enhancers or promoters of the top-ranked DEGs.

The method was included in a previous document named "Supplementary methods", but it is now in the main manuscript and expanded as requested. The method for quantifying activity levels of NFkB targets in LT-HSC, ST-HSC and MPP from accessions GSE126428 and GSE126625 and LK

from publicly available accession PRJNA774277, was adapted from pathway level analysis of gene expression (Tomfohr et al, 2005). The method performs PCA on the subset of the gene expression matrix that only includes those genes that are members of the gene set of interest (i.e. NFκB target genes from Synapse ID syn4956655, Marbach et al, 2016). The sample scores on the first principal component capture the major pattern of variation for the gene set and are used as an estimate of overall gene set activity. Three modifications were made to the original microarray-based method to adapt it for RNA-Seq data. First, due to the high number of genes with very low counts present in cell lines, the step in the algorithm where gene expression is normalized by variance was removed. This reduces the risk of low count genes being overestimated by the normalization. Second, as activity levels are compared between different datasets, it is necessary to correct for the arbitrary choice of positive direction of PCA. As such, a check of the PCA loadings between datasets was included and the positive direction chosen to obtain the maximum consistency between the PCA loadings, or rotation matrixes, between datasets. Third, to simplify interpretation and plotting, the scale was adjusted by subtracting the minimum activity score for a dataset to obtain only positive activity values. Stem cell relevant NFκB target genes were taken from the human cd34+_stem_cells_-_adult_bone_marrow_derived network found in Synapse ID syn4956655 (Marbach et al, 2016). These targets were determined by integrating transcription factor sequence motifs with promoter and enhancer activity data from the FANTOM5 project that uses cap analysis of gene expression (CAGE-analysis) (Consortium et al, 2014). Human target genes were converted to the mouse genome using biomaRt (Durinck et al, 2016) before their use as a gene set to calculate NFκB activities. Per sample transcription factor activities between groups were compared using a standard t-test. This NFκB gene list (n=1832) is detailed in Supplementary Table S4.

We are also providing new volcano plots for LT-HSC, ST-HSC, MPP from GSE126428 (Fig. 2O) showing genes with an absolute $-0.5 > \log_2 FC > 0.5$, where those genes in red have an adjusted $p < 0.05$. Differentially expressed genes in blue are identified as NFκB targets from three different databases; cd34+_stem_cells_-_adult_bone_marrow transcription factor network from Synapse ID syn4956655 (Marbach et al, 2016), <https://bioinfo.lifl.fr/NF-KB/> (Dr. K. Gosselin, Institute of Biology of Lille, France) and <https://www.bu.edu/nf-kb/gene-resources/target-genes/> (Dr. T. Gilmore, University of Boston, USA). NFκB target genes found in <https://www.bu.edu/nf-kb/gene-resources/target-genes/> and <https://bioinfo.lifl.fr/NF-KB/> contain mainly target genes in human with their corresponding RefSeq accessions and link to the publications where the association to NFκB was established. These compilations mainly include genes with checked binding, i.e. strong experimental evidence of NFκB direct control, such as chromatin immunoprecipitation or promoter transactivation, but also few putative genes, i.e. the gene has a κB site in the promoter, but has not clearly been shown to be controlled by NFκB, or the gene expression is associated with increased NFκB activity, but has not been shown to be a target directly. They also contain checked and putative NFκB target genes in mice, which we also used here. Human target genes were converted to their mouse orthologs using the one2one association from Ensemble Compara Database. The complete NFκB gene list (n=2132) is detailed in Supplementary Table S4. The plots depict the top 15 differentially expressed NFκB target genes, NFκB target genes of interest, and genes from those lists shared by another subset, and illustrate the enrichment of NFκB target genes among the differentially expressed genes in LT-HSC, ST-HSC, MPP from the BM of IL-1rn-KO versus WT mice.

Other major points:

(1) IL-1RA inhibits the activities of IL-1α and IL-1β equally well. Constitutive functions of IL-1α have been described; in particular, the uncleaved IL-1α precursor can enter the nucleus and activate gene expression. The authors should discuss the possibility that IL-1α also contributes to the hematopoietic phenotypes of cells lacking IL-1RA. Have they tried anti IL-1α antibodies?

Thanks for this comment. The rescue of the phenotype of IL-1rn-KO mice by in vivo treatment with a monoclonal antibody against IL-1β was virtually complete, suggesting that a role for IL-1α was unlikely. However, we agree with the Referee that it cannot be ruled out. Driven by the Referee

question, we are now including new data, showing no effect on the biased myelopoiesis of the treatment with a monoclonal antibody against IL-1a in IL-1rn-KO mice (Supplementary Fig. S2L).

(2) A small number of children have been reported that carry LOF mutations of the IL1RA locus. These children develop severe auto-inflammatory disease, which is called DIRA syndrome. If DIRA patients also suffer from abnormal hematopoiesis, this would lend strong support to the main conclusions of this study. The authors should cover this point in their discussion.

Thanks for bringing up this interesting topic. DIRA is a rare congenital disease described in 2009 that causes death from severe autoinflammation and multiorgan failure if not treated with anakinra. Hematopoietic symptoms in these patients include leukocytosis, extramedullary hematopoiesis in liver and spleen, bone marrow fibrosis and abnormal ossifications, reminiscent of the phenotype observed in the IL-1rn-KO mouse model and of early hematopoietic disease. We have discussed this point in the new version of our manuscript.

Reviewer #3 (Remarks to the Author):

The studies by Villatoro and colleagues make important and novel contributions to our understanding of inflammatory (focusing on IL1) signaling and its roles in myeloid leukemogenesis. The studies are comprehensive and quite rigorous. Statistical analyses are solid, and the model systems are largely quite physiological and disease relevant.

They find lower expression of IL1RN (IL1 receptor antagonist) in more primitive AMLs, and lower IL1RN is clearly shown to correspond to worse outcomes for patients with AML. Using KO mice, they present convincing and novel data showing that endogenous IL-1RN represses myelopoiesis in HSPC and enables B-cell development under steady-state conditions. They further show that IL1RN expression in both HSPC and in stromal cells limits oncogenic NRAS mediated myeloid pre-leukemic disease, using mixed chimeras with HSPC and BM stroma from mice of different genotypes. Their analyses of both HSPC autonomous and BM stromal roles for IL1RN is a particular strength of their studies. Importantly, they show that blocking IL1 (using anakinra or anti-IL1) can reduce NRAS mediated disease, and also alleviate the impacts of IL1RN KO on myelopoiesis (and also reduce the expansion of human AML cells in immunocompromised mice, consistent with earlier work).

There are a few issues, most relatively minor but a couple more significant, that should be addressed in order to realize the full impact of this work. Most of these issues could be addressed either in text changes or by new analyses of datasets (including of others).

We thank the Referee for the revision of our work, the positive comments and constructive criticism.

Significant issues:

1) Materials and Methods are too cursory for some parts. We need to know the genetic background for mouse models (and how many backcrosses into a background). More information is needed for transplantations – for example, what dose and schedule of irradiation (X-ray? Gamma?) was used. Details for treatments with anakinra or antibody, as well as for BrdU incorporation in cells in mice, are also needed. Finally, I get that they calculated NFkB activity based on target levels in RNAseq data, but I didn't see details on how this was done (or a reference for it).

All this information was in a previous document named "Supplementary methods". The complete set of methods with full details has now been incorporated in the main manuscript. For clarity, we have highlighted in blue only the information asked here as well as the new methodology. We apologize for the confusion.

2) While they clearly show that low IL1RN is associated with worse survival from AML, since the primitive phenotype is associated with poor responses to chemo, and they show that more primitive AMLs express less IL1RN, the authors need to separately analyze more primitive and more differentiated (M4/5) leukemias for the predictive value of IL1RN. IL1RN expression may simply mark maturation state. Also, it is surprising that low IL1RN is associated with a more primitive state, as IL1 signaling (and genetic disruption of IL1RN as shown here) promotes myelopoiesis; this deserves some sort of comment.

Our data show that low levels of expression of IL1RN is a common event in AML patients, but it is indeed particularly evident for more primitive phenotypes. We have now separately analyzed more differentiated AML for the predictive value of IL1RN and found that low IL1RN is a prognostic marker of reduced survival in M4-M5 AML patients (Fig. 1A). This indicates that low IL1RN expression does not simply mark maturation state. However, there are so few M0-M3 patients with high IL1RN expression (n = 7 / 217) that makes the analysis of the prognostic value of IL1RN not feasible in this group of patients. Please note that we have now also revisited and refined the survival analysis by removing the samples with unknown cytogenetic risk group and plotting the actual survival and censored data points, rather than the Cox model prediction results.

When it comes to the second part of this point, those are interesting and seemingly paradoxical observations. Together, loss of repression resulting in enhanced stimulation of IL-1 β signaling may underlie both pre-leukemic lesion and AML progression. Low IL-1rn and subsequent high IL-1b promotes myeloproliferation and inhibits the most primitive HSC, with the contribution of HSC to the low IL-1rn and high IL-1b. These conditions may pose a favorable or pressuring environment for the long-term selection of clones with lighter sensitivity to the inhibitory effect of low IL-1rn and high IL-1b. In theory, the potential benefit of this process would be to prevent the exhaustion of the HSC pool. However, HSC clones that thrive and expand in the presence of an otherwise inhibitory signal meet the criteria of functionally malignant leukemia stem cells, and selection of these clones would result in aggressive malignancy. Further work is required to validate this intriguing hypothesis, but if it holds true it could explain the association of low IL-1RN with both myeloproliferation at early or mild stages of disease and with a more primitive state and poor survival in AML patients who are experiencing late or aggressive transformation events. It would also provide grounds to the therapeutic potential of IL-1RN at both stages. We have included these ideas in the Discussion.

3) They present comprehensive analyses of the IL1RN KO mice, showing promotion of myelopoiesis and clear changes in gene expression programs in HSPC (HSC and other more short-term progenitors). The relevance of these results would be expanded by comparisons to previous phenotypic and gene expression analyses of HSC and other HSPC in mice chronically treated with IL1b – how overlapping or non-overlapping are results? The studies that come to mind are from Pietras and colleagues (in addition to the cited paper, there are two recent reports in JEM with such analyses: PMID 33914855 and 33857288).

Myeloid differentiation bias at the expense of B cell development, reduced self-renewal of primitive HSC and expansion of committed MPP seem consistent hematopoietic effects of chronic (20 day) high dose (0.5 μ g) IL-1 β treatment across studies despite variability particularly under experimental conditions of transplantation (Pietras et al, 2016; Chavez et al, 2021; Higa et al, 2021), and in vivo deletion of IL-1rn phenocopies these results under steady-state. IL-1 β -induced hematopoietic effects are dependent on transcriptional programs activated by *Spi1* and *Cebpa* (Pietras et al, 2016; Chavez et al, 2021; Higa et al, 2021). We compared our RNA-Seq datasets with these previous publicly available datasets studying the transcriptional programs activated by IL-1 β treatment in C57BL/6 WT mice using LSK Flt3⁻CD48⁻CD150⁺ HSC in a native microenvironment (GSE165810; Chavez et al, 2021) or MPP3, expressing or not YFP, in the context of busulfan conditioning and transplantation (GSE166629; Higa et al, 2021). There is no RNA-Seq data in Pietras et al (2016). Those datasets were reanalyzed using the same methodology used for our RNA-Seq datasets for consistency. We found a partial overlap that was cell type-specific and particularly remarkable for the transcriptional activation of genes induced by IL-1rn deletion in LT-

HSC and ST-HSC, and by IL-1 β treatment in LSK Flt3⁻CD48⁻CD150⁺ HSC (New Supplementary Fig. S3I and S3J). Of note, the upregulated genes common to all cell subsets and experimental conditions were enriched in NF κ B targets, including *Cebpd*, *Csf2rb*, *Csf2rb2* or *Spi1* (Supplementary Fig. S3I and S3J).

4) For the Mx1-Cre model of oncogenic NRAS activation, they need to acknowledge potential complications of studying inflammatory signaling in a model that requires an inflammatory signal (pl/C) to induce oncogene expression. Unless I'm not following the model, it also appears that NRAS will be oncogenically activated in the majority of HSPC plus many other cell types that would activate CRE with pl/pC. This is a complication, but this could be fixed by ensuring that key results with the mice with direct activation of oncogenic NRAS are reproduced in the experiments using chimeras created with the cells from these mice. Some of this is indeed demonstrated in Fig 5, particularly for biased myelopoiesis. I don't expect that they repeat the RNAseq analyses in the chimeric context, but if they could reproduce a couple of key results (like for NF κ B target activation) in the chimeric context, it would provide validation for their non-chimeric results.

Thanks for this comment. In the previous version of our manuscript, we used the *Mx1-Cre NRAS^{G12D}* mouse model of pre-leukemic myelopoiesis in all our experiments. This model is inducible allowing fine-tuned controlled levels of Cre expressed during adulthood, as opposed to constitutive conditional Cre where the resulting phenotype is a cumulative result for all cells expressing Cre at any time, including potential confounding effects originated during development and younger ages. One of the main drawbacks of *Mx1-Cre NRAS^{G12D}* is indeed that the model is inducible with an inflammatory signal that may have effects on HSCs. However, bone marrow isolated from mice treated three times with similar doses of polyI:polyC (followed by 10 days recovery) was serially transplanted into irradiated recipients and no significant difference in the repopulation activity of polyI:polyC treated cells was observed (Essers et al, 2009). The authors concluded that this transient activation of IFN α signaling does not affect the number of functional HSC, as opposed to chronic activation of this pathway. Our mice were monitored for long periods of time and the controls used were *NRAS^{G12D}* with no Cre treated with polyI:polyC, to reduce the potential confounding effects derived from polyI:polyC-induced inflammation.

Another advantage of *Mx1-Cre NRAS^{G12D}* is that it is conditional and thus allows restriction of Cre expression, mainly to hematopoietic cells but also osteolineage cells (Park et al, 2012) and likely other cells in additional organs to bone marrow. To rule out this challenge, we used the chimera systems that allowed restriction of Cre within the hematopoietic system only. For these experiments, donors were at least 4 weeks after polyI:polyC induction and *Mx1-Cre NRAS^{G12D}* donor mice displayed splenomegaly. The potential complications derived from polyI:polyC-induced inflammation are not overcome with this approach.

To give answer to the Referee request, we are now providing complementary data on a constitutive and conditional mouse model restricted to the hematopoietic system, i.e. *Vav-Cre NRAS^{G12D}* (Carr et al, 2021; You et al, 2022). This mouse model reproduces the main hematopoietic abnormalities of *Mx1-Cre NRAS^{G12D}* including increased circulating white blood cells particularly monocytes, hypercellularity in the bone marrow, and higher numbers of immunophenotypically defined LSK HSPC, among others (Supplementary Fig. S7A-F). In addition, using publicly available gene expression profiling by RNA-Seq of lin-c-kit⁺ progenitors (You et al, 2022), we found that NF κ B transcription factor calculated activity was higher in *Vav-Cre NRAS^{G12D}* than in control mice (Supplementary Fig. S7G).

Minor:

1) They show that "The percentage of PB CD34⁺ progenitors activated through NF κ B activation was higher in AML patients (Fig. 1G)." It's not really clear what this tells us, as CD34⁺ progenitors are a mixed population of multiple progenitors (including stem cells and more short-term progenitors). So if ~78% of these are positive for NF κ B, what does that mean, particularly when comparing to AML?

Thanks for this question. We have included some lines to discuss on this aspect: “CD34+ cells are a mixture of LT-HSC, ST-HSC and progenitors, so these changes may reflect the differential cell composition and/or selective changes per cell subset, between health and AML. These data are indicative of the enrichment of this pathway in AML CD34+ cells and consistent with previous observations of NFκB activation in human CD34+CD38– leukemia stem cells”.

2) Regarding “Analysis of IL-1rn-KO mice also revealed increased apoptosis in total BM cells (Supplementary Fig. S2F)”, they can't conclude this, as the increased detection of apoptotic cells could be due to reduced phagocytosis.

Indeed, the increased numbers of apoptotic cells in total BM could result from different processes, and not only from increased apoptosis. We have changed the sentence for clarity and now state “Analysis of IL-1rn-KO mice also revealed increased numbers of apoptotic cells in total BM”.

3) For experiments using bortezomib, they conclude that biased myelopoiesis is “dependent on NFκB activation in HSPC”; given that proteasome inhibition will impact other targets, this conclusion needs to be tempered.

We agree, thanks for pointing that out. We have replaced “dependent” by “at least partially dependent”.

4) For Fig 4I, the apoptotic rates for stromal cells appear unrealistic, and likely reflect cell death that occurs during harvest of the MSC. These data doubtfully reflect apoptosis happening in vivo. There may instead be differences in cellular “fragility”.

We agree that the apoptosis seems surprisingly high, and it is consistent with the reduced numbers of MSC harvested and observed by flow cytometry. Given the processing ex vivo, it cannot be guaranteed that these values faithfully reflect the in vivo status. We have changed the sentence in the Discussion to: “Our data showed that BM CD63+ MSC contributed to the lower levels of IL-1rn in the BM of NRAS-G12D+ mice by their cellular fragility, as demonstrated by both their increased apoptotic rates and reduced cell numbers, and selective downregulated expression of *Il1rn*”.

5) In the Discussion, they note that “Anakinra provides the advantage of potentially better regulated physiological responses.” This is purely optional, but I would suggest that they may want to note that a concern with the Cantos trial with canakinumab was deaths due to infection/sepsis. The ability to quickly discontinue IL1 inhibition could be actually advantageous.

We thank the Reviewer for this note that has been included in the Discussion for a better balanced comparison between anakinra and canakinumab.

REVIEWER COMMENTS

Reviewer #1 (Remarks to the Author):

Reviewer #2 (Remarks to the Author):

The authors have sufficiently addressed all my previous concerns. The extensive revisions both experimentally and in writing, as well as the new scRNA-seq and RNA-FISH data shown in Fig. 4 strengthen the conclusion of a prevailing IL1b / IL1RN paracrine feedback loop controlling homeostatic and premalignant processes in the bone marrow. The identification of the important roles of IL1RN in restricting IL1b functions in the hematopoietic system advance our knowledge on the constitutive functions of the IL1 system and open new therapeutic avenues for treating hematopoietic malignancies. I congratulate the authors to this very interesting study.

Reviewer #3 (Remarks to the Author):

The authors have done an excellent job addressing prior concerns. Most notably, the addition of new scRNAseq data (revealing the sources of IL1rn and the consequences of its loss), new analyses of their RNAseq data, and addition of analyses of additional NRAS initiated leukemogenesis mouse models, greatly strengthen and expand the impact. This manuscript makes important contributions in showing how IL1rn restricts normal myelopoiesis and leukemogenesis.

Reviewer #4 (Remarks to the Author):

If the authors want to look at what is different then they need to be looking at the differences within the clusters of the two mice? So what is different between for example between neutrophils in Wt vs IL-1rn KO mice?

Figure 4A: CD11b should not be expressed on mouse hematopoietic stem cells, so I am unsure as to why such a large percentage of the UMAP in Figure 4A would be labelled as stem cells. This is more of a cell identification issue, rather than a data/overall analysis issue.

Annotation of the cell types of all the scRNA-seq analysis needs improvement.

Figure 4A: Within the cells labelled as "neutrophil" and "monocyte" there appears to be an effect of the IL-1rn KO as the cells are not integrated but rather are next to each other on the UMAP, is this a batch effect in the data integration?

It is not clear from the text that the authors have used the data to be able to identify the clusters, it rather sounds as though they are re-using the data from the referenced article.

There are several issues with the label transfer used to be able to identify the clusters within the data.

In addition to the IPA, it would be advantageous to know which specific genes are actually included in this analysis, which genes are responsible for giving these enriched or reduced signatures?

Again in Figure 4E, mouse models do not overlap, if the authors want to show a difference in composition of the MPP cells, why did they not do this with FACS?

Is the UMAP shown in Figure 4 E including both the WT and KO mouse? If this is the case that the cellular populations do not overlap and there is something incorrect with the label transfer, batch correction or analysis.

If the IL-1rn KO mouse has an increase in myeloid biased MPP2-MPP3 cells, can this be seen downstream, for example were there more CD11b+ cells in the KO mouse?

The scRNA-seq analysis of LSK cells is based on n=1

The UMAPs in 4E and Supplementary Figure 5c are not the same.

In addition to the pie charts for the expression of Il1rn and Il1b, would actual plots of the expression be more advantageous to the reader as this would give a feel for the levels of expression, then it would be possible to also see the levels of expression of both factors, (IL-1b on y-axis and Il-1rn on x-axis).

The authors state "At single-cell resolution, these results suggest that IL-1 β and IL-1rn control homeostasis and inflammatory responses in primitive HSC and MPP5, expand myeloid output through the control of transcriptomic pathways in MPP2-MPP3 and repress lymphoid output on the

cluster MPP1 leading to unidentifiable prototypical MPP4 transcriptional program.” But in the introduction to the article they claim that there are no differences in the blood system and only observed changes under specific conditions. “IL-1 receptor 1 knockout (IL-1r1-KO) mice have unaffected blood production and normal stem and progenitor bone marrow (BM) compartments, suggesting that IL-1 β -induced myeloid priming of HSC occurs under conditions of injury or infection only and tonic IL-1 signaling has none or small basal hematopoietic effects”. The claims in line 295 – 298 are complete speculation, the authors show no evidence for these claims.

Figure 4G: The authors label “sinusoidal cells”, but there are many clusters of cells which are spread across the UMAP, it would be better to distinguish these different sub-populations rather than labelling them all the same, as the different clusters change between the WT and KO mice. The authors do not mention in the materials and methods the numbers of mice used for each experiment. In the figure legends it appears that the scRNA-seq was performed on n=1, this is not sufficient, there could be animal to animal variation.

REVIEWER COMMENTS-

Reviewer #2 (Remarks to the Author):

The authors have sufficiently addressed all my previous concerns. The extensive revisions both experimentally and in writing, as well as the new scRNA-seq and RNA-FISH data shown in Fig. 4 strengthen the conclusion of a prevailing IL1b / IL1RN paracrine feedback loop controlling homeostatic and premalignant processes in the bone marrow. The identification of the important roles of IL1RN in restricting IL1b functions in the hematopoietic system advance our knowledge on the constitutive functions of the IL1 system and open new therapeutic avenues for treating hematopoietic malignancies. I congratulate the authors to this very interesting study.

We thank the Referee for the positive comments to our study. Thank you very much for the constructive and helpful feed-back during the revision of our work.

Reviewer #3 (Remarks to the Author):

The authors have done an excellent job addressing prior concerns. Most notably, the addition of new scRNAseq data (revealing the sources of IL1rn and the consequences of its loss), new analyses of their RNAseq data, and addition of analyses of additional NRAS initiated leukemogenesis mouse models, greatly strengthen and expand the impact. This manuscript makes important contributions in showing how IL1rn restricts normal myelopoiesis and leukemogenesis.

We thank the Referee for the positive comments to our work. Thank you very much for your support, it has been a pleasure to interact with you during the revision of our work and receive both thorough and kind feed-back from you.

Reviewer #4 (Remarks to the Author):

We thank the Referee for the revision of our scRNA-Seq data, which has helped us improve both the visualization and conclusions of this section.

If the authors want to look at what is different then they need to be looking at the differences within the clusters of the two mice? So what is different between for example between neutrophils in Wt vs IL-1rn KO mice?

The purpose of the scRNA-Seq was to identify the sources of IL-1 β and IL-1rn in the bone marrow at the single cell level. Following Referee #2 recommendation, we studied the expression of *Il1b* and *Il1rn* using single-cell techniques, including both scRNA-Seq and RNA-FISH in FACS-sorted cells.

We are now providing a new Supplementary Figure S5 where we look at differences within the clusters from IL-1rn-KO versus WT mice.

Please see lines 274-285: Ingenuity Pathway Analysis (IPA) of "Diseases and Functions" of the different CD11b⁺ myeloid clusters from IL-1rn-KO versus WT mice revealed significant enrichment in pathways related to inflammation, proliferation, migration, endocytosis and survival (Supplementary Fig. S5A). Inflammation was a major pathway activated in CD11b⁺ myeloid clusters from IL-1rn-KO versus WT mice, with the exceptions of G1 neutrophils that showed no changes and type I monocytes that showed a reduced inflammatory signature (Supplementary Fig. S5A). Among the differentially expressed genes responsible for these enriched signatures, we found upregulation of several major pro-inflammatory genes (*Hif1a*, *Csf2rb*, *Myd88*, *Cxcr2*, *Lmo4*) 40-44 as well as

downregulation of anti-inflammatory genes (*Nfkbia*, *Cebpb*) 45,46. A few genes that control inflammation (*Lyn*) 47 were upregulated in CD11b+ myeloid clusters from IL-1rn-KO versus WT mice (Supplementary Fig. S5B).

Lines 303-315: IPA of “Diseases and Functions” of the different LSK clusters from IL-1rn-KO versus WT mice revealed significant enrichment in pathways related to tumorigenesis and cell death which were activated in HSC, HSC-MPP1, MPP4 and MPP5 (Supplementary Fig. S5C). MPP2 and MPP3 behaved differently and showed significant activation of pathways related to survival (Supplementary Fig. S5C). Further, tumorigenic pathways were activated in MPP2 but slightly reduced in MPP3 (Supplementary Fig. S5C). Looking into the differentially expressed genes responsible for these enriched signatures, we found upregulation of important genes known to be involved in AML (*Nmt1*, *Ifitm3*, *Crip1*, *Cd52*) 48-51 and downregulation of genes whose reduced expression is a common feature of acute myeloid leukemogenesis (*Junb*) 52. We also found upregulation of genes with antiproliferative role (*Ifitm1*) 53 and downregulation of genes whose loss promote cell exhaustion (*Hlf*) 54 in LSK clusters from IL-1rn-KO versus WT mice (Supplementary Fig. S5D).

Lines 325-339: IPA of “Diseases and Functions” of the CD63+ stromal cluster fibroblasts from IL-1rn-KO versus WT mice revealed significant enrichment in pathways related to differentiation into adipocytes, chondrocytes and osteocytes that were activated whereas gene sets coordinating functions like proliferation and inflammation were reduced, as opposed to MSC (Supplementary Fig. S5F). Among the differentially expressed genes responsible for the enriched differentiation signatures, we found important genes involved in adipocyte (*Nr1d1*, *Zbtb16*, *Nfia*) 55-57, chondrocyte (*Hspg2*, *Bmp4*, *Col3a1*) 58-60 and osteocyte lineage differentiation (*Fbn1*, *Twist1*, *Spp1*) 61-63 (Supplementary Fig. S5G). All arteriolar, sinusoidal and pericyte clusters showed significant enrichment and activation in pathways associated with angiogenesis (Supplementary Fig. S5H). In addition, the sinusoidal cluster displayed activation of pathways related to inflammation, atherosclerosis and recruitment of leukocytes (Supplementary Fig. S5H). We checked the differentially expressed genes responsible for these enriched signatures and found upregulation of major pro-inflammatory genes in the CD63+ sinusoidal cluster from IL-1rn-KO mice versus WT (*Il1b*, *Il6*, *Nlrp3*, *Csf1*, *Ccl4*, *Ccl6*, *Ccl9*, *Osm*) (Supplementary Fig. S5I).

Figure 4A: CD11b should not be expressed on mouse hematopoietic stem cells, so I am unsure as to why such a large percentage of the UMAP in Figure 4A would be labelled as stem cells. This is more of a cell identification issue, rather than a data/overall analysis issue.

CD11b is expressed in HSPCs, with higher expression in MPP1 than in HSC (Cabezas-Wallsheid et al., 2014). However, we agree with the Referee that such a large fraction of cells labelled as stem cells relative to monocytes and neutrophils suggested a cell identification issue. Thank you for pointing it out.

We have now refined the annotation of CD11b+ cells. Please see new Figure 4A. Neutrophil-like cells have been relabelled as G0-G4 representing neutrophils differentiating in BM from the most immature G0 that would consist mainly of granulocyte monocyte progenitors to the most mature G4 population of neutrophils, according to cell type signatures imported from the original analysis of Xie et al. (2020; GSE137539). Monocyte-like cells have been relabelled as monocytes I and II, common monocyte progenitors, dendritic cell precursors and a minor fraction of HSC, according to cell type signatures imported from the original analysis of Krenkel et al. (2019; GSE131834). Briefly, WT cells from each cluster have been used to score the gene signatures and apply the maximum enrichment score ± 0.02 to label the cluster, following the same protocol described in Sommerkamp et al. (2021) and subsequent manual curation.

Annotation of the cell types of all the scRNA-seq analysis needs improvement.

To improve clustering and annotation of samples in each experiment we have decided to reanalyze each experiment (CD11b+, LSK and CD63+ stromal cells) applying an integration algorithm based on CCA as implemented in Seurat R package, to reduce any potential technical bias due to FACS-sorting of cells from IL-1rn-KO and WT mice at different days. After this integration, a new process

of clustering has been applied and, based on those clusters, subsequent cell type annotations have been performed as described above using the method of Sommerkamp et al. (2021). Please see complete description of the method in the Material and Methods section, lines 1120-1143.

After this reanalysis only the LSK experiment has changed more remarkably, probably due to a major effect of IL-1rn deletion on those cell types, as all the QCs on the library preparation and sequencing look very similar (please see below). The other experiments had minor adjustments compared to the non-integrated previous analysis.

Figure 4A: Within the cells labelled as “neutrophil” and “monocyte” there appears to be an effect of the IL-1rn KO as the cells are not integrated but rather are next to each other on the UMAP, is this a batch effect in the data integration?

We did not perform any kind of data integration in the previous version of our manuscript.

The experimental design of our scRNA-Seq analyses was as follows. For FACS-sorted CD11b+ cells and HSPC, BM nucleated cells were obtained by flushing BM from 1 femur from one mouse and 1 tibia from another mouse of the same genotype pooled together through centrifugation. The rest of the bones from both mice were pooled for CD63+ stromal cell sorting. The same protocol was performed for samples obtained from mice of both genotypes at similar times of the day, in different days close in time. Sorting of both genotypes was not performed the same day to prevent differences in the quality of the cells of both genotypes, derived from the waiting times of the sorting, which would have resulted in remarkable batch effect. The libraries were performed simultaneously.

Although we cannot discard a minor batch effect in our scRNA-Seq data, we have several evidence that argue against major batch effect. We will discuss below on all our scRNA-Seq data together as this is a comment that has been brought up by the Referee several times and for all cell subsets.

1. High and similar quality and similar quantity of the cDNA samples derived from similar cell types obtained from mice of both genotypes and used to generate the libraries (Figure 1 below).

LSK
 WT: 4 ng/ul
 IL-1rn-KO: 5.74 ng/ul

CD63+
 WT: 3.77 ng/ul
 IL-1rn-KO: 2.31 ng/ul

Figure 1. Quality profiles and quantity of cDNA samples from each cell type and genotype used for library generation and scRNA-Seq. Bioanalyzer profiling and measurements are shown. Myeloid, Upper panel; HSPC, middle panel; Stromal cells, lower panel.

2. High and similar quality of the 10x data derived from similar cell types obtained from mice of both genotypes (Figure 2 below). Estimated numbers of cells were in accordance with numbers of FACS-sorted cells.

CD11b_WT

CD11b_KO

LSK_WT

LSK_KO

Figure 2. Summaries of quality controls of 10x Genomics data from each cell type and genotype after scRNA-Seq. Summaries from Cell Ranger count are shown. Myeloid, Upper panel; HSPC, middle panel; Stromal cells, lower panel.

3. A batch effect caused by extracting cells on different days would be visible in the t-SNEs across all cell types and clusters. However, the overlap in the visualization of our scRNA-Seq data was dependent on the cell types and clusters, being high for CD63+ clusters, total for the previous cluster of precursors within CD11b+ cells but partial for neutrophils and monocytes within the same scRNA-Seq data, and absent for LSK clusters. Thus, in fact, t-SNEs of the same cell types but different genotypes are identical or very similar independently of the day of extraction of the cells, without data integration.

4. A batch effect caused by extracting cells of each genotype on different days would result in at least partially similar DEG across all cell types in IL-1rn-KO versus WT mice. However, the DEG that we observe in our scRNA-Seq data are specific of cell type and cell cluster.

However, taking the Referee concern into consideration, we have decided to reanalyze each experiment (CD11b+, LSK and CD63+ stromal cells) applying an integration algorithm based on CCA as implemented in Seurat R package. After this integration, a new clustering has been applied and subsequent cell type annotations have been performed using the method of Sommerkamp et al. (2021). Please see complete description of the method in the Material and Methods section, lines 1120-1143. After this reanalysis only the LSK experiment has changed substantially, probably due to a major effect of IL-1rn deletion on the progenitor cell subsets. The other FACS-sorted experiments had only minor adjustments.

It is not clear from the text that the authors have used the data to be able to identify the clusters, it rather sounds as though they are re-using the data from the referenced article.

We have used the data of the referenced articles to identify the clusters in our scRNA-Seq data. We have reworded explaining that we have labeled the clusters in our scRNA-Seq using the cell type signatures imported from the specified studies. Please see lines 265-269, 294-297 and 321-322.

There are several issues with the label transfer used to be able to identify the clusters within the data.

We have not performed label transfer in this version of our manuscript. We have reannotated following the maximum enrichment score ± 0.2 as previously described by Sommerkamp et al. (2021) for all experiments, using gene signatures of interest for each FACS-sorted experiment imported from different previous manuscripts. Please see complete description of the method in the Material and Methods section, lines 1120-1143.

In addition to the IPA, it would be advantageous to know which specific genes are actually included in this analysis, which genes are responsible for giving these enriched or reduced signatures?

Complete lists of detected genes, including differentially expressed genes responsible for signatures have been included for all cell clusters. Please see new Supplementary Table S5-S7.

Again in Figure 4E, mouse models do not overlap, if the authors want to show a difference in composition of the MPP cells, why did they not do this with FACS?

The models overlap now after application of an integration algorithm based on CCA as implemented in Seurat R package. A new process of clustering has been applied and subsequent cell type annotations have been performed using the method of Sommerkamp et al. (2021). Please see complete description of the method in the Material and Methods section, lines 1120-1143, and new Figure 4E.

We in fact performed both, immunophenotype (Fig. 2B; Supplementary Fig. S2D) and cluster annotation of our scRNA-Seq data identified according to HSPC subset signatures in E-MTAB-9208 dataset (Fig. 4E). Both were presented in the previous version of our manuscript.

The purpose of the scRNA-Seq was to identify the cell sources expressing IL-1 β and IL-1rn in the bone marrow at the single cell level. For accuracy on these cell sources, we needed to identify the cell clusters first and then check the expression of the cytokines of interest per cluster.

FACS analysis of HSPC revealed reduction of HSC and MPP5, and increase of MPP2 in the BM of IL-1rn-KO mice (Fig. 2B; Supplementary Fig. S2D). Reanalysis of the scRNA-seq data has now allowed us to improve the identified clusters into HSC, HSC-MPP1, MPP2, MPP3, MPP4 and MPP5 (Fig. 4E). The proportions of BM LSK cells after clustering of the scRNA-Seq data shows similar variation to the immunophenotypic analysis in IL-1rn-KO versus WT mice, with reductions in HSC and MPP5, and expansion of MPP2 (Fig. 4E). The scRNA-Seq data further reveals expansion of MPP3 transcriptional program and reduction in lymphoid-biased MPP4 in the BM LSK compartment of IL-1rn-KO mice versus WT (Fig. 4E).

Of note, MPP can be reprogrammed under certain circumstances. For example, this is the case of lymphoid-biased MPP4 fate that is reprogrammed towards the myeloid lineage under conditions of regeneration (Pietras et al, 2015), and this may happen in the absence of remarkable changes in the limited amount of membrane markers used to identify MPP4. Thus, we believe that immunophenotype and annotation of cell types after clustering of scRNA-Seq data are complementary and equally relevant in the context of hematopoiesis.

Is the UMAP shown in Figure 4 E including both the WT and KO mouse? If this is the case that the cellular populations do not overlap and there is something incorrect with the label transfer, batch correction or analysis.

The data have been reanalyzed, please see new Figure 4E. Briefly, raw counts from WT and IL-1rn-KO samples were merged and low-quality cells were filtered removing those with less than 2000 counts, less than 800 genes and more than 10% mitochondrial genes content. WT and KO samples were then normalized, log transformed and integrated together using the Canonical Cross-correlation Analysis algorithm with default parametrization as implemented in Seurat R package to avoid technical biases. Integrated samples were then clustered using Louvain algorithm over a SNN graph using 20 PCs and a 0.5 resolution. tSNE based dimensionality reduction was used for representation purposes. WT cells from each cluster were used for annotation. Gene modules from Sommerkamp et al. (2021) were scored over cells using the AddModuleScore from Seurat package.

Cluster assignment to each cell type was obtained using the maximum enrichment score ± 0.2 as described by the authors.

If the IL-1rn KO mouse has an increase in myeloid biased MPP2-MPP3 cells, can this be seen downstream, for example were there more CD11b+ cells in the KO mouse?

FACS analysis also showed an increase of MPP2 in the BM of IL-1rn-KO mice (Fig. 2B; Supplementary Fig. S2D). Indeed, there is a myeloid differentiation bias with an increase in CD11b+ cells identified by FACS in the bone marrow of IL-1rn-KO versus WT mice. Please refer to Figure 2D of the manuscript.

The scRNA-seq analysis of LSK cells is based on n=1

The purpose of the scRNA-Seq was to identify the sources of IL-1 β and IL-1rn in the bone marrow at the single cell level. Following Referee #2 recommendation, we studied the expression of *Il1b* and *Il1rn* by single-cell techniques, including both scRNA-Seq and RNA-FISH in FACS-sorted cells. We decided to use FACS-sorted cells given that in situ methods would not allow to dissect cell compartments of interest simultaneously to their expression of *Il1b* and *Il1rn*, particularly considering the numbers of markers required for HSC identification. Although this way we miss the information of local interactions, FACS-sorting allows better identification of the cell sources of *Il1b* and *Il1rn*. We also combine two techniques for better accuracy, meaning scRNA-Seq which allows quantification but depicts only highest expressing cells in a minimum percentage of 10% of the cells, with imaging as suggested by Referee #2 by highly sensitive RNA-FISH that in turn does not allow quantification as only a limited number of cells is imaged. Given that two complementary techniques were used with the same purpose, in case of LSK cells we used n=3 (3 mice) per experimental group in RNA-FISH and n=1 (2 mice) per experimental group for scRNA-Seq.

There was however a mistake in the report of how FACS-sorted HSPC were prepared for scRNA-Seq. We have now corrected this mistake, we apologize for the confusion. For scRNA-Seq of FACS-sorted HSPC, BM nucleated cells were obtained by flushing BM from 1 femur from one mouse and 1 tibia from another mouse of the same genotype pooled together through centrifugation. So, n=1 is derived from 2 mice.

When the majority of differentially expressed genes should be identified, it is of course preferable to have as many replicates per condition as possible. However, it is also a standard procedure in the field of HSPC to use one single pool of BM as input for scRNA-Seq analyses, from at least 2 mice of the same condition (Herault et al., 2021; Dahlin et al., 2018; Rodriguez-Fraticelli et al., 2018).

The UMAPs in 4E and Supplementary Figure 5c are not the same.

Thanks for noticing this mistake, this t-SNE has been corrected (new Supplementary Figure S5E).

In addition to the pie charts for the expression of *Il1rn* and *Il1b*, would actual plots of the expression be more advantageous to the reader as this would give a feel for the levels of expression, then it would be possible to also see the levels of expression of both factors, (*Il1b* on y-axis and *Il1rn* on x-axis).

We agree with the Referee that both are relevant, and the actual t-SNE plots with the visual representation of the expression of *Il1b* and *Il1rn* were found in Figure 4B (myeloid) and 4H (stromal cells), in addition to Supplementary Figure 5C (HSPC), of the previous version of our manuscript. Higher color intensity of the events in the t-SNE are indicative of higher levels of expression (color bars, $\log(\text{NormCounts})$). To see levels of expression of both factors in a specific cell type in WT, please simply check both right and left panels of the same figure. t-SNE plots are a standard way to visualize single gene expression levels for sets of genes in scRNA-Seq data. In the corresponding figure legends, we have added “t-SNE plots visualizing expression *levels* of *Il1b* and *Il1rn*” for accuracy.

Following the Referee recommendation, we have tested the visual representation of both factors in WT, with *Il1b* on y-axis and *Il1rn* on x-axis (Figure 3 below). The plots are visually poor as many of

the dots overlap and hence we lose the visual information of the fraction of cells expressing one or both cytokines at different levels, which is also relevant information. To try to solve this, we have added a histogram to the axes representing the density of points. For myeloid and stromal cells, the plots show cells expressing only *Il1b* or *Il1rn*, or both cytokines, at a great range of levels of expression. We believe that this kind of representation adds little value over the t-SNE already presented in the manuscript, so we have kept the t-SNE in the paper and hope that the Referee will agree with us.

Figure 3. Scaled expression levels of *Il1b* and *Il1rn* in wild-type (WT) cells. Scatter plots to visualize the expression levels of *Il1b* and *Il1rn* simultaneously in WT cells, as measured by scRNA-Seq. HSPC, left panel; Myeloid, middle panel; Stromal cells, right panel.

The authors state “At single-cell resolution, these results suggest that IL-1 β and IL-1rn control homeostasis and inflammatory responses in primitive HSC and MPP5, expand myeloid output through the control of transcriptomic pathways in MPP2-MPP3 and repress lymphoid output on the cluster MPP1 leading to unidentifiable prototypical MPP4 transcriptional program.” But in the introduction to the article they claim that there are no differences in the blood system and only observed changes under specific conditions. “IL-1 receptor 1 knockout (IL-1r1-KO) mice have unaffected blood production and normal stem and progenitor bone marrow (BM) compartments, suggesting that IL-1 β -induced myeloid priming of HSC occurs under conditions of injury or infection only and tonic IL-1 signaling has none or small basal hematopoietic effects”.

Here, we explored the role of the endogenous repressor cytokine of IL-1 β , IL-1 receptor antagonist (IL-1rn), in both healthy and abnormal hematopoiesis. Indeed, our data challenge the current dogma and demonstrate that HSC differentiation is controlled by balanced IL-1 β /IL-1rn levels under steady-state. This means that IL-1r1 is mainly repressed by IL-1rn under basal healthy conditions, which allows only tonic IL-1 β signaling and explains the minor effects of IL-1r1 deletion on hematopoiesis under steady-state (Please see schematic model). This is demonstrated by the finding that deletion of IL-1rn alone drives IL-1 β -induced HSC differentiation into the myeloid lineage, in the absence of injury or infection. We carefully controlled for this and our mice were maintained under SOPF environment and were not treated with IL-1 β , TLR agonists or any other inflammatory signal. We further provide evidence that IL-1rn deregulation occurs during experimental pre-leukemic myelopoiesis in mice, that it contributes to disease, and that it originates from both the hematopoietic and stromal compartments of the bone marrow. We show the prognostic value of IL1RN in a publicly available big cohort of AML patients (GSE14468), as well as in matched-pair diagnosis-relapsed AML patients (dbGap phs001027). We further provide translational data in primary samples from patients showing, for the first time, IL1RN deregulation in newly diagnosed AML patients.

Please note that the results and conclusions of the scRNA-Seq data related to HSC and MPP have been corrected in view of the reanalysis performed as specified in previous points.

The claims in line 295 – 298 are complete speculation, the authors show no evidence for these claims.

We have corrected the results and conclusions of the scRNA-Seq data based on the refined reanalysis as specified in previous points.

Please see lines 298-315: The proportions of BM LSK cells after clustering of the scRNA-Seq data showed similar changes to the immunophenotypic analysis of BM LSK in IL-1rn-KO versus WT mice

(Fig. 2B), with reduction in HSC and MPP5, and expansion of MPP2 (Fig. 4E). The scRNA-Seq data further revealed expansion of the MPP3 transcriptional program and reduction in lymphoid-biased MPP4 in the BM LSK compartment of IL-1rn-KO mice versus WT (Fig. 4E). IPA of “Diseases and Functions” of the different LSK clusters from IL-1rn-KO versus WT mice revealed significant enrichment in pathways related to tumorigenesis and cell death which were activated in HSC, HSC-MPP1, MPP4 and MPP5 (Supplementary Fig. S5C). MPP2 and MPP3 behaved differently and showed significant activation of pathways related to survival (Supplementary Fig. S5C). Further, tumorigenic pathways were activated in MPP2 but slightly reduced in MPP3 (Supplementary Fig. S5C). Looking into the differentially expressed genes responsible for these enriched signatures, we found upregulation of important genes known to be involved in AML (*Nmt1*, *Ifitm3*, *Crip1*, *Cd52*) 48-51 and downregulation of genes whose reduced expression is a common feature of acute myeloid leukemogenesis (*Junb*) 52. We also found upregulation of genes with antiproliferative role (*Ifitm1*) 53 and downregulation of genes whose loss promote cell exhaustion (*Hlf*) 54 in LSK clusters from IL-1rn-KO versus WT mice (Supplementary Fig. S5D).

We therefore concluded in the Discussion section, lines 505-507: These opposing and cell type-specific responses in HSPC subsets may be main factors underlying the chronic nature of the hematopoietic abnormalities in IL-1rn-KO mice, which will require additional investigation.

Figure 4G: The authors label “sinusoidal cells”, but there are many clusters of cells which are spread across the UMAP, it would be better to distinguish these different sub-populations rather than labelling them all the same, as the different clusters change between the WT and KO mice.

Thanks for this comment, which led us to revise and improve clustering of the stromal CD63+ experiment. We have now applied filtering to remove low quality cells (low number of genes detected, high MT, low counts), which were the reason of the sparse outcome of the previous analysis. Cells with less than 800 genes, less than 2000 counts and more than 10% of MT genes were removed from the analysis. WT and IL-1rn-KO samples were integrated using CCA methodology and clustering was performed over integrated cells. After this unbiased clustering, cells were annotated using cell type gene signatures obtained from Scadden (Baryawno et al, 2019) dataset and applying the Sommerkamp et al (2021) methodology (maximum gene module enrichment score ± 0.02) followed by manual curation.

The authors do not mention in the materials and methods the numbers of mice used for each experiment. In the figure legends it appears that the scRNA-seq was performed on n=1, this is not sufficient, there could be animal to animal variation.

The numbers of mice used in the scRNA-Seq were provided both in the manuscript, section “Mouse hematopoietic and stromal cell fraction extraction”, page 22, as well as in the reporting summary, section “Sample preparation”, page 8. There was however a mistake in the report of how FACS-sorted CD11b+ myeloid cells and HSPC were prepared for scRNA-Seq. We have now corrected this mistake, apologies for the confusion.

For scRNA-Seq of FACS-sorted CD11b+ cells and HSPC, BM nucleated cells were obtained by flushing BM from 1 femur from one mouse and 1 tibia from another mouse of the same genotype pooled together through centrifugation. The rest of the bones from both mice were pooled for CD63+ stromal cell sorting to be used for scRNA-Seq. Thus, all scRNA-Seq was performed on n=1 derived from 2 mice to reduce animal to animal variation.

REVIEWERS' COMMENTS

Reviewer #4 (Remarks to the Author):

Endogenous IL-1 receptor antagonist restricts healthy and malignant myeloproliferation

Alicia Villatoro, Vincent Cuminetti, Aurora Bernal, Carlos Torroja, Itziar Cossío, Alberto Benguría, Marc Ferré, Joanna Konieczny, Enrique Vázquez, Andrea Rubio, Peter Utnes, Xiaona You, Christopher G. Fenton, Ruth H. Paulssen, Jing Zhang, Fátima Sánchez-Cabo, Ana Dopazo, Anders Vik, Endre Anderssen, Andrés Hidalgo, and Lorena Arranz.

Figure 4A: Within the cells labelled as "neutrophil" and "monocyte" there appears to be an effect of the IL-1rn KO as the cells are not integrated but rather are next to each other on the UMAP, is this a batch effect in the data integration?

I thank the authors for replying with such a detailed response. I apologise if my comments were not clear, what I was trying to get at was that the populations were not in the same location within the multi-dimensional space within the t-SNE plots, so the integration of the WT and IL-1rn-KO cells. This has now been rectified with the new analysis pipeline.

In many articles, $n=2$, but these would be two independent 10x samples. By mixing the two mice together it is impossible to know which cells come from which mouse, which defeats the point in running two biological replicates unless they can be retrospectively identified. However, in this case all of the cells from the WT or IL-1rn-KO mice cluster together so no differences can be seen between the individual mice.

There are several issues with the label transfer used to be able to identify the clusters within the data.

"We have not performed label transfer in this version of our manuscript. We have reannotated following the maximum enrichment score ± 0.2 as previously described by Sommerkamp et al. (2021) for all experiments, using gene signatures of interest for each FACS-sorted experiment imported from different previous manuscripts. Please see complete description of the method in the Material and Methods section, lines 1120-1143."

Thank you for providing the detail of how the clusters were identified. Could the authors please show, similarly to the Sommerkamp paper the output of this comparison, as the cluster labels were not definitively labelled as one cell type or another, as there was significant overlap of the gene expression signatures between several of the clusters in the Sommerkamp paper.

The claims in line 295 – 298 are complete speculation, the authors show no evidence for these claims.

"We have corrected the results and conclusions of the scRNA-Seq data based on the refined reanalysis as specified in previous points.

Please see lines 298-315: The proportions of BM LSK cells after clustering of the scRNA-Seq data showed similar changes to the immunophenotypic analysis of BM LSK in IL-1rn-KO versus WT mice (Fig. 2B), with reduction in HSC and MPP5, and expansion of MPP2 (Fig. 4E). The scRNA-Seq data further revealed expansion of the MPP3 transcriptional program and reduction in lymphoid-biased MPP4 in the BM LSK compartment of IL-1rn-KO mice versus WT (Fig. 4E). IPA of "Diseases and Functions" of the different LSK clusters from IL-1rn-KO versus WT mice revealed significant enrichment in pathways related to tumorigenesis and cell death which were activated in HSC, HSC-MPP1, MPP4 and MPP5 (Supplementary Fig. S5C). MPP2 and MPP3 behaved differently and showed significant activation of pathways related to survival (Supplementary Fig. S5C). Further, tumorigenic pathways were activated in MPP2 but slightly reduced in MPP3 (Supplementary Fig. S5C). Looking into the differentially expressed genes responsible for these enriched signatures, we found upregulation of important genes known to be involved in AML (Nmt1, Ifitm3, Crip1, Cd52) 48-51 and downregulation of genes whose reduced expression is a common feature of acute myeloid leukemogenesis (Junb) 52. We also found upregulation of genes with antiproliferative role (Ifitm1) 53 and downregulation of genes whose loss promote cell exhaustion (Hlf) 54 in LSK clusters from IL-1rn-KO versus WT mice (Supplementary Fig. S5D)."

The authors need to tone down the claims in regard to MPP3 and MPP4, whilst there appears to be a trend towards more MPP3 in the facs analysis, the MPP4 cells also appear to be increased which does not match with the scRNA-seq analysis. The language used is also not strictly correct

“expansion of the MPP3 transcriptional program” perhaps transcriptional state would be a better word rather than program as this suggests that the genes which are associated with the signature are increased in expression rather than the number of cells which map to this signature are increased. For the other cell types, the alteration in abundance matches the phenotypical analysis but as it would be difficult to perform any statistical tests on the scRNA-seq analysis in terms of abundance, these claims need to be toned down.

Supplementary figure 5, Expression level ($\log_2(\text{NormCounts} + 1)$) plots. It would be better to show the plots just as violin plots, as this is obscured by the dots representing the individual cells. I am concerned about the significance (p-values) of some of the genes, as the violin plots are obscured.

REVIEWER COMMENTS-

Reviewer #4 (Remarks to the Author):

Endogenous IL-1 receptor antagonist restricts healthy and malignant myeloproliferation

Alicia Villatoro, Vincent Cuminetti, Aurora Bernal, Carlos Torroja, Itziar Cossío, Alberto Benguría, Marc Ferré, Joanna Konieczny, Enrique Vázquez, Andrea Rubio, Peter Utnes, Xiaona You, Christopher G. Fenton, Ruth H. Paulssen, Jing Zhang, Fátima Sánchez-Cabo, Ana Dopazo, Anders Vik, Endre Anderssen, Andrés Hidalgo, and Lorena Arranz.

Figure 4A: Within the cells labelled as “neutrophil” and “monocyte” there appears to be an effect of the IL-1rn KO as the cells are not integrated but rather are next to each other on the UMAP, is this a batch effect in the data integration?

I thank the authors for replying with such a detailed response. I apologise if my comments were not clear, what I was trying to get at was that the populations were not in the same location within the multi-dimensional space within the t-SNE plots, so the integration of the WT and IL-1rn-KO cells. This has now been rectified with the new analysis pipeline.

In many articles, $n=2$, but these would be two independent 10x samples. By mixing the two mice together it is impossible to know which cells come from which mouse, which defeats the point in running two biological replicates unless they can be retrospectively identified. However, in this case all of the cells from the WT or IL-1rn-KO mice cluster together so no differences can be seen between the individual mice.

We thank the Referee for the positive comment to our response and agree with the further remarks.

There are several issues with the label transfer used to be able to identify the clusters within the data.

"We have not performed label transfer in this version of our manuscript. We have reannotated following the maximum enrichment score ± 0.2 as previously described by Sommerkamp et al. (2021) for all experiments, using gene signatures of interest for each FACS-sorted experiment imported from different previous manuscripts. Please see complete description of the method in the Material and Methods section, lines 1120-1143."

Thank you for providing the detail of how the clusters were identified. Could the authors please show, similarly to the Sommerkamp paper the output of this comparison, as the cluster labels were not definitively labelled as one cell type or another, as there was significant overlap of the gene expression signatures between several of the clusters in the Sommerkamp paper.

Please see the output of the comparison below (Figure 1). The annotation was performed using WT cells only. We computed for each cluster the enrichment score of each HSPC subset signature from Sommerkamp (see table in Figure 1A). Then, we assigned each cluster to those HSPC subsets which scores are within a \$\pm 0.02\$ range of the maximum score for that cluster. Using this method, all clusters identified one single subset of HSPC but one single subset of HSPC was assigned to several clusters in most cases (Figure 1B). Cluster 4 showed relatively low maximal enrichment of MPP1 signature scoring (1.09) that was followed closely by HSC (0.92) (Figure 1A). We performed a subsequent manual curation where we confirmed the presence of more HSC genes in our annotated HSC than in MPP1, and more MPP1 genes in our annotated MPP1 cluster than in HSC. Yet, there

were many HSC genes in both our annotated HSC and MPP1 (Table 1). We believe the underlying reason for this is that the HSC signature from Sommerkamp contains more genes (1186) than the MPP1 signature (185). However, based on this manual confirmation, we think it is safer to annotate cluster 4 as HSC-MPP1.

A

Cluster		HSC	MPP1	MPP2	MPP3	MPP4	MPP5
1	C0	0.4956591	0.09544820	-1.6392242	-0.3744785	0.03823289	1.3843626
2	C1	-1.1569900	-0.39389237	0.7762195	1.4775725	0.13508632	-0.8379960
3	C2	1.3930155	0.34310055	-0.9168316	-1.0775953	-0.54413344	0.8024443
4	C3	-0.9564440	-0.32665550	1.6430082	0.6173281	-0.03011809	-0.9471188
5	C4	0.9188462	1.08995807	-1.1377132	-1.1228044	-0.31057899	0.5622923
6	C5	-0.1696194	-0.63330822	-0.9736716	-0.4108483	1.82202791	0.3654196
7	C6	-1.0546402	-0.35070475	1.2393450	1.1668292	-0.07237025	-0.9284591
8	C7	-1.1398883	-0.93208507	1.2479926	0.2490838	1.01101276	-0.4361159
9	C8	0.1345296	0.80816259	1.1953544	0.2058976	-1.40699543	-0.9369488
10	C9	-0.9726960	-0.64615225	0.8832335	1.5270330	-0.07071122	-0.7207070
11	C10	0.3516652	-0.12546839	-0.9078305	-1.2828688	0.50212334	1.4623792
12	C11	-0.5999811	0.04732552	1.6922597	0.5608429	-0.76735292	-0.9330941
13	C12	1.4018485	-0.08312775	-0.7980110	-1.1751415	-0.30365294	0.9580847

B

Figure 1. Expression of HSPC gene signatures in the clusters, as previously described by Sommerkamp et al. (2021). A) Table of scores. B) Analysis of maximal enrichment was used to assign colors and HSC/MPP identity to the clusters. Red circles indicate maximal enrichment of the signature scoring ± 0.02 in the respective clusters.

Table 1. Manual curation of cluster 4. Genes from MPP1 and HSC signatures from Sommerkamp et al. expressed in our MPP1 and HSC annotated clusters were manually checked. Cluster 4 showed relatively low maximal enrichment of MPP1 signature scoring (1.09) followed by HSC (0.92). Manual curation confirmed abundance of both HSC and MPP1 genes in cluster 4, so this cluster was reannotated as HSC-MPP1.

Genes from MPP1 signature expressed in MPP1 cluster		Genes from MPP1 signature expressed in HSC cluster	
Ahctf1	1.20748	Def8	0.724097919
Kit	6.350778235	Neh1	1.315588094
Larp4	1.494008021	Tmem40	0.642778521
Lars2	3.416841702		
Rad23b	1.287286597		
Runx1	2.260056101		
Slc6a6	1.245570876		
Zfp68	0.831258555		
Genes from HSC signature expressed in MPP1 cluster		Genes from HSC signature expressed in HSC cluster	
Abcg2	0.910033497	Abcg2	0.91913
Angpt1	8.87780115	Adgrg1	4.10658566
Ankrd12	2.722789733	Angpt1	4.783926314
Aplp2	1.22611925	Anxa5	1.075225662
Arglu1	8.433561332	Aplp2	1.187084196
Arid1b	1.024303592	Capg	1.101628418
Arid4a	2.447072866	Car2	9.231890398
Atp2b1	1.411252174	Card19	1.242290334
Atp2b4	2.767398244	Ccnd2	3.671732658
Bcl11a	4.000375811	Cd63	3.797011662
Birc2	1.33301353	Cd81	2.690133918
Cd81	2.762772342	Cirbp	1.946185627
Cd84	2.180303627	Cited2	3.469497826
Cers4	0.927329426	Cox7a2l	6.162598906
Chd2	2.91154173	Dynl13	0.839129192
Clp1	1.011808878	Elf4a2	1.649394198
Crebrf	1.524698536	Epb414aos	1.22991861
Ddx17	2.667688745	Fbxo9	0.964062688
Efcab14	1.3468080545	Frbp1l	1.004861187
Ehmt1	2.259453081	Gimap1	4.305615729
Elf4a2	1.628235049	Gimap5	0.884975502
Eng	1.199487872	Gimap6	2.471864789
Epb41	0.828029332	Gng11	2.080206991
Etv6	2.737034655	H2-K1	16.36572716
Fcho2	1.382899653	H2-Q4	0.975921355
Frbp1l	1.104275574	H2-Q7	2.915652793
Gata2	1.399803048	H2-T22	0.866355436
H2-K1	17.56685335	Hscd4	1.931294096
H2-Q4	1.22821765	Hmgd3	0.775317744
Helz	1.422425327	Hsd17b10	1.588073827
Igf1r	1.942519461	Ier2	1.354266852
Ith5	2.880164639	Ifltm1	11.3058996
Lamtor3	0.820724664	Ifltm3	10.10111643
Mafk	1.268450172	Irf2	2.756847038
Malat1	555.7052698	Irf9	0.900033981
Mast4	1.32284733	Krt18	1.533692607
Mctp1	1.6678336	Lmo2	7.620986264
Mettl7a1	1.002416633	Lst1	1.722441997
Mllt3	1.035686227	Ltb	3.99105238
Nfe2l1	1.490152685	Ly6a	2.271701839
Ocri	1.429897994	Malat1	285.9309934
Ogt	6.710876221	Map1lc3b	3.143193666
Parp14	1.369157823	Mecm	1.703811861
Pbx1	2.048397309	Mettl7a1	1.121229189
Pik3ip1	2.23032333	Mllt3	1.505853214
Pik3r1	1.846351405	Mpl	1.137387597
Plcb2	1.444129167	Mprl52	3.280070204
Plgrk	2.895691643	Nyc1	1.812854541
Pnscr	7.2478596	Nyl10	11.43695719
Por	1.220847139	Neur3	0.811287869
Pura	4.113338886	Nfe2	1.517200393
Rab11fip2	1.292291184	Nfkbia	1.792239552
Rb1cc1	2.505851322	Nkx2-3	0.988791614
Rbm5	5.822718608	Ormdl3	0.98464227
Rere	1.39076996	Pdtk1ip1	0.937201811
Rgs1	1.396380238	Pigo	1.292420051
Rock2	2.021292381	Pik3ip1	1.988456607
Samhd1	1.430401185	Pik3r1	1.671179663
Sgms1	1.25935213	Ppp1r15a	1.184596736
Wis	1.739007689	Procr	0.742016837
		Prr13	1.413402082
		Ptpn18	7.844603902
		Ptprcap	5.453704622
		Rab38	1.581149865
		Rbp1	1.698247132
		Rgs1	1.195490671
		Rpl21	58.32222789
		Rplp1	72.89769918
		Rps10	53.98619742
		Rps21	44.10860091
		Saraf	1.725059501
		Sgms1	0.831623369
		Slc48a1	0.730911497
		Smarca2	1.95043262
		Smpd13a	1.611969914
		Sord	0.827065963
		Stat1	0.9551009
		Tbxas1	1.007103355
		Tnfr3	0.958622033
		Tsc22d1	2.259283055
		Tsc22d3	1.314485811
		Txnip	7.459302479
		Uba7	0.911741269
		Use1	2.052388416
		Ypel3	1.882511636
		Zbtb20	2.20204047
		Zfand5	2.683811888
		Zfand6	1.454350007

The claims in line 295 – 298 are complete speculation, the authors show no evidence for these claims.

"We have corrected the results and conclusions of the scRNA-Seq data based on the refined reanalysis as specified in previous points. Please see lines 298-315: The proportions of BM LSK cells after clustering of the scRNA-Seq data showed similar changes to the immunophenotypic analysis of BM LSK in IL-1rn-KO versus WT mice (Fig. 2B), with reduction in HSC and MPP5, and expansion of MPP2 (Fig. 4E). The scRNA-Seq data further revealed expansion of the MPP3 transcriptional program and reduction in lymphoid-biased MPP4 in the BM LSK compartment of IL-1rn-KO mice versus WT (Fig. 4E). IPA of "Diseases and Functions" of the different LSK clusters from IL-1rn-KO versus WT mice revealed significant enrichment in pathways related to tumorigenesis and cell death which were activated in HSC, HSC-MPP1, MPP4 and MPP5 (Supplementary Fig. S5C). MPP2 and MPP3 behaved differently and showed significant activation of pathways related to survival (Supplementary Fig. S5C). Further, tumorigenic pathways were activated in MPP2 but slightly reduced in MPP3 (Supplementary Fig. S5C). Looking into the differentially expressed genes responsible for these enriched signatures, we found upregulation of important genes known to be involved in AML (Nmt1, Ifitm3, Crip1, Cd52) 48-51 and downregulation of genes whose reduced expression is a common feature of acute myeloid leukemogenesis (Junb) 52. We also found upregulation of genes with antiproliferative role (Ifitm1) 53 and downregulation of genes whose loss promote cell exhaustion (Hlf) 54 in LSK clusters from IL-1rn-KO versus WT mice (Supplementary Fig. S5D)."

The authors need to tone down the claims in regard to MPP3 and MPP4, whilst there appears to be a trend towards more MPP3 in the facs analysis, the MPP4 cells also appear to be increased which does not match with the scRNA-seq analysis. The language used is also not strictly correct "expansion of the MPP3 transcriptional program" perhaps transcriptional state would be a better word rather than program as this suggests that the genes which are associated with the signature are increased in expression rather than the number of cells which map to this signature are increased. For the other cell types, the alteration in abundance matches the phenotypical analysis but as it would be difficult to perform any statistical tests on the scRNA-seq analysis in terms of abundance, these claims need to be toned down.

These claims have been toned down. Please see lines 298-303: "The proportions of BM LSK cells after clustering of the scRNA-Seq data showed overall similar changes to the immunophenotypic analysis of BM LSK in IL-1rn-KO versus WT mice (Fig. 2B), with qualitative reduction in HSC and MPP5, and expansion of MPP2 (Fig. 4E). Unlike FACS quantification, the scRNA-Seq data showed qualitative expansion of the MPP3 transcriptional state and reduction in lymphoid-biased MPP4 in the BM LSK compartment of IL-1rn-KO mice versus WT (Fig. 4E)", and lines 497-500: "Our scRNA-Seq analyses confirmed overall the immunophenotypic changes in the proportions of BM HSPC cells but showed qualitative expansion of MPP3 transcriptional state and reduction in lymphoid-biased MPP4 cells in the BM LSK compartment of IL-1rn-KO versus WT mice."

Supplementary figure 5, Expression level (log2 (NormCounts + 1)) plots. It would be better to show the plots just as violin plots, as this is obscured by the dots representing the individual cells. I am concerned about the significance (p-values) of some of the genes, as the violin plots are obscured.

We are now showing the violin plots without individual cells as requested.